# GENERALIZING DYNAMICS MODELING EASIER FROM REPRESENTATION PERSPECTIVE

## ABSTRACT

Learning system dynamics from observations is a critical problem in many applications over various real-world complex systems, *e.g.,* climate, ecology, and fluid systems. Recently, the neural-based dynamics modeling method has become the prevalent solution, where its basic idea is to embed the original states of objects into a latent space before learning the dynamics using neural-based methods such as neural Ordinary Differential Equations (ODE). Given observations from different complex systems, the existing dynamics modeling methods offer a specific model for each observation, resulting in poor generalization. Inspired by the great success of pre-trained models, we raise a question: whether we can conduct a generalized **P**re-trained **D**ynamic **E**nco**DER** (**PDE**DER), which, for various complex systems, can embed their original states into a latent space, where the dynamics can be easier captured. To conduct this generalized PDEDER, we collect 153 sets of real-world and synthetic observations from 24 complex systems. Inspired by the success of time series forecasting using Pre-trained Language Models (PLM), we can employ any PLM and further update it over these dynamic observations by tokenization techniques to achieve the generalized PDEDER. Given any future dynamic observation, we can fine-tune PDEDER with any specific dynamics modeling method. We evaluate PDEDER on 18 dynamic systems by long/short-term forecasting under both in-domain and cross-domain settings and the empirical results indicate the effectiveness of PDEDER.

## 1 INTRODUCTION

System dynamics, which describes the object states evolving over time, is a powerful methodology to conceptualize complex systems from various domains, *e.g.,* climate, ecology, and fluid systems (Alon, 2006; Bashan et al., 2016; Gao et al., 2016; Gerstner et al., 2014; Li et al., 2019; Lu et al., 2018; Zang et al., 2016; 2018; 2019a;b). In parallel with physical methods, learning system dynamics from abundant observations has drawn much attention, and the neural-based dynamics modeling method such as neural Ordinary Differential Equations (ODE) become the representative Chen et al. (2018).

To our knowledge, the basic idea of the representative method is to embed the original states of objects into a latent space before learning the dynamics using neural-based methods and finally followed by a decoder Zang & Wang (2020). Although the existing methods have been successfully applied in various systems, most of them must train a specific model given observations from different systems, resulting in limited generalizability. To meet this challenge, several recent studies investigate generic methods that can simultaneously handle multiple dynamics from various systems and environments Kirchmeyer et al. (2022); Huang et al. (2023). Unfortunately, due to the potential huge gap between various dynamics, developing generic dynamics modeling methods faces significant challenges and is still an open problem.

Inspired by the great success of pre-trained models, we raise a question: instead of developing generic dynamics modeling methods, whether we can conduct a generalized **P**re-trained **D**ynamic **E**nco**DER** (**PDE**DER), which, for various complex systems, can embed their original states into a latent space, where the dynamics can be more easily captured. To conduct this generalized PDEDER, we collect 153 sets of real-world and synthetic observations from 24 complex systems. Inspired by the success of time series forecasting using Pre-trained Language Models (PLM), we can employ any

PLM and further update it over these dynamic observations by tokenization techniques to achieve the generalized PDEDER. Given any future dynamic observation, we can fine-tune PDEDER with any specific dynamics modeling method. We evaluate PDEDER on 18 dynamic systems by long/short-term forecasting under both in-domain and cross-domain settings and the empirical results indicate the effectiveness of PDEDER.

In a nutshell, the major contributions of this paper can be outlined below:

- We propose a novel idea of updating PLM to build a generic encoder PDEDER for dynamics modeling.
- We collect extensive real-world and synthetic observations from various complex systems to train PDEDER.
- We conduct numbers of experiments to evaluate PDEDER on by long/short-term forecasting under both in-domain and cross-domain settings.

## 2 RELATED WORK

**Dynamics Modeling Methods** Currently, mainstream dynamics modeling methods primarily fall into the data-driven manner. Zang & Wang (2020) combines neural ordinary differential equations Chen et al. (2018) with GNNs Wu et al. (2020) to approximate continuous-time dynamics of networks at an arbitrary time on the interaction graph. Shi et al. (2023) performs integral operations to the derivatives of the changing on time and spatial dimensions, demonstrating the ability of adapting to spatial and temporal dependencies. Huang et al. (2023) studies cross-environment learning of continuous multi-agent system dynamics. It models this using parameterized neural ordinary differential equations (ODEs)Chen et al. (2018), describing the dynamics of each system through shared ODE functions and environment-specific vectors for latent exogenous factors. Huang et al. (2020) learns dynamics from irregularly sampled partial observations. Wang & Yu (2023) argues that data-driven methods lack the ability of understanding hidden dynamics and responding to naturally occurring data distribution changes. It proposes incorporating prior physical knowledge into existing deep learning approaches to enhance model generalization in unknown environments. Kirchmeyer et al. (2022) associates different dynamics with multiple environments separately, adjusting the dynamic model based on context parameters specific to each environment, allowing for rapid adaptation and better generalization in low-sample environments. Gupta et al. (2022) decomposes complex systems into subsystems, modeling each subsystem as a neural ODE module and simulating various coupled topologies through the combination of these modules. Huang et al. (2024b) enhanced the modeling capability of physical systems through the time-reversal symmetry regularization term, improving the forecasting accuracy and robustness for complex systems.Luo et al. (2023) incorporates second-order graph convolution to capture non-neighborhood semantic information, as well as second-order graph ODEs to model higher-order temporal dependencies. Huang et al. (2024a)uses graph neural networks (GNN) as an ODE function to capture the dynamic effects of treatments over time and the combined effects of multiple treatments. Wu et al. (2024) provides a new approach for OOD fluid dynamics modeling and conducts extensive experiments on multiple benchmarks to validate the superiority of the method. Gravina et al. (2024) proposes to reinterpret existing graph neural networks as a discretized solution of an ODE, thereby extending them to handle graph streams with irregular sampling. Luo et al. (2024) proposed a novel graph ODE model that significantly enhances the modeling capability and generalization performance of multi-agent dynamical systems through the introduction of contextual prototypes.

**Pre-trained Language Models for Time Series Forecasting** Recently, PLMs have been successfully applied to various tasks. In handling sequence data tasks, Gruver et al. (2024) encodes time series as a string of numbers, forecasting the next token in the text to achieve sequence forecasting results, allowing time series input to adapt unilaterally to the input format of language models. PromptCast Xue & Salim (2023) converts the numerical inputs and outputs of time series into prompts, constructing forecasting tasks in a sentence-by-sentence manner. Nie et al. (2023) segments the time series into sub-sequence-level patches to serve as input for Transformers. AutoTimes Liu et al. (2024) transforms time series data into a format understandable by LLMs for auto-regressive forecasting. LLM4TS Chang et al. (2024) proposes a two-phase fine-tuning method, first aligning the model with the characteristics of the time series to better adapt the LLM, and then

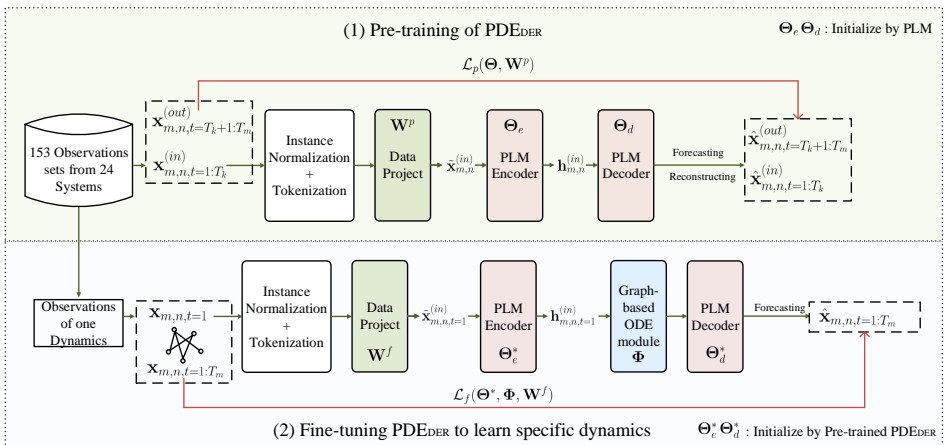

Figure 1: The flowchart of PDEDER.

further fine-tuning the model guided by downstream forecasting tasks in the second phase. Time-LLM Jin et al. (2024) requires no fine-tuning of any layers in the LLM, simply freezing the LLM and using two learnable modules to process inputs and outputs. Additionally, Zhou et al. (2023) provides a unified framework for various time series tasks. Zhang et al. (2024) proposed a framework that transitions from univariate pre-training to multivariate fine-tuning. Through self-supervised pre-training and cross-channel dependency fine-tuning, it demonstrates excellent performance across various time series tasks.

## 3 PRELIMINARIES

Commonly speaking, a dynamics contains a set of interacting objects whose states co-evolve over time on a weighted interaction graph. The dynamics could be formalized into a graph $\mathcal{G} = (\mathcal{V}, \mathcal{E})$, where the set of nodes $\mathcal{V} = \{\mathbf{x}_n\}_{n=1}^N$ consists of $N$ interacting objects and $\mathcal{E} = \{\mathbf{e}_{i,j}\}_{i,j=1}^N$ denotes the interactions among them. The observations of each node $\mathbf{x}_n$ is a trajectory of states along time $T$. On time $t$, the state of object $n$ can be represented as a vector $\mathbf{x}_{n,t} \in \mathbb{R}^V$ where $V$ is the system-specific state dimension. The evolution of object states are governed by some hidden regularities for the most part. Given the states observations $\mathcal{G}$, we aim to extract the hidden governing dynamics and can help forecast the states at an arbitrary time $t$.

**Ordinary Differential Equations (ODEs) for Dynamical System** Conventionally, the evolution of each object states in a dynamics system can be described by Ordinary Differential Equations (ODEs): $\dot{\mathbf{x}}_{n,t} := \frac{d\mathbf{x}_{n,t}}{dt} = g(\mathbf{x}_{1,t}, \ldots, \mathbf{x}_{N,t}; \mathcal{G})$, where $g(\cdot; \mathcal{G})$ is a hand-crafted function to model the characteristic from the observations; $\mathcal{G}$ denotes the objects interactions. The differential equations describe the instantaneous changing rate of each object state under mutual influences. Given the initial states of each object $\{\mathbf{x}_{1,t=1}, \ldots, \mathbf{x}_{N,t=1}\}$, the states at an arbitrary time point $\tau$ can be calculated by integrating the differential equation over timeline:

$$\mathbf{x}_{n,\tau} = \mathbf{x}_{n,1} + \int_1^\tau g(\mathbf{x}_{1,t}, \ldots, \mathbf{x}_{N,t}; \mathcal{G})dt. \tag{1}$$

The above integration is also called an ODE initial value problem Boyce et al. (2021) for this differential equation, which could be solved by numerical ODE solvers such as Euler's method, Dormand-Prince DOPRI5 Dormand (2018), Runge-Kutta Schober et al. (2019), *etc.* Then the dynamics model could be approximated with these numerical methods at an arbitrary time.

# 4 METHODOLOGY

## 4.1 OVERVIEW OF PDEDER

In this section, we introduce our proposed **P**re-trained **D**ynamic Enco**DER** (**PDEDER**). To learn an encoding model with outstanding generalizability, we first collect massive observations from both synthetic and real-world systems to ensure the diversity of training datasets. Then we pre-train PDEDER with the collected observations on two tasks. By ingesting the input observation into the model, we train PDEDER from two aspects, reconstructing the input observation, and forecasting unseen future states. With our pre-trained PDEDER, we can generate dynamics-enriched embeddings and approximate dynamics on these embeddings. Specifically, given the initial states of objects and their interactions, we encode the initial states by our pre-trained PDEDER into dynamics-enriched embeddings. Then we can learn dynamics on the interaction graph by approximating the observations using any dynamics modeling method. The flowchart is presented in Figure 1.

In this section, we first introduce the pre-training of our PDEDER. Secondly, we gave two examples of learning specific dynamics by fine-tuning PDEDER. We adopt two dynamics modeling methods as examples, including a black-box GNN-based neural method and a white-box method SINDy Brunton et al. (2021).

## 4.2 PRE-TRAINING OF THE PRE-TRAINED DYNAMIC ENCODER (PDEDER)

**Benchmark Generation** Firstly, we introduce the collection of dynamics observations. We collect 153 sets of observations including 122 synthetic sets from 14 systems with various hyper-parameters, and 31 sets of real-world observations from 10 systems. The domains consist of physics, fluid, biology, climate and traffic system. For each synthetic system $s \in [S]$, we set $P_s$ different parameter settings, including numbers of objects and sequence lengths. For each parameter setting, we generate $M_s$ sets of observations $\{\mathcal{G}_m = (\mathcal{V}_m, \mathcal{E}_m)\}_{m=1}^{M_s}$. $\mathcal{V}_m = \{\mathbf{x}_{m,n}\}_{n=1}^{N_m}$ denotes the observations of $N_m$ objects and $\mathcal{E}_m = \{< \mathbf{x}_{m,i}, \mathbf{x}_{m,j} >\}_{i,j \in [N_m]}$ denotes the interactions among them. The observations $\mathbf{x}_{m,n} \in \mathbb{R}^{T_m \times V_s}$ of object $n$ are denoted as a trajectory of states along time $T_m$, where $V_s$ denotes the system-specific state dimension. For example, on system "Charged", we set 4 different numbers of objects $\{5, 10, 15, 20\}$ and 2 different sequence lengths $\{400, 600\}$. We generate 8 sets of observations using all combinations of the two sets of parameters. Each of the 8 sets corresponds with different system-specific hyper-parameters. Then we generate 5000 observation sequences under each parameter setting with random initial values. For all systems, we vary the number of objects and sequence length both from $[5, 1024]$. The statistics of observations are illustrated in Table 1 and the detailed descriptions of each system are presented in Appendix A. To pre-train PDEDER on multiple tasks, We split the observations $\mathbf{x}_{m,n}$ into two sub-observations $\mathbf{x}_{m,n}^{(in)} = \{\mathbf{x}_{m,n,t}\}_{t=1}^{t=T_k}$ and $\mathbf{x}_{m,n}^{(out)} = \{\mathbf{x}_{m,n,t}\}_{t=T_k+1}^{t=T_m}$. By ingesting the $\mathbf{x}_{m,n}^{(in)}$, we learn PDEDER by reconstructing $\mathbf{x}_{m,n}^{(in)}$ and forecasting $\mathbf{x}_{m,n}^{(out)}$.

**Tokenization** To adapt the input observations with various lengths and serve as input tokens for transformer-based PLMs, following Nie et al. (2023), we tokenize the observed states into sub-observations to adapt the input states with various lengths. For object $n$, we patch the input states $\mathbf{x}_{m,n}^{(in)} \in \mathbb{R}^{T_k \times V_s}$ into $\overline{\mathbf{x}}_{m,n}^{(in)} \in \mathbb{R}^{P_m \times L_p \times V_s}$, where $L_p$ denotes the patch length; $P_m = \lfloor \frac{(T_k - L_p)}{R} \rfloor + 2$ denotes the number of patches and $R$ denotes the stride. In this manner, the trajectory lengths are reduced by $R$ times, which can simultaneously maintain the local semantics in long-term dynamics modeling and significantly reduce the space and time costs during model learning. Besides, to benefit domain adaptation and generalization, we add Gaussian noises and apply instance normalization before tokenization to handle the distribution shift among various domains following Kim et al. (2021).

**Data Projection** To handle dimension diversity of states across different systems, we adopt a flatten-linear data projection module to align the observations by mapping into same dimensions. For each patched tokens $\overline{\mathbf{x}}_{m,n}^{(in)} \in \mathbb{R}^{P_m \times L_p \times V_s}$, we first flatten it into $\overline{\mathbf{x}}_{m,n}^{(in)(fl)} \in \mathbb{R}^{P_m \times (L_p \cdot V_s)}$, and then project it by a linear layer into the dimension of $L_p$ for all systems $\tilde{\mathbf{x}}_{m,n}^{(in)} = f(\overline{\mathbf{x}}_{m,n}^{(in)(fl)}; \mathbf{W}_{dp}^s)$, where $\mathbf{W}_{dp}^s \in \mathbb{R}^{(L_p \cdot V_s) \times L_p}$ denotes the system-specific trainable parameters.

Table 1: Statistics of collected dynamics. $N_m$ denotes the number of objects; $T_m$ denotes the length of timestamp; $V_s$ denotes the feature dimension; $M_s$ denotes the number of samples generated; $P_s$ denotes the number of different hyper-parameter settings.

| System | Type | Domain | $N_m$ | $T_m$ | $V_s$ | $M_s$ | $N_p$ |
|---|---|---|---|---|---|---|---|
| Charged | Synthetic | Physics | {5,10,15,20} | {400,600} | 4 | 5000 | 8 |
| Springs | Synthetic | Physics | {20,25,30,35,40} | {200,300} | 4 | 3500 | 10 |
| Mutualistic | Synthetic | Physics | {100,121,169,196,225} | {300,350,400} | 1 | 1500 | 15 |
| Heat | Synthetic | Physics | {225,256,289,324} | {200,250,300} | 1 | 1500 | 12 |
| 1D Diff-Reaction | Synthetic | Fluid | {256,368,464,512} | {200,225,250,275} | 1 | 700 | 16 |
| 1D CFD | Synthetic | Fluid | {300,350,400} | {600,625} | 3 | 300 | 6 |
| 2D CFD | Synthetic | Fluid | {400,625,784,1024} | {100,150,200} | 4 | 200 | 12 |
| Burgers | Synthetic | Fluid | {400,425,450} | {512,768,960,1024} | 1 | 250 | 12 |
| Advection | Synthetic | Fluid | {500,550} | {700,725,750} | 1 | 500 | 6 |
| DarcyFlow | Synthetic | Fluid | {625,676} | {400,425,450} | 1 | 700 | 6 |
| Gene | Synthetic | Biology | {729,841,900,1024} | {125,150,175,200} | 1 | 500 | 16 |
| Shallow-Water | Synthetic | Fluid | 768 | 500 | 1 | 3000 | 1 |
| 2D Diff-Reaction | Synthetic | Fluid | 900 | 120 | 2 | 5000 | 1 |
| Diff-Sorption | Synthetic | Fluid | 1024 | 101 | 1 | 10000 | 1 |
| LA | Real-world | Climate | 274 | 384 | 10 | 1 | - |
| SD | Real-world | Climate | 282 | 384 | 10 | 1 | - |
| NYCTaxi | Real-world | Traffic | 75 | 17520 | 2 | 1 | - |
| CHIBike | Real-world | Traffic | 270 | 4416 | 2 | 1 | - |
| TDrive | Real-world | Traffic | 1024 | 3600 | 2 | 1 | - |
| PEMS03 | Real-world | Traffic | 358 | 26208 | 1 | 1 | - |
| PEMS04 | Real-world | Traffic | 307 | 16992 | 3 | 1 | - |
| PEMS07 | Real-world | Traffic | 883 | 28224 | 1 | 1 | - |
| PEMS08 | Real-world | Traffic | 170 | 17856 | 3 | 1 | - |
| NOAA | Real-world | Climate | {17,24,27,29,40,40, 40,46,49,53,55,65, 77,89,93,108,160, 179,199,216,225,253} | 7305 | 3 | 22 | - |

**Learn with PLM** With the projected $\tilde{\mathbf{x}}_{m,n}^{(in)}$, we reconstruct the input states and forecast future states by a pre-trained language model. First, we encode $\tilde{\mathbf{x}}_{m,n}^{(in)}$ by a convolutional layer $f(\cdot; \mathbf{W}_c)$ and the encoder of a PLM $f(\cdot; \mathbf{\Theta}_e)$. The convolutional layer is capable of maintaining the local semantic information and adapt the states dimension $H$ into which of PLM Chang et al. (2023). Then, we decode the hidden features by the decoder of PLM $f(\cdot; \mathbf{\Theta}_d)$ attached by two flatten-linear layers $f(\cdot; \mathbf{W}_r^s)$ and $f(\cdot; \mathbf{W}_p^s)$, which serves for reconstructing and forecasting, respectively. Detailed calculations are as below:

$$
\begin{aligned}
\mathbf{h}_{m,n} &= f(f(\tilde{\mathbf{x}}_{m,n}^{(in)}; \mathbf{W}_c); \mathbf{\Theta}_e), \\
\hat{\mathbf{x}}_{m,n}^{(in)} &= f(f(\mathbf{h}_{m,n}; \mathbf{\Theta}_d); \mathbf{W}_r^s), \quad \hat{\mathbf{x}}_{m,n}^{(out)} = f(f(\mathbf{h}_{m,n}; \mathbf{\Theta}_d); \mathbf{W}_p^s),
\end{aligned}
\tag{2}
$$

Finally, the model is learnt by minimizing the reconstructing loss against the original input states $\mathbf{x}_{m,n}^{(in)}$ and the forecasting loss against the ground-truth future states $\mathbf{x}_{m,n}^{(out)}$:

$$
\mathcal{L}_p(\mathbf{\Theta}, \mathbf{W}^p) = \sum_{s=1}^{S} \sum_{p=1}^{P_s} \sum_{m=1}^{M_s} \sum_{n=1}^{N_m} \left( \sum_{t=1}^{T_k} \ell(\hat{\mathbf{x}}_{m,n,t}^{(in)}, \mathbf{x}_{m,n,t}^{(in)}) + \sum_{t=T_k+1}^{T_m} \ell(\hat{\mathbf{x}}_{m,n,t}^{(out)}, \mathbf{x}_{m,n,t}^{(out)}) \right), \tag{3}
$$

where $\ell(\cdot)$ denotes the L1 loss; $\mathbf{\Theta} = \{\mathbf{\Theta}_e, \mathbf{\Theta}_d\}$ denotes the parameters set of the encoder/decoder of the PLM; $\mathbf{W}^p = \{\mathbf{W}_{dp}^s, \mathbf{W}_c, \mathbf{W}_r^s, \mathbf{W}_p^s\}_{s=1}^{S}$.

### 4.3 EXAMPLES OF LEARNING SPECIFIC DYNAMICS WITH PDEDER

We now introduce the usage of PDEDER when learning a specific dynamics. We introduce two examples of learning dynamics with a black-box GNN-based dynamics learner and a white-box dynamics learner SINDy Brunton et al. (2021).

Conventionally, given the observations of $N_d$ objects $\{\mathbf{x}_{m,1}, \ldots, \mathbf{x}_{m,N_d}\}$ across time $T_m$, we can model the hidden dynamics by solving the ODE initial value problem with the initial observations $\{\mathbf{x}_{m,1,1}, \ldots, \mathbf{x}_{m,N_d,1}\}$ as mentioned in preliminaries. Following the pre-training processes in PDEDER, we tokenize and project the states into $\{\tilde{\mathbf{x}}_{m,1}, \ldots, \tilde{\mathbf{x}}_{m,N_d}\}$ and adopt the first token $\tilde{\mathbf{x}}_{m,n,1} \in \mathbb{R}^{L_p}$ as the initial value to learn dynamics. Then we encode the initial observations to $\mathbf{h}_{m,n,1} \in \mathbb{R}^H$ by the encoder of pre-trained PDEDER $\mathbf{h}_{m,n,1} = f(f(\tilde{\mathbf{x}}_{m,n,1}; \mathbf{W}_c); \mathbf{\Theta}_e^*)$, where $\mathbf{\Theta}_e^*$ denotes the pre-trained parameters of encoder in PDEDER.

**Example: Fine-tune PDEDER with a Black-box Dynamics Learner.** To model the dynamics where the objects affect each other along with evolution, following Zang & Wang (2020), we adapt a GNN-based module $g(\cdot)$ to model dynamics by incorporating the interactions among objects in the latent space. Let $\mathbf{A}_m \in \mathbb{R}^{N_m \times N_m}$ denotes the adjacent matrix of the interaction graph $\mathcal{G}_m$ and $\mathbf{h}_{m,\cdot,\tau} = [\mathbf{h}_{m,1,\tau}, \ldots, \mathbf{h}_{m,N_m,\tau}] \in \mathbb{R}^{N_m \times H}$ denotes the embeddings at an arbitrary time $\tau$ ($1 < \tau \leq T_m$), we describe the dynamics by the following ODE:

$$\frac{d\mathbf{h}_{m,\cdot,\tau}}{dt} = g(\mathbf{h}_{m,\cdot,\tau}) = \psi(\mathbf{W}_g^\top \mathbf{\Lambda}_m \mathbf{h}_{m,\cdot,\tau}), \tag{4}$$

where $\mathbf{\Lambda}_m = \mathbf{D}_m^{-\frac{1}{2}}(\mathbf{D}_m - \mathbf{A}_m)\mathbf{D}_m^{-\frac{1}{2}} \in \mathbb{R}^{N_m \times N_m}$ denotes the Laplacian normalization of $\mathbf{A}_m$; $\mathbf{D}_m$ denotes the degree matrix of $\mathbf{A}_m$; $\mathbf{W}_g$ denotes the trainable parameters shared across timeline; $\psi(\cdot)$ denotes the ReLU activation function. With the above ODE, we can model the dynamics by integrating over continuous time:

$$\mathbf{h}_{m,\cdot,\tau} = \mathbf{h}_{m,\cdot,1} + \int_1^\tau \psi(\mathbf{W}_g^\top \mathbf{\Lambda}_m \mathbf{h}_{m,\cdot,t}) dt. \tag{5}$$

The hidden representations $\mathbf{h}_{m,\cdot,t}$ for all time points $t \in (1, T_m]$ could be calculated with the above integration. Then we reconstruct the states by the decoder of pre-trained PDEDER $\hat{\mathbf{x}}_{m,n} = f(f(\mathbf{h}_{m,n}; \mathbf{\Theta}_d^*); \mathbf{W}_r^s)$ and learn dynamics on system $s$ by minimizing the forecasting loss against the ground-truth observations $\mathbf{x}_{m,n}$:

$$\mathcal{L}_f(\mathbf{\Theta}^*, \mathbf{\Phi}, \mathbf{W}^f) = \sum_{m=1}^{M_s} \sum_{n=1}^{N_m} \ell(\hat{\mathbf{x}}_{m,n}, \mathbf{x}_{m,n}), \tag{6}$$

where $\mathbf{\Theta}^* = \{\mathbf{\Theta}_e^*, \mathbf{\Theta}_d^*\}$ denotes the encoder (decoder) parameters of the pre-trained PDEDER; $\mathbf{\Phi} = \{\mathbf{W}_g\}$ denotes the parameters of the neural ODE module; $\mathbf{W}^f = \{\mathbf{W}_{dp}^s, \mathbf{W}_c, \mathbf{W}_r^s\}_{s=1}^S$.

### 4.4 MODEL TRAINING.

We first pre-train PDEDER on all collected dynamics observations (without graph) with Eq.3 for $E_p$ epochs. To handle the massive observations and various numbers of samples on different systems, we randomly choose 10 dynamics systems for each training round and train PDEDER for 5 epochs with all the observations from these systems. When learning a specific dynamics, we fine-tune PDEDER with Eq.6 for $E_f$. The training details are presented in Algorithm 1 and 2.

## 5 EXPERIMENT

### 5.1 EXPERIMENTAL SETTINGS

**Datasets** In fine-tuning, we adopt 17 dynamics owning object interactions for validation, including 7 sets of synthetic observations: Mutualistic, Heat Diffusion, 2D Compressible Navier-Stokes, Darcy Flow, Gene, Shallow Water, 2D Diffusion Reaction; and 10 real-world observations: LA, SD, TDrive, CHIBike, NYCTaxi, PEMS03, PEMS04, PEMS07, PEMS08 and NOAA. Detailed descriptions are introduced in Appendix A.

**Baselines** We apply 5 baseline methods which models dynamics on interaction graph for comparison, including GNSSanchez-Gonzalez et al. (2020), NDCN Zang & Wang (2020), ST-GODE Fang et al. (2021), MT-GODE Jin et al. (2022) and TANGO Huang et al. (2024b). Following PDEDER, we adopt instance normalization on observations for all methods. Details of baseline methods are listed in Appendix B:

---

**Algorithm 1** Pre-training PDEDER to learn dynamics-enriched embeddings.

---

**Input:** 153 Observations sets $\{\mathcal{V}_m\}_{m=1}^{M_s}$ from $S$ systems
**Output:** Optimal parameters of PDEDER $\{\Theta^*, \mathbf{W}^{p*}\}$
1: Initialize $\Theta$ by pre-trained LM;
2: **for** round $r = 1$ to $MaxRound$ **do**
3:     Sample 10 systems for training;
4:     **for** epoch $e = 1$ to $MaxEpoch_p$ **do**
5:         **for** iter $it = 1$ to $MaxIter_p$ **do**
6:             Sample $B$ observations from 10 systems as a batch by ratios ;
7:             Encode and decode observations by Eq.2;
8:             Calculate the pre-training objective of Eq.3;
9:             Update $\{\Theta, \mathbf{W}^p\}$ by Eq.3;
10:        **end for**
11:    **end for**
12: **end for**

---

**Algorithm 2** Fine-tuning PDEDER to learn specific dynamics.

---

**Input:** Observations $\{\mathcal{G}_m = (\mathcal{V}_m, \mathcal{E}_m)\}_{m=1}^{M_s}$ of a dynamics system $s$
**Output:** Optimal parameters of approximated dynamics $\mathbf{W}_g$
1: Initialize $\Theta$ by $\Theta^*$;
2: **for** epoch $e = 1$ to $MaxEpoch_f$ **do**
3:     **for** iter $it = 1$ to $MaxIter_f$ **do**
4:         Encode initial values by pre-trained PDEDER;
5:         Calculate integration for each time point by Eq.5;
6:         Calculate the fine-tuning objective of Eq.6;
7:         Update $\{\Theta^*, \mathbf{W}_g, \mathbf{W}^f\}$ by Eq.6.
8:     **end for**
9: **end for**

---

**Implementation Details** We adopt the pre-trained T5 to initialize the PLM module. We apply all available 153 sets of observations for pre-training PDEDER. To handle the massive observations and different numbers of samples on different systems, we randomly sample 10 sets of observations for each training round, and sample trajectories according to the proportions of their amounts. We train 5 epochs on each sets. The learning rates are set as $1e-3$ for the PLM module and $1e-2$ for rest parameters. The patch length and stride are set as 30 and 6, respectively. To align the observations with different lengths, we split each observation by a look-back window of length 150 and stride 50. $T_k$ is set as 120.

In fine-tuning to learning a specific dynamics, the lengths of training observations is set as 60 for 2D Diffusion Reaction and 120 for the rest systems. The rest observations are left for testing. We fine-tune each observations for 20 epochs. The learning rates are tuned over $\{1e-4, 1e-5, 1e-6\}$ for the PLM module and $\{1e-2, 1e-3, 1e-4\}$ for rest parameters. The patch length and stride are set as 50 and 10, respectively. We adopt look-back window to handle overlong observations. The window length and stride are set as 840 and 50 for NYCTaxi, CHIBike, TDrive, PEMS03, PEMS04, PEMS07, PEMS08 and NOAA. Specifically, we adopt the last $L_p$ states in training observations as initial states to forecast the test observations when fine-tuning PDEDER.

**Short/Long-term Forecasting Settings** The training sequence length are same for both short and long term forecasting. For NYCTaxi, CHIBike, TDrive, PEMS03, PEMS04, PEMS07, PEMS08 and NOAA, the short- and long-term forecasting lengths are set as $\{24, 48\}$ and $\{96, 192, 336, 720\}$. For the rest dynamics, due to the diversity of convergence characteristics on each system, we truncate the test sequences by ratios to form the short/long-term forecasting sequences. The ratios for short- and long-term are set as $\{10\%, 20\%\}$ and $\{50\%, 70\%, 80\%, 100\%\}$, respectively. For example, when the test sequence length is 200, we set $10\% \times 200 = 20$ and $20\% \times 200 = 40$ as the forecasting lengths.

Table 2: Average results of dynamics forecasting. The best scores are in **boldface**. % denotes the results are scaled by 1/100.

| | Short-term Forecasting | | | | | | | | | | | | | | |
|---|---|---|---|---|---|---|---|---|---|---|---|---|---|---|---|
| System | GNS | | | NDCN | | | ST-GODE | | | MT-GODE | | | PDEDER | | |
| | MSE | MAE | MRAE | MSE | MAE | MRAE | MSE | MAE | MRAE | MSE | MAE | MRAE | MSE | MAE | MRAE |
| Mutualistic | 0.283 | 0.418 | 1.717 | 0.424 | 0.542 | 3.560 | 1.058 | 0.901 | 2.729 | 0.961 | 0.781 | 1.299 | 0.362 | 0.452 | 5.761 |
| Heat | 0.113 | 0.280 | 0.510 | 0.490 | 0.551 | 2.903 | 0.676 | 0.666 | 3.813 | 0.910 | 0.795 | 1.968 | 0.003 | 0.045 | 0.286 |
| 2D CFD | 0.302 | 0.383 | 31.315 | 0.490 | 0.482 | 8.418 | 0.566 | 0.434 | 1.518 | 0.546 | 0.432 | 12.360 | 0.223 | 0.303 | 1.185 |
| DarcyFlow | 0.233% | 4.093% | 1.077 | 0.524% | 4.960% | 21.085 | 0.195% | 3.338% | 10.209 | 0.660% | 6.214% | 23.893 | 0.067% | 2.016% | 1.398 |
| Gene | 0.045 | 0.116 | 0.757 | 0.616 | 0.644 | 1.919 | 0.841 | 0.755 | 2.924 | 0.805 | 0.661 | 1.006 | 0.035 | 0.136 | 1.513 |
| ShallowWater | 0.985 | 0.542 | 1.273 | 0.966 | 0.569 | 1.053 | 1.013 | 0.513 | 1.063 | 1.002 | 0.573 | 1.255 | 0.674 | 0.358 | 1.897 |
| 2D Diff-Reac | 1.013 | 1.059 | 20.029 | 1.157 | 0.846 | 8.604 | 1.000 | 0.853 | 1.063 | 0.967 | 0.761 | 4.373 | 0.960 | 0.723 | 4.942 |
| LA | 0.493 | 0.489 | 3.552 | 0.993 | 0.789 | 2.563 | 0.944 | 0.713 | 3.320 | 1.077 | 0.780 | 2.499 | 0.581 | 0.516 | 2.325 |
| SD | 0.418 | 0.450 | 6.053 | 1.027 | 0.742 | 3.770 | 0.309 | 0.430 | 3.682 | 1.096 | 0.780 | 2.337 | 0.634 | 0.472 | 3.943 |
| NYCTaxi | 0.361 | 0.354 | 56.987 | 0.327 | 0.398 | 44.061 | 0.540 | 0.555 | 54.988 | 1.038 | 0.770 | 24.018 | 0.181 | 0.257 | 85.898 |
| CHIBike | 0.592 | 0.237 | 33.041 | 0.719 | 0.258 | 32.404 | 0.506 | 0.258 | 21.940 | 1.039 | 0.428 | 10.450 | 0.349 | 0.174 | 17.906 |
| TDrive | 0.302 | 0.290 | 16070.4 | 0.238 | 0.274 | 10110.9 | 0.461 | 0.459 | 28766.0 | 0.827 | 0.605 | 10122.5 | 0.119 | 0.169 | 15789.4 |
| PEMS03 | 0.173 | 0.276 | 4.498 | 0.996 | 0.815 | 8.196 | 0.200 | 0.310 | 6.882 | 0.998 | 0.824 | 2.729 | 0.186 | 0.284 | 4.177 |
| PEMS04 | 0.505 | 0.420 | 7.491 | 1.030 | 0.688 | 3.931 | 0.500 | 0.509 | 5.543 | 0.983 | 0.696 | 2.522 | 0.691 | 0.505 | 4.585 |
| PEMS07 | 0.578 | 0.535 | 6.373 | 1.101 | 0.825 | 4.090 | 0.716 | 0.627 | 3.331 | 0.988 | 0.769 | 1.563 | 0.263 | 0.344 | 4.912 |
| PEMS08 | 1.010 | 0.628 | 9.361 | 0.934 | 0.688 | 2.969 | 0.849 | 0.750 | 3.639 | 1.021 | 0.717 | 2.992 | 0.643 | 0.489 | 7.809 |
| NOAA | 0.503 | 0.521 | 11.201 | 0.585 | 0.570 | 19.107 | 0.585 | 0.572 | 2.905 | 0.957 | 0.712 | 5.669 | 0.362 | 0.432 | 19.468 |
| | Long-term Forecasting | | | | | | | | | | | | | | |
| System | GNS | | | NDCN | | | ST-GODE | | | MT-GODE | | | PDEDER | | |
| | MSE | MAE | MRAE | MSE | MAE | MRAE | MSE | MAE | MRAE | MSE | MAE | MRAE | MSE | MAE | MRAE |
| Mutualistic | 0.989 | 0.774 | 5.493 | 0.913 | 0.807 | 2.590 | 1.034 | 0.885 | 1.686 | 0.999 | 0.857 | 1.082 | 0.809 | 0.675 | 2.859 |
| Heat | 0.176 | 0.336 | 1.497 | 0.516 | 0.586 | 19.415 | 0.724 | 0.700 | 15.172 | 0.973 | 0.838 | 9.610 | 0.006 | 0.052 | 3.930 |
| 2D CFD | 0.238 | 0.313 | 39.102 | 0.440 | 0.425 | 12.810 | 0.573 | 0.437 | 1.703 | 0.378 | 0.348 | 14.149 | 0.152 | 0.236 | 1.808 |
| DarcyFlow | 0.163% | 3.254% | 3.827 | 0.505% | 4.909% | 21.042 | 0.150% | 2.901% | 7.479 | 0.856% | 7.305% | 29.049 | 0.072% | 2.064% | 1.457 |
| Gene | 0.200 | 0.283 | 3.064 | 0.640 | 0.636 | 2.128 | 0.979 | 0.789 | 2.268 | 0.991 | 0.761 | 1.445 | 0.076 | 0.172 | 1.994 |
| ShallowWater | 1.002 | 0.545 | 1.316 | 0.993 | 0.578 | 1.018 | 1.012 | 0.513 | 1.046 | 1.006 | 0.579 | 1.134 | 1.145 | 0.527 | 1.960 |
| 2D Diff-Reac | 1.007 | 1.060 | 10.992 | 1.133 | 0.837 | 5.076 | 1.001 | 0.852 | 1.129 | 1.005 | 0.792 | 2.535 | 1.057 | 0.794 | 4.037 |
| LA | 0.487 | 0.484 | 3.267 | 0.995 | 0.788 | 2.696 | 0.883 | 0.707 | 3.035 | 0.963 | 0.747 | 1.731 | 0.571 | 0.510 | 2.033 |
| SD | 0.428 | 0.454 | 7.476 | 1.027 | 0.747 | 3.285 | 0.332 | 0.444 | 3.175 | 0.968 | 0.741 | 2.008 | 0.642 | 0.482 | 3.764 |
| NYCTaxi | 0.361 | 0.354 | 61.015 | 0.340 | 0.406 | 55.474 | 0.580 | 0.579 | 63.998 | 1.018 | 0.763 | 11.699 | 0.208 | 0.271 | 53.717 |
| CHIBike | 0.598 | 0.237 | 69.580 | 0.722 | 0.259 | 60.684 | 0.536 | 0.263 | 93.543 | 1.015 | 0.422 | 15.124 | 0.350 | 0.178 | 62.679 |
| TDrive | 0.367 | 0.332 | 18430.2 | 0.303 | 0.320 | 14311.6 | 0.520 | 0.468 | 24255.0 | 0.898 | 0.646 | 7457.8 | 0.161 | 0.194 | 15649.1 |
| PEMS03 | 0.303 | 0.385 | 5.766 | 1.122 | 0.872 | 7.408 | 0.205 | 0.317 | 7.102 | 1.016 | 0.840 | 2.755 | 0.302 | 0.378 | 7.305 |
| PEMS04 | 0.742 | 0.548 | 14.116 | 1.026 | 0.687 | 4.195 | 0.394 | 0.456 | 5.569 | 1.003 | 0.708 | 2.483 | 0.836 | 0.586 | 6.029 |
| PEMS07 | 0.575 | 0.533 | 7.386 | 1.090 | 0.820 | 4.594 | 0.715 | 0.627 | 3.282 | 1.000 | 0.781 | 1.761 | 0.435 | 0.464 | 6.400 |
| PEMS08 | 1.006 | 0.625 | 12.804 | 0.935 | 0.688 | 3.822 | 0.872 | 0.757 | 4.179 | 1.008 | 0.715 | 2.805 | 0.763 | 0.565 | 8.161 |
| NOAA | 0.782 | 0.655 | 14.856 | 0.855 | 0.691 | 21.577 | 0.712 | 0.636 | 3.695 | 1.007 | 0.741 | 6.280 | 0.699 | 0.607 | 22.449 |

## 5.2 MODELING DYNAMICS

**In-domain Forecasting**  We first examine PDEDER by short/long-term forecasting on in-domain settings against 4 baseline methods. We pre-train PDEDER on all 153 sets of observations and fine-tune on one set of observations for each system. We examine the performance by MSE and MAE. Due to the memory limitation, we compare with TANGO only on systems with less objects, including LA, NYCTaxi, PEMS08 and NOAA. The average results of short/long-term forecasting are presented in Table 2 and 6. The full results are presented in Appendix C. According to the forecasting results, we can find that our PDEDER outperforms baseline methods in most settings and improve the performance significantly. These observations directly indicate the effectiveness of our PDEDER which can approximate hidden dynamics elegantly in the latent space.

**Cross-domain Forecasting**  We examine the generalizability of our PDEDER on cross-domain settings. We set two Leave-One-Out (LOO) cross-domain settings, leaving one system out and leaving one set of hyper-parameters out. For LOO on system $s$, we pre-train PDEDER by observations excluding all sets of observations on system $s$ and fine-tune on one set of observations of system $s$ for validation. For LOO on hyper-parameters, we pre-train PDEDER with observations excluding observations of a specific hyper-parameter $\mathcal{G}_m$ and fine-tune on $\mathcal{G}_m$ for validation. The two versions are denoted as "PDEDER-sys" and "PDEDER-para". We compare PDEDER with the two cross-domain

Table 3: Average results of short/long-term forecasting comparing in-domain and cross-domain settings. The best scores are in **boldface**. The dataset names are abbreviated for briefness. "N/A" denotes the system contains no system-specific hyper-parameters. "%" denotes the results are scaled by 1/100.

| System | PDEDER | | | | | | PDEDER-sys | | | | | | PDEDER-para | | | | | |
|---|---|---|---|---|---|---|---|---|---|---|---|---|---|---|---|---|---|---|
| | short-term | | | long-term | | | short-term | | | long-term | | | short-term | | | long-term | | |
| | MSE | MAE | MRAE | MSE | MAE | MRAE | MSE | MAE | MRAE | MSE | MAE | MRAE | MSE | MAE | MRAE | MSE | MAE | MRAE |
| Mutualistic | 0.362 | 0.452 | 5.761 | 0.809 | 0.675 | 2.859 | 0.395 | 0.488 | 6.172 | 0.906 | 0.738 | 3.074 | 0.407 | 0.493 | 6.832 | 0.904 | 0.754 | 3.145 |
| Heat | 0.003 | 0.045 | 0.286 | 0.006 | 0.052 | 3.930 | 0.009 | 0.078 | 0.616 | 0.010 | 0.080 | 3.778 | 0.004 | 0.048 | 0.442 | 0.006 | 0.051 | 3.856 |
| 2D CFD | 0.223 | 0.303 | 1.185 | 0.152 | 0.236 | 1.808 | 0.222 | 0.297 | 1.096 | 0.151 | 0.232 | 1.676 | 0.449 | 0.412 | 1.208 | 0.384 | 0.354 | 1.984 |
| DarcyFlow | 0.001 | 0.020 | 1.398 | 0.001 | 0.021 | 1.457 | 0.001 | 0.020 | 1.389 | 0.001 | 0.021 | 1.438 | 0.001 | 0.021 | 1.413 | 0.001 | 0.021 | 1.463 |
| Gene | 0.035 | 0.136 | 1.513 | 0.076 | 0.172 | 1.994 | 0.052 | 0.184 | 1.621 | 0.071 | 0.192 | 1.962 | 0.050 | 0.180 | 1.636 | 0.086 | 0.213 | 1.728 |
| ShallowWater | 0.674 | 0.358 | 1.897 | 1.145 | 0.527 | 1.960 | 0.773 | 0.368 | 1.960 | 1.032 | 0.464 | 1.448 | N/A | N/A | N/A | N/A | N/A | N/A |
| 2D Diff-Reac | 0.960 | 0.723 | 4.942 | 1.057 | 0.794 | 4.037 | 0.972 | 0.727 | 5.268 | 1.034 | 0.786 | 3.748 | N/A | N/A | N/A | N/A | N/A | N/A |
| LA | 0.581 | 0.516 | 2.325 | 0.571 | 0.510 | 2.033 | 0.584 | 0.516 | 2.430 | 0.572 | 0.508 | 2.124 | N/A | N/A | N/A | N/A | N/A | N/A |
| SD | 0.634 | 0.472 | 3.943 | 0.642 | 0.482 | 3.764 | 0.641 | 0.479 | 3.985 | 0.648 | 0.487 | 3.741 | N/A | N/A | N/A | N/A | N/A | N/A |
| NYCTaxi | 0.181 | 0.257 | 85.898 | 0.208 | 0.271 | 53.717 | 0.192 | 0.266 | 87.668 | 0.217 | 0.278 | 54.779 | N/A | N/A | N/A | N/A | N/A | N/A |
| CHIBike | 0.349 | 0.174 | 17.906 | 0.350 | 0.178 | 62.679 | 0.308 | 0.167 | 19.382 | 0.333 | 0.175 | 62.413 | N/A | N/A | N/A | N/A | N/A | N/A |
| Tdrive | 0.119 | 0.169 | 15789.4 | 0.161 | 0.194 | 15649.1 | 0.118 | 0.171 | 16136.9 | 0.161 | 0.197 | 16000.9 | N/A | N/A | N/A | N/A | N/A | N/A |
| PEMS03 | 0.186 | 0.284 | 4.177 | 0.302 | 0.378 | 7.305 | 0.188 | 0.288 | 4.629 | 0.308 | 0.384 | 7.871 | N/A | N/A | N/A | N/A | N/A | N/A |
| PEMS04 | 0.691 | 0.505 | 4.585 | 0.836 | 0.586 | 6.029 | 0.681 | 0.497 | 4.555 | 0.832 | 0.583 | 6.104 | N/A | N/A | N/A | N/A | N/A | N/A |
| PEMS07 | 0.263 | 0.344 | 4.912 | 0.435 | 0.464 | 6.400 | 0.233 | 0.324 | 4.664 | 0.431 | 0.464 | 6.715 | N/A | N/A | N/A | N/A | N/A | N/A |
| PEMS08 | 0.643 | 0.489 | 7.809 | 0.763 | 0.565 | 8.161 | 0.639 | 0.490 | 8.181 | 0.762 | 0.566 | 8.305 | N/A | N/A | N/A | N/A | N/A | N/A |
| NOAA | 0.362 | 0.432 | 19.468 | 0.699 | 0.607 | 22.449 | 0.373 | 0.440 | 20.447 | 0.707 | 0.607 | 22.387 | 0.364 | 0.430 | 20.960 | 0.832 | 0.639 | 23.498 |

versions and present the averaged results of short/long-term forecasting in Table 3. Detailed results are presented in Appendix C. We can find that, the performance of in-domain setting outperforms the cross-domain settings in most cases. Meanwhile, the performance of excluding one system also beat the in-domain setting in some cases and the overall performance gaps are not too large. These phenomena indicate the strong generalizability of our PDEDER, even pre-training under cross-domain settings, our PDEDER can generate performance on a par with in-domain settings.

**Impact Evaluation of Pre-training on downstream Dynamics Modeling**  We examine the impact of pre-training on downstream dynamics modeling by modifying the initialization of the encoder and decoder when fine-tuning. We set two comparable versions, 1) initializing PDEDER by pre-trained LM, denoted as "PDEDER w/o pre"; 2) initializing PDEDER by pre-trained

We conduct ablative study to examine the effectiveness of pre-training on PDEDER. We set two ablative versions: 1) fine-tuning PDEDER without pre-training, denoted as "PDEDER w/o pre"; 2) fine-tuning PDEDER with freezing the pre-trained encoder and decoder, denoted as "PDEDER freeze $\Theta^*$". The averaged results are presented in Table 4 and full results are presented in Appendix C. We can find that the full version with pre-training PDEDER consistently outperforms the ablative versions, indicating the effectiveness of our pre-training mechanism on massive dynamics observations when learning hidden dynamics in latent space. Besides, we surprisingly find that the version freezing the pre-trained encoder and decoder outperforms the version without pre-training in most settings. This phenomena indicate that the pre-training processes can effectively capture the dynamics properties, leading to less efforts on fine-tuning processes when learning specific dynamics. According to these results, our PDEDER can be directly adopted as an effective embedder on learning specific dynamics in real-world applications when fine-tuning are unavailable.

**Forecasting Visualization**  We present forecasting visualizations on Heat and Mutualistic on variants of PDEDER and the results are presented in Fig. 2 3 of Appendix. We may find that our PDEDER performs comparable dynamics behaviors against the ground-truth values.

**Sensitivity Analysis**  WE conducted sensitivity analysis on the patch length and stride of fine-tuning period on Mutualistic, Heat, DarcyFlow, Gene and 2D Diffusion-Reaction. The results are presented in Fig. 4. We may find that the fine-tuning process are quite insensitive to these two parameters, leading to robustness in practical usages.

**Prediction of Incidence Proportion**  We evaluate the performance of prediction on incidence proportions by MSE and MAE. Incidence Proportion (IP) Noordzij et al. (2010) measures the probability of a special event (*e.g.,* the infection of epidemic diseases) in a certain period. IP $= \frac{D_e}{D_o}$, where $D_e$ denotes the number of occurrence of a certain event; $D_o$ denotes the total number of monitored

Table 4: Average forecasting results of MSE and MAE under on Impact Evaluation of Pre-training on downstream Dynamics Modeling. The best scores are in **boldface**. The dataset names are abbreviated for briefness. "%" denotes the results are scaled by 1/100.

| System | PDEDER | | | | | | PDEDER w/o pre | | | | | | PDEDER freeze $\Theta^*$ | | | | | |
|---|---|---|---|---|---|---|---|---|---|---|---|---|---|---|---|---|---|---|
| | short-term | | | long-term | | | short-term | | | long-term | | | short-term | | | long-term | | |
| | MSE | MAE | MARE | MSE | MAE | MARE | MSE | MAE | MARE | MSE | MAE | MARE | MSE | MAE | MARE | MSE | MAE | MARE |
| Mutualistic | 0.362 | 0.452 | 5.761 | 0.809 | 0.675 | 2.859 | 0.531 | 0.592 | 6.435 | 1.003 | 0.812 | 3.233 | 0.313 | 0.430 | 6.012 | 0.818 | 0.690 | 3.014 |
| Heat | 0.003 | 0.045 | 0.286 | 0.006 | 0.052 | 3.930 | 0.027 | 0.136 | 0.868 | 0.033 | 0.138 | 10.895 | 0.008 | 0.068 | 0.724 | 0.010 | 0.072 | 3.903 |
| 2D CFD | 0.223 | 0.303 | 1.185 | 0.152 | 0.236 | 1.808 | 0.224 | 0.302 | 1.195 | 0.151 | 0.234 | 1.988 | 0.225 | 0.311 | 1.223 | 0.155 | 0.244 | 1.964 |
| DarcyFlow | 0.067% | 2.016% | 1.398 | 0.072% | 2.064% | 1.457 | 0.067% | 2.023% | 1.389 | 0.073% | 2.086% | 1.438 | 0.075% | 2.108% | 1.413 | 0.077% | 2.141% | 1.463 |
| Gene | 0.035 | 0.136 | 1.513 | 0.076 | 0.172 | 1.994 | 0.140 | 0.311 | 1.747 | 0.163 | 0.321 | 1.912 | 0.113 | 0.236 | 2.376 | 0.327 | 0.428 | 4.038 |
| ShallowWater | 0.674 | 0.358 | 1.897 | 1.145 | 0.527 | 1.960 | 1.000 | 0.416 | 1.091 | 1.057 | 0.437 | 1.122 | 0.754 | 0.371 | 1.244 | 1.034 | 0.478 | 1.456 |
| 2D Diff-Reac | 0.960 | 0.723 | 4.942 | 1.057 | 0.794 | 4.037 | 0.981 | 0.737 | 5.265 | 1.006 | 0.777 | 3.523 | 0.958 | 0.730 | 4.634 | 1.009 | 0.778 | 3.372 |
| LA | 0.581 | 0.516 | 2.325 | 0.571 | 0.510 | 2.033 | 0.590 | 0.519 | 2.721 | 0.577 | 0.512 | 2.367 | 0.580 | 0.514 | 2.315 | 0.570 | 0.508 | 2.041 |
| SD | 0.634 | 0.472 | 3.943 | 0.642 | 0.482 | 3.764 | 0.666 | 0.500 | 4.093 | 0.675 | 0.509 | 3.733 | 0.642 | 0.480 | 4.007 | 0.652 | 0.490 | 3.805 |
| NYCTaxi | 0.181 | 0.257 | 85.898 | 0.208 | 0.271 | 53.717 | 0.188 | 0.265 | 87.134 | 0.219 | 0.284 | 55.841 | 0.212 | 0.297 | 64.056 | 0.224 | 0.286 | 47.551 |
| CHIBike | 0.349 | 0.174 | 17.906 | 0.350 | 0.178 | 62.679 | 0.510 | 0.212 | 23.712 | 0.414 | 0.191 | 76.088 | 0.417 | 0.194 | 18.753 | 0.388 | 0.185 | 61.128 |
| Tdrive | 0.119 | 0.169 | 15789.4 | 0.161 | 0.194 | 15649.1 | 0.165 | 0.214 | 19681.8 | 0.186 | 0.227 | 18089.6 | 0.152 | 0.194 | 15058.4 | 0.175 | 0.211 | 15216.5 |
| PEMS03 | 0.186 | 0.284 | 4.177 | 0.302 | 0.378 | 7.305 | 0.202 | 0.311 | 4.729 | 0.333 | 0.408 | 8.353 | 0.186 | 0.295 | 4.488 | 0.302 | 0.382 | 7.392 |
| PEMS04 | 0.691 | 0.505 | 4.585 | 0.836 | 0.586 | 6.029 | 0.703 | 0.520 | 5.121 | 0.875 | 0.609 | 6.799 | 0.678 | 0.500 | 4.574 | 0.835 | 0.587 | 6.115 |
| PEMS07 | 0.263 | 0.344 | 4.912 | 0.435 | 0.464 | 6.400 | 0.302 | 0.386 | 5.670 | 0.494 | 0.509 | 7.513 | 0.238 | 0.326 | 5.077 | 0.426 | 0.461 | 6.547 |
| PEMS08 | 0.643 | 0.489 | 7.809 | 0.763 | 0.565 | 8.161 | 0.670 | 0.525 | 9.137 | 0.824 | 0.607 | 9.616 | 0.641 | 0.489 | 7.956 | 0.759 | 0.564 | 8.247 |
| NOAA | 0.362 | 0.432 | 19.468 | 0.699 | 0.607 | 22.449 | 0.358 | 0.431 | 16.131 | 0.660 | 0.592 | 18.551 | 0.402 | 0.456 | 20.144 | 0.720 | 0.615 | 20.964 |

Table 5: Average results of predicted Incidence Proportion. The best scores are in **boldface**.

| Method | PEMS03 | | PEMS04 | | PEMS07 | | PEMS08 | |
|---|---|---|---|---|---|---|---|---|
| | short | long | short | long | short | long | short | long |
| NDCN | 0.012 | 0.010 | 0.020 | 0.020 | 0.026 | 0.024 | 0.017 | 0.016 |
| MTGODE | 0.043 | 0.035 | 0.248 | 0.230 | 0.893 | 0.774 | 1.066 | 0.868 |
| TANGO | OOM | OOM | OOM | OOM | OOM | OOM | 5.210 | 6.646 |
| PDEDER | **0.005** | **0.005** | **0.017** | **0.017** | **0.008** | **0.009** | **0.012** | **0.010** |

subjects in the specified period. Following this, on traffic datasets, we calculate IP of traffice flows for each time point. We assign the object state at each time point to $D_e$ and assign the states summation of all objects at one time point to $D_o$. We measure the predicted IP by MAE and the results are illustrated in Table 5. We can find that our PDEDER significantly outperforms baseline methods in all settings. This indicate that our PDEDER can serve as an effective forecaster for instance monitoring and warning in real-world applications.

## 6 CONCLUSION

In this paper, we propose a generalized pre-trained dynamics encoder PDEDER to learn generalizable embeddings for learning specific dynamics. During pre-training, we first collect 153 sets of dynamics observations from both synthetic and real-world systems. Then we pre-train a PLM-based PDEDER with all available observations by reconstructing and forecasting tasks to learn dynamics-enriched embeddings for each of the observations. Specifically, we introduce a data projection module for aligning states dimensions from different systems before the encoder. We also present the usage of fine-tuning PDEDER to learn specific dynamics. We encode the initial states by the pre-trained PDEDER and learn dynamics in latent space by a GNN-based ODE learner. We conducted empirical studies on short/long-term forecasting under in-domain and cross-domain settings. The results indicate the effectiveness and generality of our PDEDER. Specially, when freezing the pre-trained encoder (decoder) in fine-tuning, our PDEDER can also generate excellent performance, further indicating that PDEDER can serve as an effective embedder when fine-tuning are unavailabel.

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

GENERALIZING DYNAMICS MODELING EASIER FROM REPRESENTATION PERSPECTIVE: APPENDIX

## A  DYNAMICS

We introduce the dynamics we adopted for generating observations and the descriptions on synthetic systems.

**Charged**  Similar to the Springs dataset, the Charged dataset Kipf et al. (2018) simulates the motion of charges in a two-dimensional bounded space. Five charges interact with each other through Coulomb forces, with the magnitude of the force being influenced by the distance between the charges. The expression for the Coulomb force is as follows, where $C$ is a constant.

$$\mathbf{F}_{ij} = C \cdot \text{sign}(\mathbf{q}_i \cdot \mathbf{q}_j) \frac{\mathbf{r}_i - \mathbf{r}_j}{\|\mathbf{r}_i - \mathbf{r}_j\|^3}. \tag{7}$$

**Springs**  Five particles move in a two-dimensional bounded space without external forces Kipf et al. (2018). The probability of randomly connecting each pair of particles with a spring is 0.5. Particles connected by springs will be influenced by Hooke's law $\mathbf{F}_{m,n} = -k(\mathbf{r}_m - \mathbf{r}_n)$, where $\mathbf{F}_{m,n}$ is the force exerted by particle $v_m$ on particle $v_n$, and $\mathbf{r}_m$ is the position vector of particle $v_m$. The initial position of each particle is sampled from $N(0, 0.5)$, and the initial velocity is a random vector with a norm of 0.5.

**Mutualistic Interaction Dynamics**  In the field of ecology, species interact with each other, and the expression Gao et al. (2016) is given as below :

$$\frac{d\mathbf{x_i}(\mathbf{t})}{dt} = b_i + \mathbf{x_i}\left(1 - \frac{\mathbf{x}_i}{k_i}\right)\left(\frac{\mathbf{x}_i}{c_i} - 1\right) + \sum_{j=1}^{n} \mathbf{A}_{i,j} \frac{\mathbf{x}_i \mathbf{x}_j}{d_i + e_i \mathbf{x}_i + h_j \mathbf{x}_j}. \tag{8}$$

We denote $\mathbf{x_i}$ represents the abundance of species $i$, $b_i$ denotes the immigration term, $k_i$ represents the logarithmic growth of population capacity, $c_i$ indicates the Allee effect with a cold-start threshold, and $\mathbf{A}$ is the interaction network with interaction terms.

**Heat Diffusion**  The heat diffusion dynamics is governed by Newton's law of cooling v Luikov (2012). The expression is given as below:

$$\frac{d\mathbf{x}_n^t}{dt} = -k_{nn'} \sum_{n' \in \mathcal{N}(n)} \mathbf{A}_{nn'}(\mathbf{x}_n - \mathbf{x}_{n'}). \tag{9}$$

Let $\mathbf{A}$ denotes the heat capacity matrix, for object $n$, the corresponding heat change is proportional to the temperature differences between object $n$ and its corresponding neighbor objects $n' \in \mathcal{N}(n)$.

**1D Diffusion-Reaction**  In the one-dimensional diffusion-reaction equation Takamoto et al. (2022),with the specific expression as follows:

$$\partial_t u(t, x) - \nu \partial_{xx} u(t, x) - \rho u(1 - u) = 0, \quad x \in (0, 1), \ t \in (0, 1],$$
$$u(0, x) = u_0(x), \quad x \in (0, 1). \tag{10}$$

In the equation, the variable $u$ is used to represent the ability to capture fast dynamics, where periodicity and initial conditions known to the advection equation are used

**Compressible Navier-Stokes**  The flow of compressible fluids is generally represented by compressible fluid dynamics equations Klaasen & Troy (1984), which are expressed as follows:

$$\partial_t \rho + \nabla \cdot (\rho \mathbf{v}) = 0,$$
$$\rho(\partial_t \mathbf{v} + \mathbf{v} \cdot \nabla \mathbf{v}) = -\nabla p + \eta \Delta \mathbf{v} + \left(\zeta + \frac{\eta}{3}\right) \nabla(\nabla \cdot \mathbf{v}),$$
$$\partial_t \left[\epsilon + \frac{\rho v^2}{2}\right] + \nabla \cdot \left[\left(\epsilon + p + \frac{\rho v^2}{2}\right) \mathbf{v} - \mathbf{v} \cdot \boldsymbol{\sigma}\right] = 0, \tag{11}$$

where $\rho$ represents mass density, $v$ denotes velocity, $p$ is the pressure, $\epsilon$ signifies internal energy, $\sigma$ represents the viscous tensor, $\eta$ is the shear viscosity, $\zeta$ is the bulk viscosity, and $M = \frac{|\mathbf{v}|}{c_s}$ denotes the Mach number. Following Takamoto et al. (2022), we abbreviate this dynamics as "CFD" for briefness.

**Burgers** The Burgers' equation is used to describe nonlinear behavior and diffusion processes in fluid dynamics Takamoto et al. (2022). Define the specific equation as follows:

$$\partial_t u(t, x) + \partial_x \left( \frac{u^2(t, x)}{2} \right) = \frac{\nu}{\pi} \partial_{xx} u(t, x), \quad x \in (0, 1), \ t \in (0, 2],$$
$$u(0, x) = u_0(x), \quad x \in (0, 1). \tag{12}$$

Here, $v$ is the diffusion coefficient, which is a constant, and the same initial conditions as those used for advection are applied.

**Advection** The advection equation is used to simulate nonlinear advection behavior Takamoto et al. (2022). The specific equation is as follows:

$$\partial_t u(t, x) + \beta \partial_x u(t, x) = 0, \quad x \in (0, 1), \ t \in (0, 2],$$
$$u(0, x) = u_0(x), \quad x \in (0, 1). \tag{13}$$

Set $\beta$ as the advection velocity, using the sine wave's hyperposition as the initial condition $u_0(x)$.

**Gene Regulatory** The dynamics of gene regulation are governed by the Michaelis-Menten equation, as specifically shown below Gao et al. (2016).

$$\frac{d\mathbf{x_n(t)}}{dt} = -b_n \mathbf{x}_n^f + \sum_{n'=1}^{m} \mathbf{A}_{n,n'} \frac{\mathbf{x_{n'}}^h}{\mathbf{x}_{n'}^h + 1}. \tag{14}$$

In the first term, $f$ takes the values of 1 or 2, representing degradation and dimerization, respectively. The second term reflects genetic activation regulated by the Hill coefficient $h$ Alon (2006).

**Shallow-Water** The architectures chosen to simulate free surface flow problems are mostly derived from the Navier-Stokes equations. The specific equation is shown below:

$$\partial_t h + \partial_x(hu) + \partial_y(hv) = 0,$$
$$\partial_t(hu) + \partial_x \left( u^2 h + \frac{1}{2} g_r h^2 \right) + \partial_y(uvh) = -g_r h \partial_x b, \tag{15}$$
$$\partial_t(hv) + \partial_y \left( v^2 h + \frac{1}{2} g_r h^2 \right) + \partial_x(uvh) = -g_r h \partial_y b,$$

where $u$ and $v$ represent the horizontal and vertical velocities, respectively, $h$ denotes the water depth, $b$ indicates spatial variations, and $g_r$ is the gravitational acceleration. The terms $h_u$ and $h_v$ represent the components of momentum in the horizontal and vertical directions Takamoto et al. (2022).

**2D Diffusion-Reaction** The diffusion-reaction equation in two-dimensional space is defined as follows Takamoto et al. (2022):

$$\partial_t u = D_u \partial_{xx} u + D_u \partial_{yy} u + R_u,$$
$$\partial_t v = D_v \partial_{xx} v + D_v \partial_{yy} v + R_v, \tag{16}$$

where the activator is represented by $u$ and the inhibitor by $v$. $D_u$ and $D_v$ are the diffusion coefficients for both, while $R_u$ and $R_v$ are the reaction functions Klaasen & Troy (1984), which are defined as follows:

$$R_u(u, v) = u - u^3 - k - v,$$
$$R_v(u, v) = u - v. \tag{17}$$

We set the constants $k = 5 \times 10^{-3}, D_u = 1 \times 10^{-3}, D_v = 5 \times 10^{-3}$.

**Diffusion-Sorption**   The diffusion process delayed by adsorption is typically described by the diffusion-adsorption equation. The equation is defined as follows:

$$\partial_t u(t,x) = \frac{D}{R(u)} \partial_{xx} u(t,x), \quad x \in (0,1), \ t \in (0,500],$$

$$R(u) = 1 + \frac{1-\varphi}{\varphi} \rho_s k n_f u^{n_f - 1}, \tag{18}$$

where $D$ represents the diffusion coefficient, while $R$ denotes the delay factor of adsorption that hinders the diffusion process, with the value of $R$ being dependent on $u$. The parameter $\varpi$ represents the porosity of the porous medium, $\rho_s$ represents the packing density, $k$ is the Freundlich parameter, and $n$ is the Freundlich index Limousin et al. (2007).

**LA and SD**   The dataset collected hourly climate observation data for six consecutive days in Los Angeles and San Diego (from June 28, 2012, at 21:00 to July 14, 2012, at 22:00), with each node containing 10 observation values Choi et al. (2023). The interaction graph structures are provided by the original datasets.

**NYCTaxi ,TDrive and CHIBike**   The NYCTaxi dataset Liu et al. (2021) contains bicycle trajectory data for 182 days in New York City, along with 4,392 traffic flow images.The TDrive dataset Liu et al. (2021) contains a large number of GPS trajectories from taxis in Beijing, along with 22,459 traffic flow images. The CHIBike dataset is sourced fromWang et al. (2021). The interaction graph structures are calculated by the geometry coordinate or grid distances between each station.

**PEMS**   The PeMS dataset is real-world traffic data collected by the California Department of Transportation Chen et al. (2001) and contains three traffic measurements, separated at five-minute intervals. The interaction graph structures are provided by the original datasets.

**NOAA**   Following Hwang et al. (2021), we randomly select 22 areas on the map of America and collect hourly temperatures from Online Climate Data Directory of the National Oveanic and Atmospheric Administration (NOAA)[1]. The interaction graph structures are calculated by the geometry coordinate distances between each station.

# B   BASELINE METHODS

Details of baseline methods are listed below:

- GNS[2] Sanchez-Gonzalez et al. (2020) is a discrete GNN-based dynamics modeling method. We modify the graph learning module into static graph structures.

- NDCN[3] Zang & Wang (2020) combines the ODEs with GNNs and approximate the integration differential equation systems by the GNN module. In testing, we set the state of the last time point of training observations as the initial states to forecast the test observations. We set the state of the last time point of training observations as the initial states for testing.

- STGODE[4] Fang et al. (2021) incorporates the geometry spatial information into continuous dynamics learning. STGODE constructs two types of graphs, including spatial and semantic correlations to capture the spatial temporal semantics by a continuous GNN with residual connections.

- MT-GODE[5] Jin et al. (2022) solves the multivariate time series forecasting by mapping the interacting observations into dynamic-graph and solve by learning continuous spatial-temporal dynamics in latent space. We adopt the single-step forecasting setting.

---

[1]https://www.ncdc.noaa.gov/cdo-web/
[2]https://github.com/zhouxian/GNS-PyTorch
[3]https://github.com/calvin-zcx/ndcn
[4]https://github.com/square-coder/STGODE
[5]https://github.com/TrustAGI-Lab/MTGODE

Table 6: Average results of short/long-term forecasting comparing with TANGO. Due to the memory limitation, we compare with TANGO only on systems with less objects. The best scores are in **boldface**.

| System | TANGO | | | | PDEDER | | | |
| | short-term | | long-term | | short-term | | long-term | |
| | MSE | MAE | MSE | MAE | MSE | MAE | MSE | MAE |
|---|---|---|---|---|---|---|---|---|
| LA | 0.761 | 0.685 | 0.699 | 0.649 | 0.574 | 0.509 | **0.561** | **0.502** |
| NYCTaxi | 1.051 | 0.793 | 2.459 | 1.207 | **0.184** | **0.261** | **0.208** | **0.273** |
| CHIBike | 0.453 | 0.391 | 1.231 | 0.621 | **0.339** | **0.174** | **0.345** | **0.178** |
| PEMS08 | 1.034 | 0.810 | 2.878 | 1.233 | 0.639 | 0.489 | **0.765** | **0.568** |
| NOAA | 1.064 | 0.809 | 1.965 | 1.031 | **0.351** | **0.422** | **0.687** | **0.597** |

Table 7: Average results of short/long-term forecasting comparing fine-tuning PDEDER by GNN-based module and SINGy. The best scores are in **boldface**.

| System | PDEDER | | | | PDEDER +SINDy | | | |
| | short-term | | long-term | | short-term | | long-term | |
| | MSE | MAE | MSE | MAE | MSE | MAE | MSE | MAE |
|---|---|---|---|---|---|---|---|---|
| Mutualistic | 0.362 | 0.452 | 0.809 | 0.675 | 1.014 | 1.014 | 0.334 | 0.334 |
| Heat | 0.003 | 0.045 | 0.006 | 0.052 | 0.886 | 0.884 | 1.577 | 1.586 |
| 2D CFD | 0.223 | 0.303 | 0.152 | 0.236 | 1.001 | 0.984 | 1.139 | 1.164 |
| DarcyFlow | 0.001 | 0.020 | 0.001 | 0.021 | 0.858 | 0.851 | 1.103 | 1.104 |
| Gene | 0.035 | 0.136 | 0.076 | 0.172 | 0.613 | 0.537 | 0.783 | 0.783 |
| Shallow Water | 0.674 | 0.358 | 1.145 | 0.527 | 0.538 | 0.463 | 1.040 | 1.047 |
| 2D DiffReac | 0.960 | 0.723 | 1.057 | 0.794 | 0.126 | 0.126 | 0.807 | 0.808 |

- TANGO[6] Huang et al. (2024b) introduce time-reversal symmetry into GNN-based ODE learner and models the observations and reversed observations simultaneously. In the period of model training, we set the observations of the first 60 lengths as observed states to forecast the latter 60 observations. In model testing, we set the last 40 lengths of training observations as observed initial states to forecast the test observations. Specially, due to the memory limitation, we only compare with TANGO on systems with less objects.

## C  FULL RESULTS OF SHORT/LONG-TERM FORECASTING

The full results for forecasting are presented in Tables 8,9, 10, 11, including the variants results and cross-domain results. The variant without pre-training of PDEDER is denoted as "PDEDER-nopre". The variant freezes the encoder/decoder of PDEDER is denoted as "PDEDER-frz". The variant pre-trains PDEDER excluding one system on cross-domain setting is denoted as "PDEDER-sys"

---

[6]https://github.com/wanjiaZhao1203/TREAT

Table 8: Full results of short/long-term forecasting comparing with baselines (1/2). The best scores are in **boldface**. The dataset names are abbreviated for briefness. "%" denotes the results are scaled by 1/100.

| System | | NDCN MSE | MAE | MRAE | GNS MSE | MAE | MRAE | ST-GODE MSE | MAE | MRAE | MT-GODE MSE | MAE | MRAE | PDEDER MSE | MAE | MRAE |
|---|---|---|---|---|---|---|---|---|---|---|---|---|---|---|---|---|
| Mutualistic | 10% | 0.166 | 0.338 | 1.031 | 0.328 | 0.475 | 2.840 | 1.064 | 0.909 | 2.875 | 0.947 | 0.792 | 1.297 | 0.276 | 0.386 | 5.702 |
| | 20% | 0.399 | 0.498 | 2.402 | 0.520 | 0.609 | 4.281 | 1.051 | 0.894 | 2.584 | 0.976 | 0.769 | 1.302 | 0.448 | 0.518 | 5.820 |
| | 50% | 0.886 | 0.733 | 5.202 | 0.855 | 0.770 | 3.221 | 1.040 | 0.878 | 1.985 | 0.998 | 0.833 | 1.118 | 0.750 | 0.656 | 3.675 |
| | 70% | 0.984 | 0.773 | 5.361 | 0.912 | 0.806 | 2.599 | 1.035 | 0.884 | 1.690 | 0.997 | 0.856 | 1.082 | 0.813 | 0.679 | 2.879 |
| | 80% | 1.017 | 0.785 | 5.391 | 0.930 | 0.817 | 2.405 | 1.033 | 0.887 | 1.598 | 0.998 | 0.864 | 1.071 | 0.829 | 0.683 | 2.625 |
| | 100% | 1.069 | 0.806 | 6.017 | 0.956 | 0.833 | 2.134 | 1.029 | 0.890 | 1.470 | 1.002 | 0.877 | 1.058 | 0.844 | 0.683 | 2.259 |
| Heat | 10% | 0.112 | 0.284 | 0.542 | 0.483 | 0.545 | 2.613 | 0.668 | 0.661 | 3.114 | 0.877 | 0.774 | 1.847 | 0.003 | 0.045 | 0.307 |
| | 20% | 0.114 | 0.276 | 0.478 | 0.498 | 0.558 | 3.192 | 0.683 | 0.672 | 4.511 | 0.943 | 0.817 | 2.088 | 0.003 | 0.045 | 0.264 |
| | 50% | 0.135 | 0.290 | 0.920 | 0.512 | 0.579 | 9.268 | 0.710 | 0.690 | 8.326 | 0.967 | 0.834 | 8.384 | 0.004 | 0.048 | 0.885 |
| | 70% | 0.161 | 0.321 | 1.418 | 0.518 | 0.587 | 19.715 | 0.727 | 0.702 | 15.480 | 0.978 | 0.841 | 9.091 | 0.005 | 0.051 | 3.719 |
| | 80% | 0.180 | 0.343 | 1.644 | 0.519 | 0.589 | 22.421 | 0.730 | 0.704 | 17.535 | 0.980 | 0.842 | 9.893 | 0.006 | 0.052 | 4.536 |
| | 100% | 0.227 | 0.392 | 2.008 | 0.516 | 0.589 | 26.257 | 0.729 | 0.705 | 19.346 | 0.968 | 0.836 | 11.070 | 0.009 | 0.059 | 6.581 |
| 2D CFD | 10% | 0.308 | 0.390 | 28.476 | 0.486 | 0.483 | 8.452 | 0.575 | 0.439 | 1.544 | 0.611 | 0.457 | 12.435 | 0.230 | 0.309 | 1.158 |
| | 20% | 0.295 | 0.376 | 34.153 | 0.494 | 0.480 | 8.384 | 0.557 | 0.429 | 1.492 | 0.480 | 0.406 | 12.285 | 0.216 | 0.296 | 1.212 |
| | 50% | 0.260 | 0.339 | 39.077 | 0.465 | 0.449 | 11.434 | 0.553 | 0.425 | 1.671 | 0.403 | 0.368 | 13.364 | 0.177 | 0.261 | 1.539 |
| | 70% | 0.241 | 0.317 | 37.658 | 0.444 | 0.429 | 12.306 | 0.564 | 0.432 | 1.697 | 0.377 | 0.348 | 12.581 | 0.155 | 0.240 | 1.715 |
| | 80% | 0.232 | 0.307 | 38.380 | 0.434 | 0.419 | 12.745 | 0.579 | 0.441 | 1.727 | 0.370 | 0.341 | 14.436 | 0.146 | 0.230 | 1.861 |
| | 100% | 0.218 | 0.288 | 41.295 | 0.415 | 0.401 | 14.753 | 0.598 | 0.452 | 1.717 | 0.363 | 0.335 | 16.217 | 0.130 | 0.212 | 2.118 |
| DarcyFlow | 10% | 0.241% | 4.178% | 1.069 | 0.532% | 4.981% | 21.489 | 0.192% | 3.404% | 10.887 | 0.522% | 5.580% | 21.086 | 0.065% | 1.998% | 1.404 |
| | 20% | 0.226% | 4.008% | 1.084 | 0.516% | 4.939% | 20.680 | 0.197% | 3.272% | 9.532 | 0.797% | 6.848% | 26.700 | 0.069% | 2.034% | 1.392 |
| | 50% | 0.186% | 3.545% | 1.169 | 0.507% | 4.914% | 21.351 | 0.162% | 2.992% | 7.970 | 0.845% | 7.193% | 28.617 | 0.072% | 2.063% | 1.486 |
| | 70% | 0.166% | 3.288% | 1.308 | 0.505% | 4.909% | 20.978 | 0.148% | 2.888% | 7.462 | 0.857% | 7.320% | 28.951 | 0.072% | 2.065% | 1.468 |
| | 80% | 0.157% | 3.180% | 1.452 | 0.505% | 4.908% | 21.000 | 0.148% | 2.887% | 7.384 | 0.897% | 7.497% | 29.882 | 0.072% | 2.065% | 1.449 |
| | 100% | 0.144% | 3.004% | 11.379 | 0.504% | 4.906% | 20.839 | 0.142% | 2.836% | 7.100 | 0.825% | 7.211% | 28.746 | 0.072% | 2.065% | 1.426 |
| Gene | 10% | 0.038 | 0.100 | 0.645 | 0.596 | 0.633 | 1.854 | 0.841 | 0.764 | 2.838 | 0.752 | 0.618 | 0.974 | 0.033 | 0.135 | 1.528 |
| | 20% | 0.053 | 0.132 | 0.870 | 0.636 | 0.654 | 1.984 | 0.841 | 0.746 | 3.010 | 0.858 | 0.705 | 1.038 | 0.036 | 0.138 | 1.499 |
| | 50% | 0.121 | 0.220 | 2.048 | 0.648 | 0.650 | 2.199 | 0.901 | 0.762 | 2.332 | 0.977 | 0.760 | 1.567 | 0.052 | 0.152 | 1.796 |
| | 70% | 0.182 | 0.273 | 2.897 | 0.643 | 0.639 | 2.136 | 0.961 | 0.782 | 2.207 | 0.990 | 0.763 | 1.451 | 0.069 | 0.167 | 1.928 |
| | 80% | 0.214 | 0.297 | 3.346 | 0.639 | 0.633 | 2.067 | 0.994 | 0.793 | 2.252 | 0.998 | 0.763 | 1.406 | 0.079 | 0.176 | 1.985 |
| | 100% | 0.283 | 0.342 | 3.966 | 0.631 | 0.622 | 2.111 | 1.062 | 0.818 | 2.282 | 0.997 | 0.758 | 1.357 | 0.103 | 0.194 | 2.268 |
| ShallowWater | 10% | 0.982 | 0.541 | 0.804 | 0.955 | 0.565 | 1.057 | 1.012 | 0.515 | 0.893 | 0.998 | 0.571 | 1.306 | 0.582 | 0.327 | 1.611 |
| | 20% | 0.988 | 0.542 | 1.741 | 0.978 | 0.572 | 1.049 | 1.014 | 0.510 | 1.232 | 1.005 | 0.576 | 1.204 | 0.766 | 0.389 | 2.183 |
| | 50% | 0.997 | 0.544 | 1.303 | 0.991 | 0.577 | 1.023 | 1.012 | 0.514 | 1.021 | 1.006 | 0.580 | 1.132 | 1.085 | 0.500 | 1.930 |
| | 70% | 1.001 | 0.545 | 1.334 | 0.993 | 0.578 | 1.019 | 1.012 | 0.514 | 1.033 | 1.006 | 0.578 | 1.125 | 1.152 | 0.529 | 1.938 |
| | 80% | 1.003 | 0.545 | 1.273 | 0.994 | 0.578 | 1.017 | 1.013 | 0.513 | 1.008 | 1.006 | 0.579 | 1.120 | 1.165 | 0.536 | 1.866 |
| | 100% | 1.006 | 0.545 | 1.355 | 0.995 | 0.579 | 1.015 | 1.012 | 0.513 | 1.122 | 1.006 | 0.579 | 1.158 | 1.179 | 0.543 | 2.106 |
| 2D DiffReac | 10% | 1.013 | 1.061 | 24.661 | 1.168 | 0.850 | 10.476 | 1.000 | 0.854 | 1.046 | 0.949 | 0.747 | 5.338 | 0.945 | 0.715 | 4.918 |
| | 20% | 1.012 | 1.057 | 15.398 | 1.146 | 0.841 | 6.732 | 1.001 | 0.853 | 1.080 | 0.985 | 0.775 | 3.408 | 0.975 | 0.731 | 4.965 |
| | 50% | 1.006 | 1.060 | 11.152 | 1.135 | 0.837 | 5.386 | 1.001 | 0.852 | 1.108 | 1.000 | 0.789 | 2.744 | 1.061 | 0.789 | 4.292 |
| | 70% | 1.007 | 1.060 | 11.343 | 1.133 | 0.837 | 5.156 | 1.001 | 0.852 | 1.146 | 1.006 | 0.792 | 2.576 | 1.053 | 0.792 | 4.123 |
| | 80% | 1.007 | 1.060 | 11.107 | 1.132 | 0.836 | 4.989 | 1.001 | 0.852 | 1.135 | 1.007 | 0.793 | 2.433 | 1.055 | 0.795 | 3.984 |
| | 100% | 1.006 | 1.061 | 10.366 | 1.131 | 0.836 | 4.775 | 1.001 | 0.852 | 1.128 | 1.008 | 0.794 | 2.387 | 1.061 | 0.800 | 3.751 |
| LA | 10% | 0.486 | 0.485 | 3.696 | 0.992 | 0.789 | 2.552 | 0.936 | 0.701 | 3.387 | 1.126 | 0.794 | 2.873 | 0.581 | 0.516 | 2.405 |
| | 20% | 0.500 | 0.494 | 3.408 | 0.994 | 0.789 | 2.574 | 0.953 | 0.725 | 3.252 | 1.027 | 0.766 | 2.126 | 0.582 | 0.516 | 2.245 |
| | 50% | 0.491 | 0.488 | 3.405 | 0.995 | 0.789 | 2.625 | 0.910 | 0.716 | 3.133 | 0.971 | 0.750 | 1.780 | 0.575 | 0.512 | 2.087 |
| | 70% | 0.486 | 0.484 | 3.338 | 0.995 | 0.788 | 2.666 | 0.883 | 0.706 | 3.092 | 0.963 | 0.747 | 1.719 | 0.572 | 0.510 | 2.039 |
| | 80% | 0.487 | 0.484 | 3.234 | 0.995 | 0.788 | 2.793 | 0.879 | 0.707 | 3.007 | 0.961 | 0.746 | 1.718 | 0.570 | 0.509 | 2.019 |
| | 100% | 0.483 | 0.481 | 3.090 | 0.995 | 0.787 | 2.700 | 0.861 | 0.701 | 2.907 | 0.957 | 0.745 | 1.707 | 0.568 | 0.508 | 1.988 |
| SD | 10% | 0.408 | 0.443 | 5.532 | 1.027 | 0.741 | 4.052 | 0.304 | 0.427 | 3.838 | 1.149 | 0.797 | 2.285 | 0.634 | 0.472 | 2.958 |
| | 20% | 0.429 | 0.456 | 6.573 | 1.027 | 0.743 | 3.489 | 0.314 | 0.434 | 3.526 | 1.043 | 0.763 | 2.388 | 0.634 | 0.473 | 4.927 |
| | 50% | 0.425 | 0.454 | 7.190 | 1.027 | 0.746 | 3.391 | 0.319 | 0.434 | 3.228 | 0.980 | 0.746 | 2.113 | 0.639 | 0.478 | 4.018 |
| | 70% | 0.424 | 0.453 | 7.637 | 1.027 | 0.747 | 3.241 | 0.330 | 0.444 | 3.120 | 0.968 | 0.742 | 1.976 | 0.642 | 0.481 | 3.749 |
| | 80% | 0.430 | 0.456 | 7.557 | 1.027 | 0.748 | 3.225 | 0.335 | 0.446 | 3.084 | 0.966 | 0.741 | 2.014 | 0.643 | 0.483 | 3.677 |
| | 100% | 0.431 | 0.455 | 7.520 | 1.026 | 0.747 | 3.282 | 0.345 | 0.453 | 3.267 | 0.958 | 0.736 | 1.929 | 0.644 | 0.484 | 3.613 |
| NYCTaxi | 24 | 0.360 | 0.354 | 72.023 | 0.323 | 0.396 | 51.990 | 0.510 | 0.533 | 69.029 | 1.057 | 0.776 | 30.944 | 0.174 | 0.252 | 112.286 |
| | 48 | 0.361 | 0.354 | 41.951 | 0.330 | 0.400 | 36.132 | 0.570 | 0.577 | 40.947 | 1.019 | 0.765 | 17.091 | 0.189 | 0.261 | 59.510 |
| | 96 | 0.362 | 0.354 | 51.430 | 0.336 | 0.403 | 43.938 | 0.569 | 0.574 | 43.910 | 1.023 | 0.766 | 13.353 | 0.196 | 0.265 | 58.008 |
| | 192 | 0.361 | 0.354 | 58.848 | 0.341 | 0.407 | 53.038 | 0.574 | 0.573 | 71.913 | 1.015 | 0.763 | 10.608 | 0.208 | 0.271 | 50.885 |
| | 336 | 0.360 | 0.353 | 69.981 | 0.341 | 0.407 | 62.130 | 0.588 | 0.584 | 70.197 | 1.016 | 0.762 | 11.105 | 0.210 | 0.271 | 49.687 |
| | 720 | 0.360 | 0.353 | 63.802 | 0.341 | 0.407 | 62.789 | 0.589 | 0.585 | 69.971 | 1.018 | 0.761 | 11.730 | 0.216 | 0.276 | 56.289 |
| CHBike | 24 | 0.586 | 0.236 | 11.287 | 0.719 | 0.259 | 15.736 | 0.483 | 0.253 | 20.722 | 1.033 | 0.427 | 13.046 | 0.348 | 0.172 | 17.738 |
| | 48 | 0.598 | 0.238 | 54.795 | 0.720 | 0.258 | 49.071 | 0.528 | 0.263 | 23.159 | 1.044 | 0.429 | 7.855 | 0.350 | 0.177 | 18.073 |
| | 96 | 0.600 | 0.238 | 40.558 | 0.720 | 0.258 | 39.272 | 0.533 | 0.261 | 62.846 | 1.026 | 0.426 | 14.586 | 0.349 | 0.176 | 45.471 |
| | 192 | 0.597 | 0.237 | 54.833 | 0.721 | 0.259 | 53.351 | 0.536 | 0.263 | 76.555 | 1.016 | 0.423 | 15.592 | 0.347 | 0.175 | 57.547 |
| | 336 | 0.597 | 0.236 | 72.382 | 0.722 | 0.259 | 60.742 | 0.538 | 0.265 | 98.041 | 1.011 | 0.421 | 13.229 | 0.344 | 0.174 | 61.071 |
| | 720 | 0.600 | 0.237 | 110.547 | 0.723 | 0.259 | 89.368 | 0.538 | 0.264 | 136.728 | 1.006 | 0.416 | 17.090 | 0.361 | 0.187 | 86.629 |

Table 9: Full results of short/long-term forecasting comparing with baselines (2/2). The best scores are in **boldface**. The dataset names are abbreviated for briefness. "%" denotes the results are scaled by 1/100.

| System | | NDCN | | | GNS | | | ST-GODE | | | MT-GODE | | | PDEDER | | |
|---|---|---|---|---|---|---|---|---|---|---|---|---|---|---|---|---|
| | | MSE | MAE | MRAE | MSE | MAE | MRAE | MSE | MAE | MRAE | MSE | MAE | MRAE | MSE | MAE | MRAE |
| TDrive | 24 | 0.292 | 0.283 | 15736.0 | 0.225 | 0.266 | 7574.2 | 0.431 | 0.446 | 27129.3 | 0.814 | 0.595 | 10453.3 | 0.116 | 0.168 | 15286.5 |
| | 48 | 0.311 | 0.297 | 16404.9 | 0.250 | 0.283 | 12647.7 | 0.491 | 0.471 | 30402.6 | 0.841 | 0.615 | 9791.7 | 0.121 | 0.169 | 16292.3 |
| | 96 | 0.328 | 0.309 | 16976.4 | 0.269 | 0.296 | 15117.7 | 0.483 | 0.461 | 28853.0 | 0.858 | 0.622 | 10139.2 | 0.132 | 0.175 | 17015.3 |
| | 192 | 0.345 | 0.318 | 18483.8 | 0.284 | 0.306 | 14503.9 | 0.494 | 0.463 | 26906.7 | 0.880 | 0.635 | 8501.7 | 0.142 | 0.180 | 16026.1 |
| | 336 | 0.373 | 0.333 | 18978.9 | 0.309 | 0.324 | 14032.6 | 0.530 | 0.473 | 23672.2 | 0.905 | 0.651 | 6640.1 | 0.163 | 0.194 | 14970.5 |
| | 720 | 0.421 | 0.367 | 19281.9 | 0.350 | 0.352 | 13592.2 | 0.574 | 0.476 | 17588.1 | 0.947 | 0.675 | 4550.2 | 0.205 | 0.226 | 14584.5 |
| PEMS03 | 24 | 0.148 | 0.254 | 2.772 | 0.969 | 0.803 | 8.454 | 0.199 | 0.309 | 6.150 | 0.980 | 0.814 | 2.181 | 0.158 | 0.261 | 3.235 |
| | 48 | 0.198 | 0.297 | 6.223 | 1.022 | 0.826 | 7.937 | 0.200 | 0.311 | 7.615 | 1.016 | 0.835 | 3.276 | 0.213 | 0.306 | 5.120 |
| | 96 | 0.264 | 0.352 | 5.665 | 1.088 | 0.856 | 7.184 | 0.203 | 0.315 | 6.767 | 1.005 | 0.833 | 2.787 | 0.283 | 0.361 | 6.089 |
| | 192 | 0.317 | 0.395 | 6.167 | 1.141 | 0.881 | 7.096 | 0.206 | 0.319 | 6.655 | 1.014 | 0.839 | 3.075 | 0.328 | 0.398 | 8.279 |
| | 336 | 0.298 | 0.381 | 5.582 | 1.120 | 0.871 | 6.993 | 0.205 | 0.317 | 7.959 | 1.022 | 0.843 | 2.689 | 0.287 | 0.366 | 7.562 |
| | 720 | 0.335 | 0.411 | 5.651 | 1.138 | 0.879 | 8.358 | 0.205 | 0.317 | 7.027 | 1.024 | 0.844 | 2.469 | 0.311 | 0.389 | 7.290 |
| PEMS04 | 24 | 0.465 | 0.396 | 6.577 | 1.032 | 0.689 | 4.049 | 0.534 | 0.526 | 4.554 | 0.980 | 0.694 | 2.481 | 0.663 | 0.488 | 4.007 |
| | 48 | 0.545 | 0.443 | 8.405 | 1.029 | 0.688 | 3.813 | 0.466 | 0.493 | 6.532 | 0.986 | 0.699 | 2.563 | 0.719 | 0.521 | 5.163 |
| | 96 | 0.655 | 0.504 | 11.508 | 1.027 | 0.687 | 4.201 | 0.408 | 0.462 | 5.515 | 0.994 | 0.704 | 2.444 | 0.803 | 0.568 | 6.158 |
| | 192 | 0.751 | 0.557 | 12.697 | 1.026 | 0.687 | 4.092 | 0.393 | 0.455 | 5.551 | 1.003 | 0.708 | 2.384 | 0.874 | 0.607 | 6.079 |
| | 336 | 0.741 | 0.547 | 14.319 | 1.026 | 0.687 | 4.168 | 0.394 | 0.456 | 5.520 | 1.008 | 0.710 | 2.596 | 0.829 | 0.581 | 5.964 |
| | 720 | 0.822 | 0.584 | 17.939 | 1.026 | 0.687 | 4.317 | 0.382 | 0.450 | 5.688 | 1.008 | 0.710 | 2.508 | 0.840 | 0.586 | 5.916 |
| PEMS07 | 24 | 0.579 | 0.536 | 6.308 | 1.104 | 0.826 | 4.080 | 0.717 | 0.627 | 3.342 | 0.987 | 0.766 | 1.575 | 0.226 | 0.318 | 4.472 |
| | 48 | 0.578 | 0.535 | 6.438 | 1.098 | 0.824 | 4.099 | 0.716 | 0.627 | 3.320 | 0.989 | 0.772 | 1.551 | 0.300 | 0.370 | 5.353 |
| | 96 | 0.574 | 0.532 | 7.255 | 1.092 | 0.821 | 4.455 | 0.715 | 0.627 | 3.070 | 1.000 | 0.779 | 1.663 | 0.402 | 0.439 | 6.687 |
| | 192 | 0.574 | 0.533 | 7.332 | 1.090 | 0.820 | 4.471 | 0.715 | 0.627 | 3.439 | 1.002 | 0.781 | 1.796 | 0.479 | 0.493 | 6.500 |
| | 336 | 0.576 | 0.533 | 7.349 | 1.089 | 0.819 | 4.663 | 0.714 | 0.626 | 3.359 | 0.999 | 0.781 | 1.817 | 0.418 | 0.454 | 5.771 |
| | 720 | 0.575 | 0.533 | 7.609 | 1.088 | 0.819 | 4.788 | 0.714 | 0.626 | 3.259 | 1.000 | 0.782 | 1.767 | 0.439 | 0.472 | 6.644 |
| PEMS08 | 24 | 1.009 | 0.628 | 8.452 | 0.933 | 0.688 | 2.788 | 0.853 | 0.751 | 3.725 | 1.024 | 0.718 | 3.145 | 0.623 | 0.476 | 8.177 |
| | 48 | 1.011 | 0.628 | 10.270 | 0.934 | 0.688 | 3.150 | 0.845 | 0.748 | 3.553 | 1.018 | 0.717 | 2.839 | 0.663 | 0.503 | 7.441 |
| | 96 | 1.011 | 0.628 | 11.651 | 0.935 | 0.688 | 3.638 | 0.862 | 0.753 | 4.006 | 1.009 | 0.715 | 2.619 | 0.729 | 0.544 | 7.716 |
| | 192 | 1.004 | 0.625 | 14.057 | 0.935 | 0.688 | 3.887 | 0.879 | 0.759 | 4.162 | 1.004 | 0.714 | 2.768 | 0.791 | 0.582 | 8.775 |
| | 336 | 1.003 | 0.624 | 13.029 | 0.935 | 0.688 | 3.784 | 0.872 | 0.756 | 4.040 | 1.008 | 0.715 | 2.721 | 0.768 | 0.569 | 8.102 |
| | 720 | 1.007 | 0.625 | 12.479 | 0.935 | 0.687 | 3.977 | 0.876 | 0.758 | 4.507 | 1.012 | 0.716 | 3.111 | 0.765 | 0.567 | 8.050 |
| NOAA | 24 | 0.485 | 0.510 | 8.817 | 0.567 | 0.560 | 17.798 | 0.578 | 0.568 | 3.129 | 0.952 | 0.708 | 5.074 | 0.324 | 0.408 | 17.031 |
| | 48 | 0.521 | 0.532 | 13.585 | 0.603 | 0.580 | 20.415 | 0.592 | 0.575 | 2.682 | 0.962 | 0.717 | 6.263 | 0.399 | 0.456 | 21.904 |
| | 96 | 0.622 | 0.584 | 15.519 | 0.700 | 0.626 | 22.065 | 0.648 | 0.604 | 2.853 | 0.990 | 0.730 | 5.623 | 0.533 | 0.524 | 26.829 |
| | 192 | 0.828 | 0.677 | 15.616 | 0.900 | 0.710 | 21.805 | 0.727 | 0.643 | 4.433 | 1.005 | 0.739 | 6.472 | 0.684 | 0.597 | 24.321 |
| | 336 | 0.840 | 0.681 | 14.823 | 0.914 | 0.715 | 21.309 | 0.738 | 0.649 | 3.735 | 1.014 | 0.745 | 6.714 | 0.742 | 0.629 | 22.113 |
| | 720 | 0.839 | 0.677 | 13.466 | 0.907 | 0.711 | 21.129 | 0.738 | 0.648 | 3.760 | 1.020 | 0.751 | 6.312 | 0.835 | 0.678 | 16.534 |

Table 10: Full results of short/long-term forecasting on variants of PDEDER (1/2). The best scores are in **boldface**. The dataset names are abbreviated for briefness. "%" denotes the results are scaled by 1/100.

| System | | PDEDER MSE | MAE | MRAE | PDEDER-nopre MSE | MAE | MRAE | PDEDER-frz MSE | MAE | MRAE | PDEDER-sys MSE | MAE | MRAE |
|---|---|---|---|---|---|---|---|---|---|---|---|---|---|
| Mutualistic | 10% | 0.276 | 0.386 | 5.702 | 0.420 | 0.516 | 6.379 | 0.213 | 0.356 | 5.949 | 0.298 | 0.415 | 6.136 |
| | 20% | 0.448 | 0.518 | 5.820 | 0.642 | 0.669 | 6.491 | 0.414 | 0.504 | 6.075 | 0.491 | 0.562 | 6.208 |
| | 50% | 0.750 | 0.656 | 3.675 | 0.971 | 0.805 | 4.140 | 0.774 | 0.678 | 3.896 | 0.840 | 0.718 | 3.947 |
| | 70% | 0.813 | 0.679 | 2.879 | 1.008 | 0.819 | 3.256 | 0.827 | 0.698 | 3.040 | 0.915 | 0.745 | 3.097 |
| | 80% | 0.829 | 0.683 | 2.625 | 1.014 | 0.817 | 2.972 | 0.836 | 0.698 | 2.762 | 0.931 | 0.747 | 2.824 |
| | 100% | 0.844 | 0.683 | 2.259 | 1.020 | 0.808 | 2.564 | 0.835 | 0.687 | 2.359 | 0.939 | 0.742 | 2.430 |
| Heat | 10% | 0.003 | 0.045 | 0.307 | 0.027 | 0.135 | 0.910 | 0.008 | 0.068 | 0.782 | 0.009 | 0.078 | 0.639 |
| | 20% | 0.003 | 0.045 | 0.264 | 0.027 | 0.136 | 0.825 | 0.007 | 0.067 | 0.667 | 0.009 | 0.078 | 0.593 |
| | 50% | 0.004 | 0.048 | 0.885 | 0.030 | 0.136 | 3.643 | 0.008 | 0.064 | 1.438 | 0.009 | 0.079 | 1.298 |
| | 70% | 0.005 | 0.051 | 3.719 | 0.031 | 0.136 | 12.385 | 0.009 | 0.067 | 3.055 | 0.010 | 0.079 | 3.448 |
| | 80% | 0.006 | 0.052 | 4.536 | 0.032 | 0.137 | 13.063 | 0.010 | 0.072 | 3.975 | 0.010 | 0.079 | 4.136 |
| | 100% | 0.009 | 0.059 | 6.581 | 0.037 | 0.142 | 14.490 | 0.015 | 0.086 | 7.144 | 0.012 | 0.082 | 6.230 |
| 2D CFD | 10% | 0.230 | 0.309 | 1.158 | 0.231 | 0.309 | 1.170 | 0.232 | 0.317 | 1.196 | 0.229 | 0.303 | 1.069 |
| | 20% | 0.216 | 0.296 | 1.212 | 0.217 | 0.296 | 1.221 | 0.218 | 0.304 | 1.249 | 0.215 | 0.291 | 1.122 |
| | 50% | 0.177 | 0.261 | 1.539 | 0.179 | 0.261 | 1.704 | 0.180 | 0.270 | 1.734 | 0.176 | 0.256 | 1.389 |
| | 70% | 0.155 | 0.240 | 1.715 | 0.158 | 0.241 | 1.878 | 0.158 | 0.249 | 1.872 | 0.154 | 0.236 | 1.590 |
| | 80% | 0.146 | 0.230 | 1.861 | 0.149 | 0.231 | 2.044 | 0.149 | 0.239 | 2.003 | 0.145 | 0.226 | 1.731 |
| | 100% | 0.130 | 0.212 | 2.118 | 0.119 | 0.205 | 2.325 | 0.133 | 0.221 | 2.248 | 0.129 | 0.209 | 1.993 |
| DarcyFlow | 10% | 0.065% | 1.998% | 1.404 | 0.092% | 2.338% | 3.679 | 0.077% | 2.140% | 2.333 | 0.065% | 2.002% | 1.376 |
| | 20% | 0.069% | 2.034% | 1.392 | 0.090% | 2.307% | 3.131 | 0.078% | 2.154% | 2.316 | 0.069% | 2.044% | 1.401 |
| | 50% | 0.072% | 2.063% | 1.486 | 0.086% | 2.256% | 2.156 | 0.080% | 2.176% | 2.479 | 0.072% | 2.079% | 1.476 |
| | 70% | 0.072% | 2.065% | 1.468 | 0.085% | 2.236% | 2.059 | 0.080% | 2.180% | 2.538 | 0.073% | 2.086% | 1.445 |
| | 80% | 0.072% | 2.065% | 1.449 | 0.085% | 2.231% | 2.079 | 0.080% | 2.181% | 2.549 | 0.073% | 2.088% | 1.428 |
| | 100% | 0.072% | 2.065% | 1.426 | 0.084% | 2.224% | 2.227 | 0.080% | 2.180% | 2.594 | 0.074% | 2.092% | 1.405 |
| Gene | 10% | 0.033 | 0.135 | 1.528 | 0.138 | 0.309 | 1.790 | 0.101 | 0.224 | 2.342 | 0.051 | 0.183 | 1.652 |
| | 20% | 0.036 | 0.138 | 1.499 | 0.141 | 0.312 | 1.704 | 0.124 | 0.248 | 2.409 | 0.053 | 0.185 | 1.590 |
| | 50% | 0.052 | 0.152 | 1.796 | 0.151 | 0.319 | 1.817 | 0.219 | 0.344 | 3.461 | 0.058 | 0.184 | 1.784 |
| | 70% | 0.069 | 0.167 | 1.928 | 0.159 | 0.320 | 1.873 | 0.303 | 0.414 | 3.963 | 0.066 | 0.188 | 1.891 |
| | 80% | 0.079 | 0.176 | 1.985 | 0.163 | 0.320 | 1.894 | 0.348 | 0.447 | 4.092 | 0.073 | 0.193 | 1.928 |
| | 100% | 0.103 | 0.194 | 2.268 | 0.177 | 0.324 | 2.066 | 0.438 | 0.508 | 4.638 | 0.089 | 0.204 | 2.243 |
| ShallowWater | 10% | 0.582 | 0.327 | 1.611 | 0.994 | 0.415 | 0.732 | 0.713 | 0.356 | 0.923 | 0.733 | 0.353 | 0.965 |
| | 20% | 0.766 | 0.389 | 2.183 | 1.006 | 0.418 | 1.450 | 0.794 | 0.385 | 1.565 | 0.813 | 0.382 | 1.643 |
| | 50% | 1.085 | 0.500 | 1.930 | 1.044 | 0.434 | 1.071 | 0.985 | 0.458 | 1.406 | 0.985 | 0.445 | 1.406 |
| | 70% | 1.152 | 0.529 | 1.938 | 1.057 | 0.438 | 1.065 | 1.036 | 0.480 | 1.421 | 1.033 | 0.466 | 1.409 |
| | 80% | 1.165 | 0.536 | 1.866 | 1.061 | 0.439 | 1.038 | 1.050 | 0.485 | 1.380 | 1.047 | 0.471 | 1.364 |
| | 100% | 1.179 | 0.543 | 2.106 | 1.066 | 0.440 | 1.315 | 1.066 | 0.490 | 1.620 | 1.063 | 0.475 | 1.612 |
| 2D DiffReac | 10% | 0.945 | 0.715 | 4.918 | 0.986 | 0.737 | 5.565 | 0.959 | 0.728 | 4.877 | 0.964 | 0.721 | 5.508 |
| | 20% | 0.975 | 0.731 | 4.965 | 0.976 | 0.738 | 4.966 | 0.957 | 0.732 | 4.391 | 0.981 | 0.733 | 5.028 |
| | 50% | 1.061 | 0.789 | 4.292 | 0.993 | 0.765 | 3.619 | 1.000 | 0.769 | 3.440 | 1.034 | 0.780 | 3.964 |
| | 70% | 1.053 | 0.792 | 4.123 | 1.004 | 0.776 | 3.628 | 1.006 | 0.777 | 3.468 | 1.030 | 0.785 | 3.819 |
| | 80% | 1.055 | 0.795 | 3.984 | 1.009 | 0.780 | 3.463 | 1.010 | 0.780 | 3.355 | 1.032 | 0.788 | 3.659 |
| | 100% | 1.061 | 0.800 | 3.751 | 1.016 | 0.786 | 3.381 | 1.021 | 0.787 | 3.224 | 1.041 | 0.793 | 3.548 |
| LA | 10% | 0.581 | 0.516 | 2.405 | 0.590 | 0.519 | 2.787 | 0.580 | 0.514 | 2.390 | 0.585 | 0.516 | 2.503 |
| | 20% | 0.582 | 0.516 | 2.245 | 0.590 | 0.519 | 2.655 | 0.580 | 0.514 | 2.240 | 0.584 | 0.515 | 2.358 |
| | 50% | 0.575 | 0.512 | 2.087 | 0.581 | 0.515 | 2.456 | 0.574 | 0.510 | 2.098 | 0.576 | 0.511 | 2.191 |
| | 70% | 0.572 | 0.510 | 2.039 | 0.577 | 0.513 | 2.380 | 0.571 | 0.509 | 2.046 | 0.572 | 0.508 | 2.134 |
| | 80% | 0.570 | 0.509 | 2.019 | 0.575 | 0.511 | 2.340 | 0.570 | 0.508 | 2.027 | 0.571 | 0.507 | 2.103 |
| | 100% | 0.568 | 0.508 | 1.988 | 0.572 | 0.510 | 2.293 | 0.567 | 0.507 | 1.990 | 0.567 | 0.505 | 2.067 |
| SD | 10% | 0.634 | 0.472 | 2.958 | 0.663 | 0.498 | 3.099 | 0.641 | 0.479 | 3.018 | 0.642 | 0.479 | 3.029 |
| | 20% | 0.634 | 0.473 | 4.927 | 0.668 | 0.502 | 5.086 | 0.642 | 0.481 | 4.996 | 0.641 | 0.479 | 4.941 |
| | 50% | 0.639 | 0.478 | 4.018 | 0.672 | 0.507 | 4.023 | 0.648 | 0.486 | 4.043 | 0.646 | 0.484 | 4.016 |
| | 70% | 0.642 | 0.481 | 3.749 | 0.675 | 0.509 | 3.723 | 0.652 | 0.489 | 3.780 | 0.648 | 0.486 | 3.723 |
| | 80% | 0.643 | 0.483 | 3.677 | 0.676 | 0.511 | 3.623 | 0.653 | 0.491 | 3.712 | 0.649 | 0.488 | 3.643 |
| | 100% | 0.644 | 0.484 | 3.613 | 0.675 | 0.511 | 3.562 | 0.654 | 0.492 | 3.683 | 0.649 | 0.489 | 3.581 |

Table 11: Full results of short/long-term forecasting on variants of PDEDER (2/2). The best scores are in **boldface**. The dataset names are abbreviated for briefness. "%" denotes the results are scaled by 1/100.

| System | | PDEDER | | | PDEDER-nopre | | | PDEDER-frz | | | PDEDER-sys | | |
|---|---|---|---|---|---|---|---|---|---|---|---|---|---|
| | | MSE | MAE | MRAE | MSE | MAE | MRAE | MSE | MAE | MRAE | MSE | MAE | MRAE |
| NYCTaxi | 24 | 0.174 | 0.252 | 112.286 | 0.179 | 0.257 | 113.806 | 0.211 | 0.300 | 83.433 | 0.185 | 0.263 | 114.767 |
| | 48 | 0.189 | 0.261 | 59.510 | 0.197 | 0.273 | 60.461 | 0.212 | 0.293 | 44.679 | 0.198 | 0.269 | 60.569 |
| | 96 | 0.196 | 0.265 | 58.008 | 0.211 | 0.281 | 60.900 | 0.214 | 0.284 | 47.414 | 0.206 | 0.272 | 57.876 |
| | 192 | 0.208 | 0.271 | 50.885 | 0.216 | 0.282 | 53.787 | 0.223 | 0.285 | 44.304 | 0.216 | 0.278 | 50.721 |
| | 336 | 0.210 | 0.271 | 49.687 | 0.213 | 0.276 | 52.057 | 0.215 | 0.278 | 46.323 | 0.214 | 0.277 | 51.752 |
| | 720 | 0.216 | 0.276 | 56.289 | 0.235 | 0.297 | 56.623 | 0.243 | 0.296 | 52.163 | 0.233 | 0.288 | 58.768 |
| CHBike | 24 | 0.348 | 0.172 | 17.738 | 0.520 | 0.213 | 23.882 | 0.431 | 0.198 | 19.476 | 0.302 | 0.165 | 19.628 |
| | 48 | 0.350 | 0.177 | 18.073 | 0.501 | 0.211 | 23.543 | 0.402 | 0.190 | 18.030 | 0.315 | 0.170 | 19.136 |
| | 96 | 0.349 | 0.176 | 45.471 | 0.466 | 0.204 | 52.778 | 0.384 | 0.183 | 43.365 | 0.322 | 0.172 | 42.505 |
| | 192 | 0.347 | 0.175 | 57.547 | 0.426 | 0.195 | 67.644 | 0.382 | 0.181 | 56.502 | 0.329 | 0.174 | 53.260 |
| | 336 | 0.344 | 0.174 | 61.071 | 0.394 | 0.187 | 69.650 | 0.374 | 0.179 | 60.060 | 0.332 | 0.174 | 56.850 |
| | 720 | 0.361 | 0.187 | 86.629 | 0.369 | 0.180 | 114.281 | 0.411 | 0.198 | 84.584 | 0.348 | 0.181 | 97.035 |
| TDrive | 24 | 0.116 | 0.168 | 15286.5 | 0.164 | 0.211 | 19079.9 | 0.154 | 0.197 | 14473.9 | 0.116 | 0.170 | 15618.7 |
| | 48 | 0.121 | 0.169 | 16292.3 | 0.167 | 0.216 | 20283.7 | 0.150 | 0.191 | 15643.0 | 0.121 | 0.171 | 16655.2 |
| | 96 | 0.132 | 0.175 | 17015.3 | 0.168 | 0.219 | 20387.7 | 0.152 | 0.192 | 16497.6 | 0.131 | 0.176 | 17389.7 |
| | 192 | 0.142 | 0.180 | 16026.1 | 0.170 | 0.219 | 18704.0 | 0.157 | 0.198 | 15550.5 | 0.142 | 0.182 | 16384.3 |
| | 336 | 0.163 | 0.194 | 14970.5 | 0.186 | 0.225 | 17032.2 | 0.174 | 0.212 | 14556.4 | 0.164 | 0.198 | 15269.7 |
| | 720 | 0.205 | 0.226 | 14584.5 | 0.220 | 0.246 | 16234.7 | 0.216 | 0.242 | 14261.4 | 0.207 | 0.230 | 14959.7 |
| PEMS03 | 24 | 0.158 | 0.261 | 3.235 | 0.170 | 0.287 | 3.684 | 0.157 | 0.273 | 3.441 | 0.158 | 0.264 | 3.560 |
| | 48 | 0.213 | 0.306 | 5.120 | 0.233 | 0.335 | 5.774 | 0.215 | 0.317 | 5.535 | 0.218 | 0.312 | 5.698 |
| | 96 | 0.283 | 0.361 | 6.089 | 0.313 | 0.393 | 6.842 | 0.287 | 0.369 | 6.283 | 0.289 | 0.367 | 6.698 |
| | 192 | 0.328 | 0.398 | 8.279 | 0.367 | 0.432 | 9.395 | 0.331 | 0.403 | 8.364 | 0.333 | 0.403 | 8.922 |
| | 336 | 0.287 | 0.366 | 7.562 | 0.321 | 0.399 | 8.728 | 0.288 | 0.369 | 7.494 | 0.287 | 0.369 | 8.140 |
| | 720 | 0.311 | 0.389 | 7.290 | 0.330 | 0.406 | 8.446 | 0.301 | 0.385 | 7.427 | 0.321 | 0.398 | 7.726 |
| PEMS04 | 24 | 0.663 | 0.488 | 4.007 | 0.671 | 0.502 | 4.484 | 0.649 | 0.482 | 3.975 | 0.652 | 0.479 | 3.946 |
| | 48 | 0.719 | 0.521 | 5.163 | 0.734 | 0.537 | 5.758 | 0.708 | 0.517 | 5.173 | 0.710 | 0.515 | 5.165 |
| | 96 | 0.803 | 0.568 | 6.158 | 0.834 | 0.590 | 6.948 | 0.798 | 0.568 | 6.221 | 0.797 | 0.565 | 6.177 |
| | 192 | 0.874 | 0.607 | 6.079 | 0.921 | 0.634 | 6.871 | 0.876 | 0.610 | 6.235 | 0.869 | 0.605 | 6.113 |
| | 336 | 0.829 | 0.581 | 5.964 | 0.868 | 0.605 | 6.748 | 0.830 | 0.582 | 6.105 | 0.825 | 0.579 | 6.034 |
| | 720 | 0.840 | 0.586 | 5.916 | 0.876 | 0.606 | 6.630 | 0.838 | 0.586 | 5.899 | 0.836 | 0.585 | 6.094 |
| PEMS07 | 24 | 0.226 | 0.318 | 4.472 | 0.256 | 0.356 | 5.075 | 0.194 | 0.294 | 4.553 | 0.190 | 0.291 | 4.123 |
| | 48 | 0.300 | 0.370 | 5.353 | 0.348 | 0.417 | 6.265 | 0.281 | 0.358 | 5.601 | 0.277 | 0.356 | 5.205 |
| | 96 | 0.402 | 0.439 | 6.687 | 0.469 | 0.491 | 7.996 | 0.394 | 0.436 | 7.013 | 0.391 | 0.435 | 6.815 |
| | 192 | 0.479 | 0.493 | 6.500 | 0.556 | 0.546 | 7.721 | 0.472 | 0.491 | 6.764 | 0.476 | 0.494 | 6.721 |
| | 336 | 0.418 | 0.454 | 5.771 | 0.480 | 0.498 | 6.809 | 0.413 | 0.452 | 5.993 | 0.417 | 0.455 | 6.063 |
| | 720 | 0.439 | 0.472 | 6.644 | 0.473 | 0.499 | 7.525 | 0.425 | 0.465 | 6.416 | 0.439 | 0.474 | 7.261 |
| PEMS08 | 24 | 0.623 | 0.476 | 8.177 | 0.642 | 0.509 | 9.626 | 0.621 | 0.476 | 8.323 | 0.617 | 0.475 | 8.631 |
| | 48 | 0.663 | 0.503 | 7.441 | 0.697 | 0.541 | 8.647 | 0.660 | 0.502 | 7.589 | 0.661 | 0.504 | 7.731 |
| | 96 | 0.729 | 0.544 | 7.716 | 0.786 | 0.588 | 9.096 | 0.727 | 0.544 | 7.804 | 0.729 | 0.547 | 7.954 |
| | 192 | 0.791 | 0.582 | 8.775 | 0.867 | 0.630 | 10.622 | 0.788 | 0.580 | 8.884 | 0.789 | 0.583 | 8.975 |
| | 336 | 0.768 | 0.569 | 8.102 | 0.831 | 0.610 | 9.663 | 0.764 | 0.567 | 8.191 | 0.764 | 0.569 | 8.254 |
| | 720 | 0.765 | 0.567 | 8.050 | 0.812 | 0.600 | 9.084 | 0.758 | 0.565 | 8.109 | 0.763 | 0.566 | 8.036 |
| NOAA | 24 | 0.324 | 0.408 | 17.031 | 0.331 | 0.415 | 15.150 | 0.375 | 0.444 | 19.176 | 0.336 | 0.418 | 17.995 |
| | 48 | 0.399 | 0.456 | 21.904 | 0.385 | 0.447 | 17.113 | 0.429 | 0.469 | 21.112 | 0.409 | 0.462 | 22.900 |
| | 96 | 0.533 | 0.524 | 26.829 | 0.499 | 0.509 | 22.450 | 0.542 | 0.525 | 23.307 | 0.554 | 0.533 | 27.712 |
| | 192 | 0.684 | 0.597 | 24.321 | 0.633 | 0.579 | 20.795 | 0.690 | 0.597 | 22.598 | 0.697 | 0.600 | 24.067 |
| | 336 | 0.742 | 0.629 | 22.113 | 0.692 | 0.610 | 17.663 | 0.774 | 0.639 | 20.318 | 0.749 | 0.626 | 21.997 |
| | 720 | 0.835 | 0.678 | 16.534 | 0.815 | 0.671 | 13.298 | 0.876 | 0.698 | 17.635 | 0.829 | 0.669 | 15.774 |

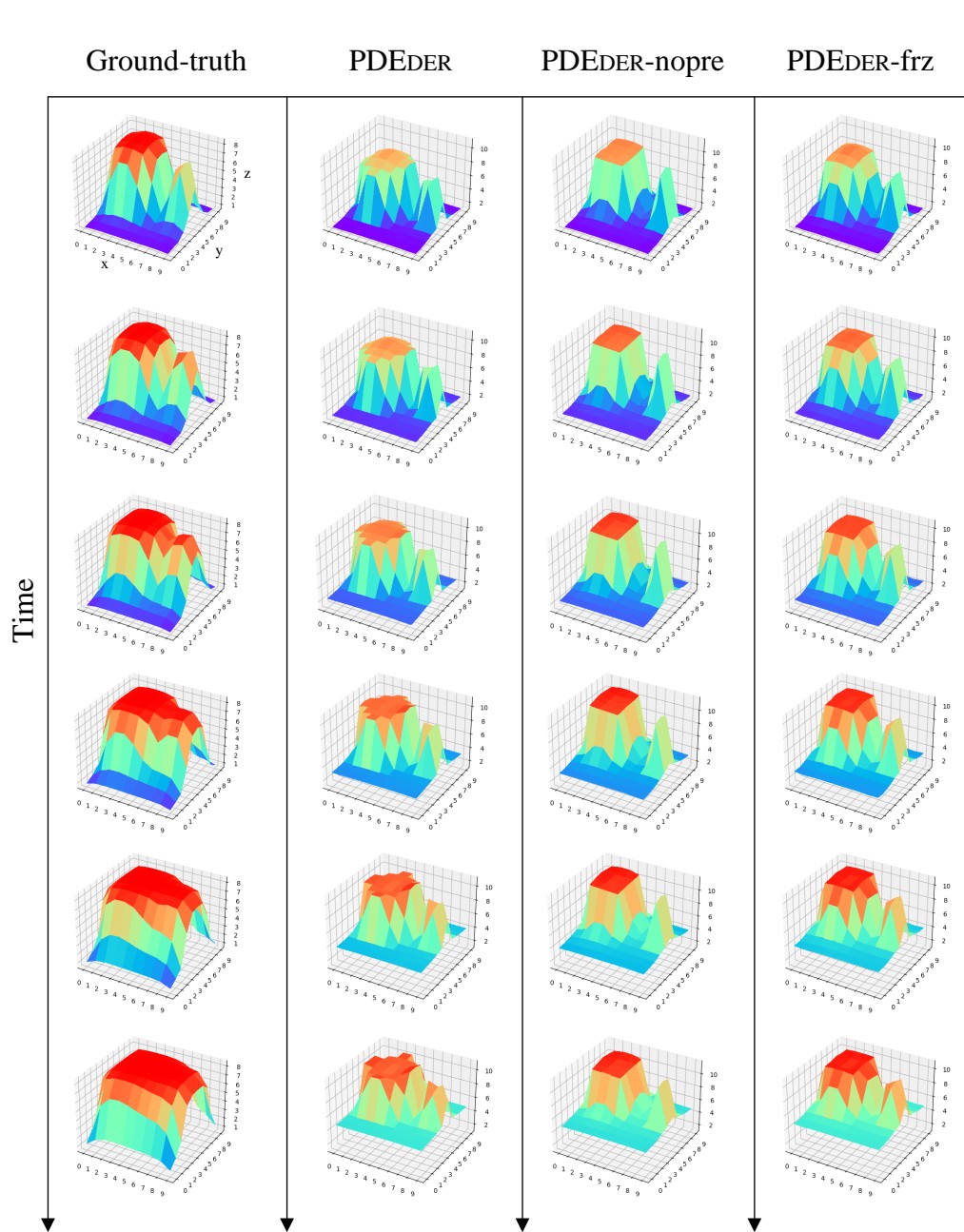

Figure 2: Forecasting visualizations on Mutualistic dynamics evolving comparisons between the ground-truth values, fine-tuning PDEDER, fine-tuning without pre-training PDEDER and fine-tuning with freezing the pre-trained PDEDER. Axes "x" and "y" denote the indexes of each object; axis "z" denotes the state values of each object.

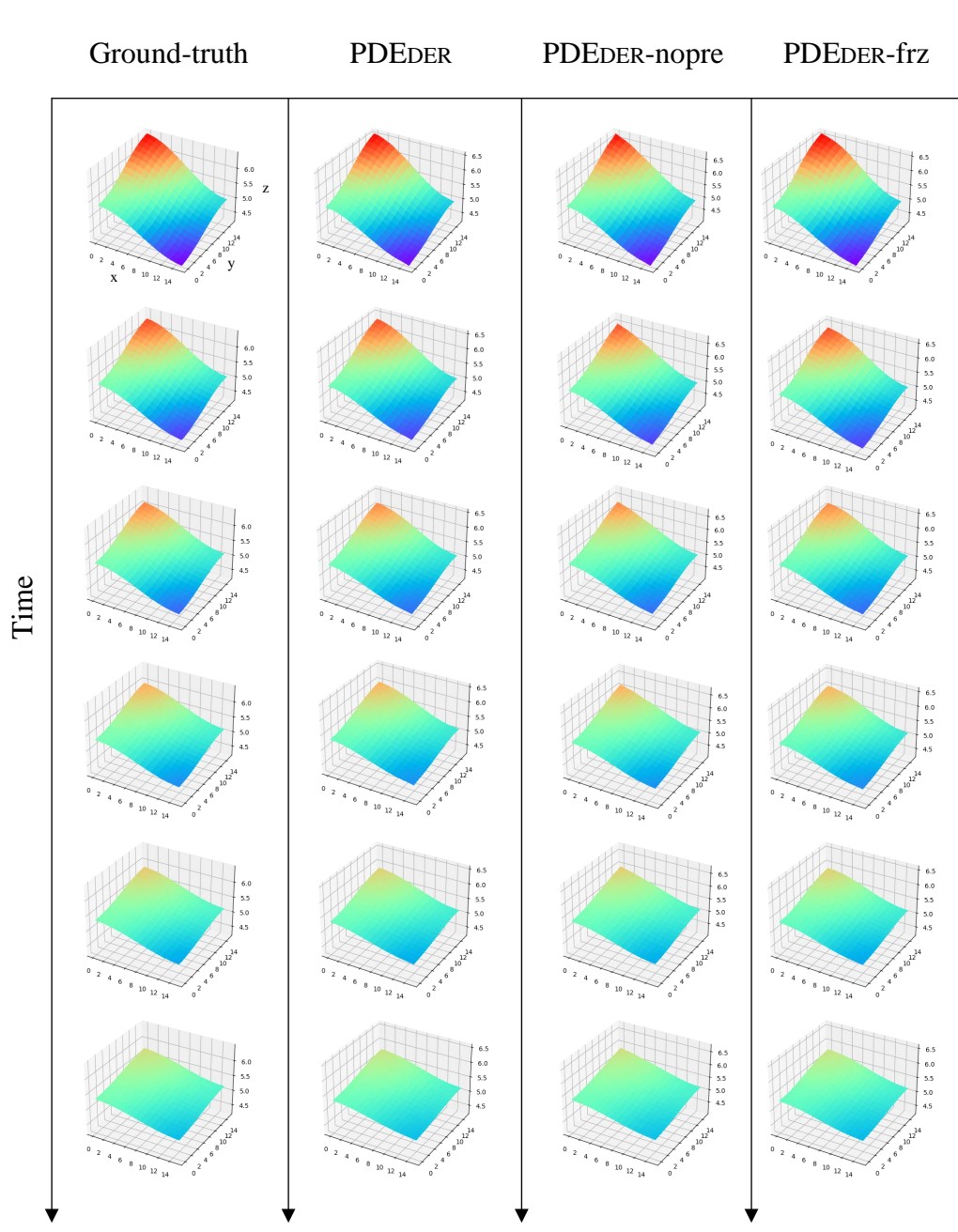

Figure 3: Forecasting visualizations on Heat dynamics evolving comparisons between the ground-truth values, fine-tuning PDEDER, fine-tuning without pre-training PDEDER and fine-tuning with freezing the pre-trained PDEDER. Axes "x" and "y" denote the indexes of each object; axis "z" denotes the state values of each object.

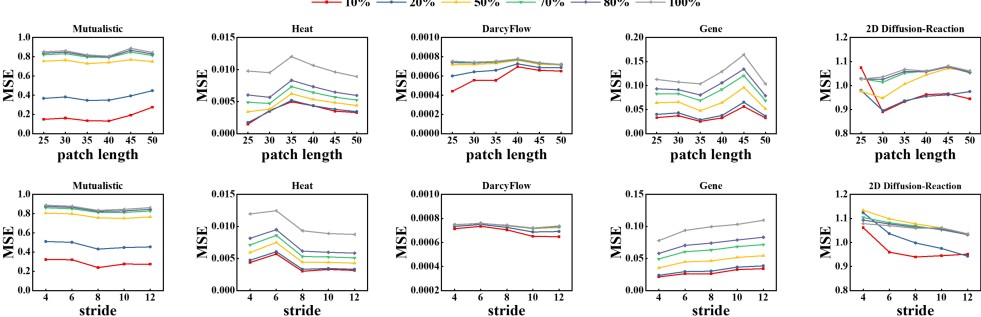

Figure 4: Sensitivity study results MSE on patch length and stride during fine-tuning PDEDER.

