# OpenReview forum: "Generalizing Dynamics Modeling Easier from Representation Perspective"
_ICLR.cc/2025/Conference — Submitted to ICLR 2025_

### Official Review · Reviewer_bAXr · 2024-11-04

**Soundness:** 3
**Presentation:** 3
**Contribution:** 2
**Rating:** 5
**Confidence:** 4

**Summary:**

This paper introduces PDEDER (Pre-trained Dynamic EncoDER), a method to generalize dynamics modeling by embedding original states into a latent space using pre-trained language models.  The key idea is to pre-train an encoder using language models that can embed states from various dynamic systems into a latent space where dynamics can be more easily captured and fine-tuned for specific systems. The key contribution is a framework that pre-trains 153 sets of observations from 24 complex systems, followed by fine-tuning for specific dynamics. The method employs data projection, tokenization, and PLM-based encoding to learn dynamics-enriched embeddings. The authors evaluate 18 dynamic systems through long/short-term forecasting under in- and cross-domain settings.

**Strengths:**

* Novel approach using pre-trained models for dynamics modeling generalization
* Comprehensive dataset collection across multiple domains
* Clear empirical validation through both in-domain and cross-domain experiments
* Strong cross-domain performance after fine-tuning, even when excluding entire systems during pre-training

**Weaknesses:**

* Difference between short-term and long-term forecasting in results  is not defined
* The connection between the pre-training objectives and dynamics modeling is not formally established
* Experimental limitations:
    1. The ablation studies don't fully isolate the contribution of each component
    2. Comparison with TANGO is limited to small-scale systems due to memory constraints. It would be useful to provide memory complexity analysis, Discuss potential solutions for scaling to larger systems
    3. The selection of 24 systems lacks justification for representativeness
* Technical details requiring clarification:
    1. How is the data projection module's dimension chosen?
    2. What is the impact of different PLM architectures?
    3. How sensitive is the method to tokenization parameters?
* The core idea of using pre-trained models for time series/dynamics has been explored in recent works (as cited in Related Work section, e.g., Gruver et al. 2024, PromptCast, AutoTimes). While this paper applies it to dynamics modeling in a new way, the conceptual novelty is incremental
Minor comment:
* Memory requirements should be discussed earlier in the paper

**Questions:**

* Why was T5 chosen as the base PLM? Have other architectures been considered?
* The data projection module reduces dimensionality to 1, but the justification for this choice isn't clear. Could this limit the model's expressiveness for complex systems?
* The data projection module seems crucial for handling different state dimensions. How do you determine the optimal projection dimension?
* The tokenization process (patch length 30, stride 6) seems to work well empirically, but how sensitive is the model to these choices? Some analysis of this would be valuable.
* Could you provide theoretical analysis/insights on why the pre-training objectives (reconstruction and forecasting) are sufficient for learning dynamics-enriched embeddings?
* How does the choice of T5 as the base PLM impact results? Have you tried other architectures?
* For the cross-domain experiments, how do you ensure the held-out systems are sufficiently different from the training systems?
What properties of the dynamics are preserved in the embedding space?

---

> ### Author Response · Authors · 2024-11-24
> **Response to Reviewer #4 (bAXr) (1/2)**
>
> ### **Q1. Difference between short-term and long-term forecasting**
> Thanks for pointing out this issue. The differences are presented below. We also rectified the corresponding descriptions in our paper (see Implementation Details in 5.2).
>
> > **Short/Long-term Forecasting** The training sequence length are same for both short and long term forecasting. For NYCTaxi, CHIBike, TDrive, PEMS03, PEMS04, PEMS07, PEMS08 and NOAA, the short- and long-term forecasting lengths are set as \{24,48\} and \{96, 192, 336, 720\}. For the rest dynamics, due to the diversity of convergence characteristics on each system, we truncate the test sequences by ratios to form the short/long-term forecasting sequences. The ratios for short- and long-term are set as \{10\%, 20\%\} and  \{50\%, 70\%, 80\%, 100\%\}, respectively. For example, when the test sequence length is 200, we set 10\% * 200=20 and 20\% * 200=40 as the forecasting lengths.
> ***
>
> ### **Q2. The connection between the pre-training objectives and dynamics modeling.**
> Thanks for pointing out this issue. We added a section to describe the model training processes detailedly and an Algorithm to clarify the connections of the two processes (see Section 4.4 and Algorithm 1 and 2).
> ***
>
> ### **Q3. Ablation study don't fully isolate the contribution of each component.**
> We are grateful for being pointed out this issue. We renamed this section from "Ablative Study" into "Impact Evaluation of Pre-training on downstream Dynamics Modeling". In this section, we examined the impact of our pre-training process on the downstream dynamics modeling by setting two different initialization strategies on en/decoder parameters in fine-tuning. The two versions corresponds with 1) fine-tuning PDEDER without pre-training and 2) fine-tuning PDEDER with freezing our pre-trained en/decoder.
> ***
>
> ### **Q4. Memory complexity discussions.**
> We kindly argue that our proposal requires less memory comparing with baseline methods. For example, fine-tuning on Mutualistic requires less than 2500 MB memories with patch length 50 and stride 10.
> ***
>
> ### **Q5. The selection of 24 systems lacks justification for representativeness.**
> We collected systems which are representative and commonly used in dynamics modeling. For variety, we collected systems from various domains, including physics, fluid, biology, climate and traffic system. These dynamics have been widely used in researches of dynamics modeling.
> ***
>
> ### **Q6. Data projection module's dimension**
> We modified this section in our latest version, and details are presented in the general comment above.
> ***
>
>
> ### **Q7. Sensitivity of tokenization parameters (patch size and stride).**
> We added sensitivity analysis on patch length and stride in our latest version. Details are presented in the general comment above and updated in our latest PDF version (see Figure 4 in Appendix).
>
> ***
>
> ### **Q8. The conceptual novelty is incremental against the pre-trained models for time series forecasting.**
> We kindly argue that our main contribution lies in pre-training to learn better representations for downstream dynamics learning rather than directly fine-tuning the pre-trained models. Our \baby concentrates on how to learn better dynamics-enriched representations. And the dynamics modeling module can be analogous to the classification head or regression head when fine-tuning a language model for downstream tasks. We want to highlight that, in the pre-training period, no system-specific dynamics are approximated, and the pre-training process only learn better embeddings for the observed sequences. And the interacting graphs are not considered in pre-training.
>
> In this way, the generalizability of our model lies in the representation level, rather than the dynamics learning level. After pre-training the embedder, we can learn generalizable embeddings for observations for any dynamics system, and these embeddings could be used for approximating dynamics by fine-tuning with any specific dynamics modeling methods. In this way, our embedder pre-training process could benefit approximating dynamics models more easily.
> ***
>
> ### **Q9. The reason of choosing PLM architectures. other PLMs.**
> We consider to choose PLMs owning encoder and decoder according to the basic idea of embedding the original states into latent space and learn dynamics in it. It could be substituted into any other suitable PLMs, even without a decoder, which may leading to the posterior collapsing. We will try to explore the effects of PLMs architectures on downstream dynamics modeling in our future work.

---

> > ### Author Response · Authors · 2024-11-24
> > **Response to Reviewer \#4 (bAXr) (2/2)**
> >
> > ### **Q11. For the cross-domain experiments, how do you ensure the held-out systems are sufficiently different from the training systems?**
> > When generating observations, we ensure the difference by setting quite different system-specific hyper-parameters, number of objects and sequence lengths. Dynamics behaviors strongly rely on the system-specific hyper-parameters. And we indeed observed obvious distinctions on systems with different parameters.
> > ***
> >
> > ### **Q12. Theoretical analysis/insights on why the pre-training objectives (reconstruction and forecasting) are sufficient for learning dynamics-enriched embeddings. What properties of the dynamics are preserved in the embedding space?**
> >
> > We mainly focus on the forecasting ability in the embedding space. Apart from the basic reconstruction task, to enhance the capacity of forecasting is essential when extracting the hidden evolving regularity of hidden dynamics. It can directly reflect how the model describe a specific dynamics and is an simple-yet-essential validation strategy measuring the quality of an approximated dynamics. And forecasting is one of most commonly used capacity in the real-world applications of dynamics models. Therefore, the pre-training objectives intuitively make sense for learning dynamics-enriched embeddings.
> > ***
> > We hope to hear back from you if you have further questions.

---

> > > ### Comment · Reviewer_bAXr · 2024-11-27
> > >
> > > Thank you for the thorough response to my review. I cannot increase my score of 5, as I still have several concerns which are not addressed:
> > >
> > > 1. The new MRAE results (>100% errors in many cases) indicate more limited practical effectiveness than MSE/MAE metrics initially suggested.
> > > 2. The data projection module's flattening approach still lacks theoretical/empirical justification. Your response describes the method but does not explain why dimension reduction to 1 is appropriate.
> > > 3. Memory complexity analysis needs to be more comprehensive beyond just a single example.
> > > 4. Renaming "Ablative Study" doesn't address the need to isolate component contributions properly.
> > > 5. The theoretical foundation linking pre-training objectives to effective dynamics-enriched embeddings remains weak.
> > >
> > > While you've improved documentation and added experiments, these fundamental issues affect the paper's potential impact on the field.

---

> > > > ### Author Response · Authors · 2024-12-04
> > > > **Response to Reviewer bAXr**
> > > >
> > > > Thank you so much for your professional and valuable comments.
> > > >
> > > >  ### **Q1. About the extremely large MRAE.**
> > > > R1. In the past few days, we carefully compared the source codes and experimental settings between our method and baselines which adopt T-Drive and other realistic datasets as benchmark. We found that the extremely large MRAE is caused by the calculation mode on the missing values (presented as 0 in the dataset). For example, more than 20% data points are missed and presented as 0 in T-Drive, CHIBike and NYCTaxi. The original baseline methods masked the missing targets and therefore perform non-abnormal results. While we didn’t mask these missing values and calculate MRAE by mean($\frac{|\hat{y}-y|}{|y+1e-8|}$). Therefore, the existence of zero targets leads to rather large results. To sovle this, following STGODE, we masked the missing values and re-computed the MRAE results. Due to time limitation, we present the detailed results of our model variants on T-Drive, CHIBike and NYCTaxi below.
> > > >
> > > > |  | PDEDER | PDEDER-nopre | PDEDER-frz |
> > > > |---------|--------|--------------|------------|
> > > > | TDrive  | 0.395  | 0.341        | 0.425      |
> > > > | CHIBike | 0.744  | 0.712        | 0.702      |
> > > > | NYCTaxi | 0.401  | 0.364        | 0.428      |
> > > > ***
> > > >
> > > >  ### **Q2. About the data projection module.**
> > > > We kindly argue that the data projection module mainly acts as a prefix to align the data dimension across different systems. In fact, the patched tokens could be projected into any dimensions by any propoer layers. Here we choose the simple linear layer  to project it into one token for generalizable downstream learning.
> > > > ***
> > > >
> > > >  ### **Q3. About the memory complexity.**
> > > > The memory complexity is denoted as $O(P_{dp}+P_c +P_e + sH+P_g + P_m\cdot(P_d+P_r))$, where $P_{dp}, P_c, P_e, P_d, P_g, P_r$ denote the model parameters number of the data projection module, the convolutional module, the encoder of PLM, the decoder of PLM, the GNN module and the reconstruction module, respectively. $P_m$ denotes the number of patches.
> > > > ***
> > > >
> > > >  ### **Q4. About the ablative study.**
> > > > We kindly argue that this part is not a standard ablative study in the strict sense and we aim to examine the effect of pre-training on downstream dynamics modeling. Therefore, we examine this by setting and comparing with the model variants of fine-tuning without pre-training and freezing the pre-trained encoder/decoder.
> > > > ***
> > > >
> > > >  ### **Q5. About theoretical foundation of the pre-training objectives.**
> > > > We kindly argue that we mainly concentrate on enhancing the forecasting capacity when learning dynamics in this early attempt. We will try to incorporate dynamics-specific objectives in our future work.
> > > > ***
> > > >
> > > > Thank you so much for your valuable comments which helps us to improve our paper. We will carefully revise our paper as you suggested in our future version!

---

### Official Review · Reviewer_XvrH · 2024-11-04

**Soundness:** 3
**Presentation:** 3
**Contribution:** 2
**Rating:** 5
**Confidence:** 4

**Summary:**

The authors propose a neural system, the Pre-trained Dynamic Encoder (PDEDER), which is pre-trained on observations from various dynamic processes that unfold on graphs. They evaluate this model on a variety of long- and short-term forecasting tasks, both within domains (in-domain) and across domains (cross-domain).

Definition of benchmark:
   -153 sets of observations from different dynamical systems: 122 synthetic datasets from 14 dynamical systems and 31 real-world datasets from 10 dynamical systems.

   - Each dynamical system has Ms sets of observations with different parameters. Observations are multivariate, capturing data for each node in the system across different time steps.

    - Dynamical systems include: Springs, Mutualistic interactions, Heat diffusion, various Fluid dynamics, Biology, Climate, and Traffic, with system sizes ranging from 5 to 1024 nodes, timestamps from 100 to 28,000 steps, dimensionality from 1 to 10, sample counts from 1 to 10,000, and varying hyperparameters (from 1 to 15) for each dynamical system.

    - Each set of observations is temporally divided into in-sample and out-of-sample portions. PDEDER is trained to reconstruct in-sample data and forecast out-of-sample data.

Tokenization: to handle observations from various lengths, sub-observations are created to the fixed patch length and number of patches with specific stride length R. Additionally, Gaussian noise and instance normalization is done on each patch.

Authors use system-specific linear projection layer.

Learning Process: The projected data is used to reconstruct input states and to perform forecasting using a pre-trained language model (T5 model). The model architecture includes a convolutional layer for encoding, a PLM encoder, and a PLM decoder with additional linear adapters to aid in reconstruction and prediction. The loss function is an L1 loss applied to both the reconstruction and prediction components.

To model the evolution of dynamics, the authors employ a Graph Neural Network (GNN) with a single-layer normalized Laplacian and a trainable linear layer to encode the infinitesimal changes in the system state. Integration is then applied to derive the evolution of the hidden state, which is decoded by the decoder component of PDEDER.

They use 4 baselines:  NDCN Zang & Wang (2020a), ST-GODE Fang et al. (2021), MT-GODE Jin et al.
(2022) and TANGO Huang et al. (2024b).

Authors show results for short term/long term forecasting, in-domain and cross-domain setting.

As an active researcher in this field, it is challenging to assess the validity of the model without visual representations of the dataset’s dynamics. Showing figures that capture these dynamics would greatly improve the clarity and reliability of the experimental evaluation.

**Strengths:**

1. Work with large number of dynamical systems
2. Single model (modulo projection layer) that aims at reconstructing and forecasting dynamical systems is very hard.

**Weaknesses:**

Experimental setting is obscure and not written clearly (see and address questions for baselines, evaluation metrics, visualization of time-series forecasts vs ground truth).

Evaluations metrics:
Error Metrics for Dynamical Systems: Metrics such as MSE and MAE may not fully capture the performance of models on dynamical systems, as they might obscure certain dynamics-specific behaviors. Baselines should be better anchored to the dynamics with a simple, interpretable baseline for comparison. Try to include mean relative absolute error  Mean[ |(y_hat - y)/y| ] . Add another simple baseline for dynamics: e.g. prediction is last value plus numerical estimate of derivative plus some time series baselines.

Baselines used are focused on GNN-type models for dynamical systems.

Why Language model like T5 should be used for dynamical systems? If I am right, you have re-used language model? Provide more intuition why do you believe it has valid grounds e.g. what kind of biases for transfer learning do you see in this pre-trained model?

**Questions:**

1. Forecasting Task Visualizations: It would be helpful to include figures illustrating the in-domain and cross-domain forecasting tasks. Specifically, showing a time series up to a certain point in time, followed by model forecasts alongside the ground truth values, would provide valuable insights.

    Limitations of Tables for Dynamical System Predictions: Tables alone do not clarify which dynamical regimes are being predicted. There is a possibility that only simple, easily predictable regimes are being tested. Visuals displaying different regimes in the time series would help clarify the difficulty of the forecasting tasks being evaluated.

2. Use of Benchmark Models for Forecasting:

    Incorporating Established Forecasting Models: It would strengthen the analysis to include well-known models for time-series forecasting in the experiments, such as SOTA models from recent M-competitions (e.g., Smyl, Slawek, Grzegorz Dudek, and Paweł Pełka. "ES-dRNN: a hybrid exponential smoothing and dilated recurrent neural network model for short-term load forecasting." IEEE transactions on neural networks and learning systems (2023)). These models could serve as benchmarks for comparison, adding credibility to the experimental findings.

3. Novel Initial Value Conditions in Forecasting Tasks:

    Generalizability Across Initial Conditions: The current experimental setup does not appear to include tests on dynamics with novel initial conditions? This raises concerns about the model's reliance on specific initial values. Evaluating the model’s performance on dynamics with varied initial conditions would clarify whether it is generalizable or inherently tied to specific initial states.

4. Discuss potential advantages or relevant biases from language models that may transfer well to dynamical systems modeling.

---

> ### Author Response · Authors · 2024-11-24
> **Response to Reviewer #3 (XvrH) (1/5)**
>
> ### **Q1. Adding forecasting visualizations, new baseline methods and the evaluation metric MRAE.**
>
> Following your suggestion, we added these empirical studies in our latest version. Details are presented below and in our updated PDF version. We kindly argue that the the seasonal characteristics considered in ES-dRNN are not available on the dynamics systems we adopted and we will try to consider this characteristics in our future research.
>
> **results of MRAE:**
>
> | **System**      |        | **GNS**      | **NDCN**     | **STGODE**   | **MTGODE**   | **PDEDER**   | **PDEDER-nopre**   | **PDEDER-frz**   | **PDEDER-sys**   |
> | --------------- | ------ | ------------ | ------------ | ------------ | ------------ | ------------ | ------------------ | ---------------- | ---------------- |
> | Mutualistic     | 10%    | 2.840        | 1.031        | 2.875        | 1.297        | 5.702        | 6.379              | 5.949            | 6.136            |
> |                 | 20%    | 4.281        | 2.402        | 2.584        | 1.302        | 5.820        | 6.491              | 6.075            | 6.208            |
> |                 | 50%    | 3.221        | 5.202        | 1.985        | 1.118        | 3.675        | 4.140              | 3.896            | 3.947            |
> |                 | 70%    | 2.599        | 5.361        | 1.690        | 1.082        | 2.879        | 3.256              | 3.040            | 3.097            |
> |                 | 80%    | 2.405        | 5.391        | 1.598        | 1.071        | 2.625        | 2.972              | 2.762            | 2.824            |
> |                 | 100%   | 2.134        | 6.017        | 1.470        | 1.058        | 2.259        | 2.564              | 2.359            | 2.430            |
> | Heat            | 10%    | 2.613        | 0.542        | 3.114        | 1.847        | 0.307        | 0.910              | 0.782            | 0.639            |
> |                 | 20%    | 3.192        | 0.478        | 4.511        | 2.088        | 0.264        | 0.825              | 0.667            | 0.593            |
> |                 | 50%    | 9.268        | 0.920        | 8.326        | 8.384        | 0.885        | 3.643              | 1.438            | 1.298            |
> |                 | 70%    | 19.715       | 1.418        | 15.480       | 9.091        | 3.719        | 12.385             | 3.055            | 3.448            |
> |                 | 80%    | 22.421       | 1.644        | 17.535       | 9.893        | 4.536        | 13.063             | 3.975            | 4.136            |
> |                 | 100%   | 26.257       | 2.008        | 19.346       | 11.070       | 6.581        | 14.490             | 7.144            | 6.230            |
> | 2D_CFD          | 10%    | 8.452        | 28.476       | 1.544        | 12.435       | 1.158        | 1.170              | 1.196            | 1.069            |
> |                 | 20%    | 8.384        | 34.153       | 1.492        | 12.285       | 1.212        | 1.221              | 1.249            | 1.122            |
> |                 | 50%    | 11.434       | 39.077       | 1.671        | 13.364       | 1.539        | 1.704              | 1.734            | 1.389            |
> |                 | 70%    | 12.306       | 37.658       | 1.697        | 12.581       | 1.715        | 1.878              | 1.872            | 1.590            |
> |                 | 80%    | 12.745       | 38.380       | 1.727        | 14.436       | 1.861        | 2.044              | 2.003            | 1.731            |
> |                 | 100%   | 14.753       | 41.295       | 1.717        | 16.217       | 2.118        | 2.325              | 2.248            | 1.993            |
> | DarcyFlow       | 10%    | 21.489       | 1.069        | 10.887       | 21.086       | 1.404        | 3.679              | 2.333            | 1.376            |
> |                 | 20%    | 20.680       | 1.084        | 9.532        | 26.700       | 1.392        | 3.131              | 2.316            | 1.401            |
> |                 | 50%    | 21.351       | 1.169        | 7.970        | 28.617       | 1.486        | 2.156              | 2.479            | 1.476            |
> |                 | 70%    | 20.978       | 1.308        | 7.462        | 28.951       | 1.468        | 2.059              | 2.538            | 1.445            |
> |                 | 80%    | 21.000       | 1.452        | 7.384        | 29.882       | 1.449        | 2.079              | 2.549            | 1.428            |
> |                 | 100%   | 20.839       | 11.379       | 7.100        | 28.746       | 1.426        | 2.227              | 2.594            | 1.405            |

---

> ### Author Response · Authors · 2024-11-24
> **Response to Reviewer #3 (XvrH) (2/5)**
>
> | **System**      |        | **GNS**      | **NDCN**     | **STGODE**   | **MTGODE**   | **PDEDER**   | **PDEDER-nopre**   | **PDEDER-frz**   | **PDEDER-sys**   |
> | --------------- | ------ | ------------ | ------------ | ------------ | ------------ | ------------ | ------------------ | ---------------- | ---------------- |
> | Gene            | 10%    | 1.854        | 0.645        | 2.838        | 0.974        | 1.528        | 1.790              | 2.342            | 1.652            |
> |                 | 20%    | 1.984        | 0.870        | 3.010        | 1.038        | 1.499        | 1.704              | 2.409            | 1.590            |
> |                 | 50%    | 2.199        | 2.048        | 2.332        | 1.567        | 1.796        | 1.817              | 3.461            | 1.784            |
> |                 | 70%    | 2.136        | 2.897        | 2.207        | 1.451        | 1.928        | 1.873              | 3.963            | 1.891            |
> |                 | 80%    | 2.067        | 3.346        | 2.252        | 1.406        | 1.985        | 1.894              | 4.092            | 1.928            |
> |                 | 100%   | 2.111        | 3.966        | 2.282        | 1.357        | 2.268        | 2.066              | 4.638            | 2.243            |
> | ShallowWater    | 10%    | 1.057        | 0.804        | 0.893        | 1.306        | 1.611        | 0.732              | 0.923            | 0.965            |
> |                 | 20%    | 1.049        | 1.741        | 1.232        | 1.204        | 2.183        | 1.450              | 1.565            | 1.643            |
> |                 | 50%    | 1.023        | 1.303        | 1.021        | 1.132        | 1.930        | 1.071              | 1.406            | 1.406            |
> |                 | 70%    | 1.019        | 1.334        | 1.033        | 1.125        | 1.938        | 1.065              | 1.421            | 1.409            |
> |                 | 80%    | 1.017        | 1.273        | 1.008        | 1.120        | 1.866        | 1.038              | 1.380            | 1.364            |
> |                 | 100%   | 1.015        | 1.355        | 1.122        | 1.158        | 2.106        | 1.315              | 1.620            | 1.612            |
> | 2D_DiffReac     | 10%    | 10.476       | 24.661       | 1.046        | 5.338        | 4.918        | 5.565              | 4.877            | 5.508            |
> |                 | 20%    | 6.732        | 15.398       | 1.080        | 3.408        | 4.965        | 4.966              | 4.391            | 5.028            |
> |                 | 50%    | 5.386        | 11.152       | 1.108        | 2.744        | 4.292        | 3.619              | 3.440            | 3.964            |
> |                 | 70%    | 5.156        | 11.343       | 1.146        | 2.576        | 4.123        | 3.628              | 3.468            | 3.819            |
> |                 | 80%    | 4.989        | 11.107       | 1.135        | 2.433        | 3.984        | 3.463              | 3.355            | 3.659            |
> |                 | 100%   | 4.775        | 10.366       | 1.128        | 2.387        | 3.751        | 3.381              | 3.224            | 3.548            |
> | LA              | 10%    | 2.552        | 3.696        | 3.387        | 2.873        | 2.405        | 2.787              | 2.390            | 2.503            |
> |                 | 20%    | 2.574        | 3.408        | 3.252        | 2.126        | 2.245        | 2.655              | 2.240            | 2.358            |
> |                 | 50%    | 2.625        | 3.405        | 3.133        | 1.780        | 2.087        | 2.456              | 2.098            | 2.191            |
> |                 | 70%    | 2.666        | 3.338        | 3.092        | 1.719        | 2.039        | 2.380              | 2.046            | 2.134            |
> |                 | 80%    | 2.793        | 3.234        | 3.007        | 1.718        | 2.019        | 2.340              | 2.027            | 2.103            |
> |                 | 100%   | 2.700        | 3.090        | 2.907        | 1.707        | 1.988        | 2.293              | 1.990            | 2.067            |

---

> ### Author Response · Authors · 2024-11-24
> **Response to Reviewer #3 (XvrH) (3/5)**
>
> | **System**      |        | **GNS**      | **NDCN**     | **STGODE**   | **MTGODE**   | **PDEDER**   | **PDEDER-nopre**   | **PDEDER-frz**   | **PDEDER-sys**   |
> | --------------- | ------ | ------------ | ------------ | ------------ | ------------ | ------------ | ------------------ | ---------------- | ---------------- |
> | SD              | 10%    | 4.052        | 5.532        | 3.838        | 2.285        | 2.958        | 3.099              | 3.018            | 3.029            |
> |                 | 20%    | 3.489        | 6.573        | 3.526        | 2.388        | 4.927        | 5.086              | 4.996            | 4.941            |
> |                 | 50%    | 3.391        | 7.190        | 3.228        | 2.113        | 4.018        | 4.023              | 4.043            | 4.016            |
> |                 | 70%    | 3.241        | 7.637        | 3.120        | 1.976        | 3.749        | 3.723              | 3.780            | 3.723            |
> |                 | 80%    | 3.225        | 7.557        | 3.084        | 2.014        | 3.677        | 3.623              | 3.712            | 3.643            |
> |                 | 100%   | 3.282        | 7.520        | 3.267        | 1.929        | 3.613        | 3.562              | 3.683            | 3.581            |
> | NYCTaxi         | 24     | 51.990       | 72.023       | 69.029       | 30.944       | 112.286      | 113.806            | 83.433           | 114.767          |
> |                 | 48     | 36.132       | 41.951       | 40.947       | 17.091       | 59.510       | 60.461             | 44.679           | 60.569           |
> |                 | 96     | 43.938       | 51.430       | 43.910       | 13.353       | 58.008       | 60.900             | 47.414           | 57.876           |
> |                 | 192    | 53.038       | 58.848       | 71.913       | 10.608       | 50.885       | 53.787             | 44.304           | 50.721           |
> |                 | 336    | 62.130       | 69.981       | 70.197       | 11.105       | 49.687       | 52.057             | 46.323           | 51.752           |
> |                 | 720    | 62.789       | 63.802       | 69.971       | 11.730       | 56.289       | 56.623             | 52.163           | 58.768           |
> | CHIBike         | 24     | 15.736       | 11.287       | 20.722       | 13.046       | 17.738       | 23.882             | 19.476           | 19.628           |
> |                 | 48     | 49.071       | 54.795       | 23.159       | 7.855        | 18.073       | 23.543             | 18.030           | 19.136           |
> |                 | 96     | 39.272       | 40.558       | 62.846       | 14.586       | 45.471       | 52.778             | 43.365           | 42.505           |
> |                 | 192    | 53.351       | 54.833       | 76.555       | 15.592       | 57.547       | 67.644             | 56.502           | 53.260           |
> |                 | 336    | 60.742       | 72.382       | 98.041       | 13.229       | 61.071       | 69.650             | 60.060           | 56.850           |
> |                 | 720    | 89.368       | 110.547      | 136.728      | 17.090       | 86.629       | 114.281            | 84.584           | 97.035           |
> | Tdrive          | 24     | 7574.2     | 15736.0   | 27129.3    | 10453.3    | 15286.5    | 19079.9          | 14473.9        | 15618.7        |
> |                 | 48     | 12647.7    | 16404.9    | 30402.6    | 9791.7     | 16292.3    | 20283.7          | 15643.0        | 16655.2        |
> |                 | 96     | 15117.7    | 16976.4    | 28853.0    | 10139.2    | 17015.3    | 20387.7          | 16497.6        | 17389.7        |
> |                 | 192    | 14503.9    | 18483.8    | 26906.7    | 8501.7     | 16026.1    | 18704.0          | 15550.5        | 16384.3        |
> |                 | 336    | 14032.6    | 18978.9    | 23672.2    | 6640.1     | 14970.5    | 17032.2          | 14556.4        | 15269.7        |
> |                 | 720    | 13592.2    | 19281.9    | 17588.1    | 4550.2     | 14584.5    | 16234.7          | 14261.4        | 14959.7        |

---

> > ### Author Response · Authors · 2024-11-24
> > **Response to Reviewer #3 (XvrH) (4/5)**
> >
> > | **System**      |        | **GNS**      | **NDCN**     | **STGODE**   | **MTGODE**   | **PDEDER**   | **PDEDER-nopre**   | **PDEDER-frz**   | **PDEDER-sys**   |
> > | --------------- | ------ | ------------ | ------------ | ------------ | ------------ | ------------ | ------------------ | ---------------- | ---------------- |
> > | PEMS03          | 24     | 8.454        | 2.772        | 6.150        | 2.181        | 3.235        | 3.684              | 3.441            | 3.560            |
> > |                 | 48     | 7.937        | 6.223        | 7.615        | 3.276        | 5.120        | 5.774              | 5.535            | 5.698            |
> > |                 | 96     | 7.184        | 5.665        | 6.767        | 2.787        | 6.089        | 6.842              | 6.283            | 6.698            |
> > |                 | 192    | 7.096        | 6.167        | 6.655        | 3.075        | 8.279        | 9.395              | 8.364            | 8.922            |
> > |                 | 336    | 6.993        | 5.582        | 7.959        | 2.689        | 7.562        | 8.728              | 7.494            | 8.140            |
> > |                 | 720    | 8.358        | 5.651        | 7.027        | 2.469        | 7.290        | 8.446              | 7.427            | 7.726            |
> > | PEMS04          | 24     | 4.049        | 6.577        | 4.554        | 2.481        | 4.007        | 4.484              | 3.975            | 3.946            |
> > |                 | 48     | 3.813        | 8.405        | 6.532        | 2.563        | 5.163        | 5.758              | 5.173            | 5.165            |
> > |                 | 96     | 4.201        | 11.508       | 5.515        | 2.444        | 6.158        | 6.948              | 6.221            | 6.177            |
> > |                 | 192    | 4.092        | 12.697       | 5.551        | 2.384        | 6.079        | 6.871              | 6.235            | 6.113            |
> > |                 | 336    | 4.168        | 14.319       | 5.520        | 2.596        | 5.964        | 6.748              | 6.105            | 6.034            |
> > |                 | 720    | 4.317        | 17.939       | 5.688        | 2.508        | 5.916        | 6.630              | 5.899            | 6.094            |
> > | PEMS07          | 24     | 4.080        | 6.308        | 3.342        | 1.575        | 4.472        | 5.075              | 4.553            | 4.123            |
> > |                 | 48     | 4.099        | 6.438        | 3.320        | 1.551        | 5.353        | 6.265              | 5.601            | 5.205            |
> > |                 | 96     | 4.455        | 7.255        | 3.070        | 1.663        | 6.687        | 7.996              | 7.013            | 6.815            |
> > |                 | 192    | 4.471        | 7.332        | 3.439        | 1.796        | 6.500        | 7.721              | 6.764            | 6.721            |
> > |                 | 336    | 4.663        | 7.349        | 3.359        | 1.817        | 5.771        | 6.809              | 5.993            | 6.063            |
> > |                 | 720    | 4.788        | 7.609        | 3.259        | 1.767        | 6.644        | 7.525              | 6.416            | 7.261            |
> > | PEMS08          | 24     | 2.788        | 8.452        | 3.725        | 3.145        | 8.177        | 9.626              | 8.323            | 8.631            |
> > |                 | 48     | 3.150        | 10.270       | 3.553        | 2.839        | 7.441        | 8.647              | 7.589            | 7.731            |
> > |                 | 96     | 3.638        | 11.651       | 4.006        | 2.619        | 7.716        | 9.096              | 7.804            | 7.954            |
> > |                 | 192    | 3.887        | 14.057       | 4.162        | 2.768        | 8.775        | 10.622             | 8.884            | 8.975            |
> > |                 | 336    | 3.784        | 13.029       | 4.040        | 2.721        | 8.102        | 9.663              | 8.191            | 8.254            |
> > |                 | 720    | 3.977        | 12.479       | 4.507        | 3.111        | 8.050        | 9.084              | 8.109            | 8.036            |

---

> ### Author Response · Authors · 2024-11-24
> **Response to Reviewer #3 (XvrH) (5/5)**
>
> | **System**      |        | **GNS**      | **NDCN**     | **STGODE**   | **MTGODE**   | **PDEDER**   | **PDEDER-nopre**   | **PDEDER-frz**   | **PDEDER-sys**   |
> | --------------- | ------ | ------------ | ------------ | ------------ | ------------ | ------------ | ------------------ | ---------------- | ---------------- |
> | NOAA            | 24     | 17.798       | 8.817        | 3.129        | 5.074        | 17.031       | 15.150             | 19.176           | 17.995           |
> |                 | 48     | 20.415       | 13.585       | 2.682        | 6.263        | 21.904       | 17.113             | 21.112           | 22.900           |
> |                 | 96     | 22.065       | 15.519       | 2.853        | 5.623        | 26.829       | 22.450             | 23.307           | 27.712           |
> |                 | 192    | 21.805       | 15.616       | 4.433        | 6.472        | 24.321       | 20.795             | 22.598           | 24.067           |
> |                 | 336    | 21.309       | 14.823       | 3.735        | 6.714        | 22.113       | 17.663             | 20.318           | 21.997           |
> |                 | 720    | 21.129       | 13.466       | 3.760        | 6.312        | 16.534       | 13.298             | 17.635           | 15.774           |
>
> ***
>
> ### **Q2. About the novel initial value conditions when forecasting.**
>
> We are grateful for being pointed out this missing issue. In practice, we generate $M_s$ set of observations with random initial values when generating observations for each parameter setting. We missed this important issue and modified the corresponding paragraphs in our latest version (see Benchmark Generation, pre-training objective function Eq.3 and Table 1 of benchmark statistics).
>
> ***
>
> ### **Q3. Potential advantages or biases from LMs to dynamics modeling.**
> One of the advantages on transferring PLM to dynamics is that we can utilize the sequence forecasting capacity of transformer. Analogous to the usage of pre-trained language models, our pre-trained \baby concentrates on how to learn better dynamics-enriched representations. And the dynamics modeling module can be analogous to the classification head or regression head when fine-tuning a language model for downstream tasks.  After pre-training the embedder, we can learn generalizable embeddings for observations for any dynamics system, and these embeddings could be used for approximating dynamics by fine-tuning with any specific dynamics modeling methods. In this way, our embedder pre-training process could benefit approximating dynamics models more easily.
>
> The variety and distinctiveness of systems ensure the capability of learning generalizable dynamics-enriched embeddings. When collecting benchmarks, we set various hyper-parameters, including system-specific parameters, number of objects and sequence lengths to generate various distinct dynamics observations, which can ensure the diversity of dynamics characteristics. The random initialization of initial states also devotes to the generalizability of learnt embeddings.
> ***
> We hope to hear back from you if you have further questions.

---

> > ### Comment · Reviewer_XvrH · 2024-11-26
> > **reviewer response to author's response**
> >
> > I would like to thank the authors for hard work.
> > By adding MARE, they have tried to address my concern on not appropriate metrics for dynamical systems.
> > If one inspects all the results, two possible conclusions can be derived: (i) now in majority of settings their method is not showing best performance. It is not necessary bad, if one has a contribution that improves understanding of problem. (ii) MARE can be very high even few orders of magnitude larger than 100% relative error. Which brings me to 2nd problem of authors not understanding how bad the forecasts for some dynamics really are. Which implies that the MSE, MAE were only showing an illusion of good performance.
> > If one would look at the time-series visualizations of trajectories ground truth vs forecast, one would see the problem of super large MARE directly. e.g. in table 9 MSE=0.116, MAE=0.168  but MARE is 15286.5.
> > Reason why the absolute errors are small is the scale issues.
> >
> > Visualization of trajectories e.g. fig 2, page 19 is done in non professional way for serious publication (fonts size, values not readable, not writing what is on axis).
> >
> > Authors also do not really test the model on different initial conditions. When one thinks about real-world applications, this becomes a problem. Which again is not necessary a problem, if the paper would improve our knowledge on modelling dynamics with neural systems. But this paper tries to do too many things at the same time, without paying enough attention to all the details, and that is why for me, I can not increase my score. Overall, interesting research, but not well done, it needs few more rounds of polishing, and critical view on the main contributions.

---

> > > ### Author Response · Authors · 2024-12-04
> > > **Response to Reviewer XvrH**
> > >
> > > Thank you so much for your professional and valuable comments.
> > >
> > >  ### **Q1. About the extremely large MRAE.**
> > > In the past few days, we carefully compared the source codes and experimental settings between our method and baselines which adopt T-Drive and other realistic datasets as benchmark. We found that the extremely large MRAE on these datasets is caused by the calculation mode on the missing values (presented as 0 in the dataset). For example, more than 20% data points are missed and presented as 0 in T-Drive, CHIBike and NYCTaxi. The original baseline methods masked the missing targets and therefore perform non-abnormal results. While we didn’t mask these missing values and calculate MRAE by mean($\frac{|\hat{y}-y|}{|y+1e-8|}$). Therefore, the existence of zero targets leads to rather large results. To sovle this, following STGODE, we masked the missing values and re-computed the MRAE results. Due to time limitation, we present the detailed results of our variants on T-Drive, CHIBike and NYCTaxi below.
> > >
> > > |  | PDEDER | PDEDER-nopre | PDEDER-frz |
> > > |---------|--------|--------------|------------|
> > > | TDrive  | 0.395  | 0.341        | 0.425      |
> > > | CHIBike | 0.744  | 0.712        | 0.702      |
> > > | NYCTaxi | 0.401  | 0.364        | 0.428      |
> > >
> > >
> > > ***
> > >
> > >  ### **Q2. About the visualization figures.**
> > > We re-draw the visualization of trajectories in our PDF as you suggested. Due to the time limitation, we will present more visualizations in our future version on more systems.
> > > ***
> > >
> > >  ### **Q3. About testing on different initial conditions.**
> > > We are sorry that we didn’t express how we solve this clearly in our latest response. Actually, we examined this problem in every task. The results presented in all tables are the averaged results of multiple trajectories with different initial values. When generating trajectories for each system, we generate $M_s$ samples with different randomly initialized values. During fine-tuning, we use all $M_s$ samples to fine-tune one model and evaluate for each system and we report the averaged results in our paper.
> > > ***
> > >
> > > Finally, we will work hard to revise our paper as you suggested and thank you so much for your professional comments which help us a lot to improve our paper!

---

### Official Review · Reviewer_3iEp · 2024-11-04

**Soundness:** 2
**Presentation:** 3
**Contribution:** 3
**Rating:** 5
**Confidence:** 3

**Summary:**

The paper introduces a method for leveraging observations from multiple systems to create a generalized model that captures shared dynamics across these systems in a common latent space. Their method, PDEder, builds on pre-trained language models, adapted to specific dynamic observations through tokenization and fine-tuning, which enables predictions of future dynamics. The authors evaluate their model on 18 dynamical systems, covering both long- and short-term forecasts.

**Strengths:**

The paper is well-written and tackles a significant challenge in modeling time series obtained from multiple systems. It leverages recent advances in language models and emphasizes generalizability, which is an important quality for such models.

**Weaknesses:**

# Critical:
1) I am concerned about the main assumptions the paper relies on. Are you assuming that different systems come from the same statistics or share fundamental dynamics? I would argue that systems from completely different worlds, time scales, and dynamical regimes should not necessarily be trained together when the overlap in their behaviors is minimal. There is an assumption the authors rely on that needs better quantification regarding how much the systems can differ in terms of dynamical regimes and time scales. Even within the same field and data modalities, recordings from different subjects often differ significantly, indicating that they should not always be learned together. Across fields, I am concerned this issue is more pronounced and must be addressed more carefully both mathematically and as a discussion with a clear list of assumptions.

2) Regarding the previous point, in Line 044 you mention generalizability across domains. While models should indeed, broadly. be generalizable, they must also capture the unique characteristics of different domains. Therefore, I believe the trade-off between generalizability, expressivity, and interpretability needs to be more thoroughly addressed.

3) Additionally, more papers should be discussed in the related work, including works on neural dynamics that leverage multi-session information via shared dynamical priors [1] or transformers [2] for inferring dynamics.

4) I am also concerned about the interpretability of the model, which you barely discussed. When using deep learning for scientific purposes, we want to ensure that our model parameters and latent variables are interpretable. However, with the use of pre-trained language models and fine-tuning across observations, it seems the model lacks interpretability. How would the authors address this?

5) It is not clear how you calculate the graphs (i.e., $\mathcal{G}$) for the real-world data. Is it known, or do you calculate it during pre-processing?

6) It is unclear how robust the system is to hyperparameter choices. Please discuss or explore this.

7) Additionally, it is unclear how you train/fit Eq. 3 in practice. Is it via EM or global optimizers? Please include an algorithm.

8) Many results are presented in the supplementary materials. While I understand that the page limit sometimes necessitates this approach, the authors did not use the full page limit in this case. Why not include more results in the main text? I suggest incorporating additional results (e.g., from Appendix B) into the main text.

9) I think an important question that needs discussion is how many observations you need for the method to succeed. I would assume that performance will improve rapidly with a low number of observations and then level off as more are added. Do you have any quantification of that?





# Minor:

1) In the abstract, you used "neural Ordinary Differential…" without capitalizing "N"; however, in the introduction, it’s capitalized. Please be consistent.

2) Line 052: Change "where the dynamics can be easier captured" to "where the dynamics can be more easily captured."

3) While the paper is well-written overall, the first paragraph of the related work section reads more like a list than a motivation to identify the research gap. I suggest rephrasing it to better highlight the gap and clarify what your method aims to address.

4) Line 147: There’s a missing period after the ODE subtitle.

5) Line 192: The fourth word (`We`) should not be capitalized.

6) Lines 209-211: This content seems out of place and might fit better in the related work section.

7) Line 247: Did you mean "serve" instead of "sever"?

8) Line 370: The last word, "we", should be capitalized.

9) Table 5: Why is the "%" sign only next to some numbers? Are all values percentages, or do those with "%" represent a different scale (1/100)? This needs clarification.


**References:**

[1] Mudrik, N., Ly, R., Ruebel, O., & Charles, A. S. (2024). Creimbo: Cross ensemble interactions in multi-view brain observations. arXiv preprint arXiv:2405.17395.

‏
[2]  Liu, R., Azabou, M., Dabagia, M., Xiao, J., & Dyer, E. (2022). Seeing the forest and the tree: Building representations of both individual and collective dynamics with transformers. Advances in neural information processing systems, 35, 2377-2391.‏

**Questions:**

1. What happens if some observations are from a different distribution than the others?

2. What is the level of similarity you assume across systems? Please clarify the assumptions regarding the level of similarity required across different systems for your model to be effective. Are you suggesting that systems must share certain statistical properties or fundamental dynamics?

3. What is the model's computational complexity, and how does it scale with the number of observations?

4. How do you choose the system-specific parameters?

5. Can you clarify the dimension of $\tilde{x}^\text{in}$? Is it equal to 1? Additionally, please specify the dimension of $W^s_{dp}$ in the `Data Projection` section.

6. Why was the architecture explained in Eq. 2 chosen?

7. Why did you choose the $\ell_1$ rather than $\ell_2$ loss? (line 258).

---

> ### Author Response · Authors · 2024-11-24
> **Response to Reviewer #2 (3iEp) (1/3)**
>
> ### **Q1. About the assumption of sharing fundamental dynamics.**
> Thank you for this comment, however, there may be a misunderstanding. To explain this, we would review the full story of this work. Developing a generalized model, that can handle all dynamics, or at least many, can be a fundamental research goal for dynamics modeling [1-2]. Learning fundamental dynamics or hidden characteristics for various dynamics is a rather basic and tough problem. Therefore we adopt an easier and lighter way which first learns better representations for dynamics observations, and these representations can then be adopted to easily learn the hidden dynamics.
>
> Therefore, we kindly argue that our main contribution lies in pre-training to learn better representations for downstream dynamics learning. Analogous to the usage of pre-trained language models, our pre-trained \baby concentrates on how to learn better dynamics-enriched representations. And the dynamics modeling module can be analogous to the classification head or regression head when fine-tuning a language model for downstream tasks. We want to highlight that, in the pre-training period, no system-specific dynamics are approximated, and the pre-training process only learn better embeddings for the observed sequences. And the interacting graphs are not considered in pre-training.
>
> In this way, the generalizability of our model lies in the representation level, rather than the dynamics learning level. After pre-training the embedder, we can learn generalizable embeddings for observations for any dynamics system, and these embeddings could be used for approximating dynamics by fine-tuning with any specific dynamics modeling methods. In this way, our embedder pre-training process could benefit approximating dynamics models more easily.
>
> [1] Lomax, H., Pulliam, T. H., Zingg, D. W., and Kowalewski, T. A. (2002). Fundamentals of computational fluid dynamics. Appl. Mech. Rev., 55(4), B61-B61.
>
> [2] Luenberger D. Dynamic equations in descriptor form[J]. IEEE Transactions on Automatic Control, 1977, 22(3): 312-321.
>
> ***
>
> ### **Q2. About the generalizability and interpretability.**
>
> To express the interpretability of our proposal, we adopt a white-box dynamics learner SINDy to learn dynamics on the observation embeddings. Details are presented in the 2rd response in general comment.
>
>
> |              | PDEDER|            |           |            | PDEDER+PDEDER+SINDy|            |           |           |
> |--------------|------------|------------|-----------|------------|--------------|------------|-----------|-----------|
> |              | short-term |            | long-term |            | short-term   |            | long-term |           |
> |              | MSE        | MAE        | MSE       | MAE        | MSE          | MAE        | MSE       | MAE       |
> | Mutualistic  | 0.362      | 0.452      | 0.809     | 0.675      | 1.014        | 1.014      | 0.334     | 0.334     |
> | Heat         | 0.003      | 0.045      | 0.006     | 0.052      | 0.886        | 0.884      | 1.577     | 1.586     |
> | 2D CFD       | 0.223      | 0.303      | 0.152     | 0.236      | 1.001        | 0.984      | 1.139     | 1.164     |
> | DarcyFlow    | 0.001      | 0.020      | 0.001     | 0.021      | 0.858        | 0.851      | 1.103     | 1.104     |
> | Gene         | 0.035      | 0.136      | 0.076     | 0.172      | 0.613        | 0.537      | 0.783     | 0.783     |
> | ShallowWater | 0.674      | 0.358      | 1.145     | 0.527      | 0.538        | 0.463      | 1.040     | 1.047     |
> | 2D DiffReac  | 0.960      | 0.723      | 1.057     | 0.794      | 0.126        | 0.126      | 0.807     | 0.808     |
>
> ***
>
> ### **Q3. How to calculate the graph $\mathcal{G}$ for real-world data.**
>
> \textbf{R3.} Thanks for pointing out these missing information. For LA, SD, PEMS03, PEMS04, PEMS07 and PEMS08, the corresponding graph structure are provided by the original datasets. As for NYCTaxi, CHIBike, TDrive and NOAA, we calculate the graph structure by distances of the provided latitude/longitude or grid coordinates of each observation station. We added these introductions into the Appendix A.

---

> > ### Author Response · Authors · 2024-11-24
> > **Response to Reviewer #2 (3iEp) (2/3)**
> >
> > ### **Q4. Robustness on the hyper-parameters.**
> > We discuss and explore the robustness from two aspects, model-related parameters and data-related hyper-parameters.
> >
> > As for model-related hyper-parameters, learning rates should not be too small comparing with directly fine-tuning a pre-trained language model on downstream tasks such as $1e-7$ or $1e-8$, etc. We finally chose $1e-3$ for pre-training. The most possible reason is that the optimal embeddings space for better learning dynamics differs from which in language modeling to some extent. In our early attempts, we indeed observed that pre-training with smaller learning rates performs quite awful results, leading to rather poor fine-tuning results.
> >
> > As for data-related hyper-parameters, the choices of patch length and stride are essential. In our early attempts, we found that the pre-training process are quite in-sensitive to these two parameters. Therefore, we chose moderate lengths following [5]. The fine-tuning process are also in-sensitive to them. For time limitation, we applied sensitivity studies on the fine-tuning process. Detailed results of sensitivity analysis are presented in our updated paper (see Fig.4 in Appendix). We will add more discussions on the sensitivity analysis in our future version.
> > ***
> >
> > ### **Q5. About how to train the objective in Eq.3. Please include an algorithm.**
> >
> > Thanks for pointing out this missing part, we added a section of model training and an algorithm to clarify the overall pre-training and fine-tuning processes. The additional introductions are listed below and we also added it in the PDF version (see Section 4.4 and Algorithm 1 and 2).
> >
> > >**Model Training** We first pre-train \baby on all collected dynamics observations (without graph) with Eq.3 for $E_p$ epochs. To handle the massive observations and various numbers of samples on different systems, we randomly choose $10$ dynamics systems for each training round and train \baby for $5$ epochs with all the observations from these systems. When learning a specific dynamics, we fine-tune \baby with Eq.6 for $E_f$. The training details are presented in Algorithm 1 and 2.
> > ***
> >
> > ### **Q6. About the full page limitation.**
> > Following your suggestion, we reformatted our paper to use the full page limit.
> >
> > ***
> >
> > ### **Q7. How many observations for the method to succeed.**
> > We kindly argue that the fine-tuning of our pre-trained PDEDER on all systems are essential for examining the effectiveness of the pre-training process. Specifically, we set several model variants for PDEDER, such as fine-tuning without our pre-training process or fine-tuning with freezing our pre-trained PLM parameters. In this way, we can explore the exact effect of pre-training on each of the system.
> >
> > ***
> >
> > ### **Q8. About the minor weaknesses.**
> >
> > Thanks for your careful comments. Following your suggestions, we corrected all the mistakes and checked our paper carefully to rectify mistakes.
> >
> > + Due to the time limitation, we will rewrite the whole passage of related works on dynamics modeling to make it clearer in our future version.
> >
> > + We rewrote the paragraph of data projection. Details are presented in general comment and response for Q12.
> >
> > + The missing period are caused by the latex format automatically. We will reformat the paper layout as you suggested.
> >
> > + \% denotes these numbers are in a different scale (1/100). We scale numbers which are rather small by 1/100. For example, the value of "0.067\%'' is "0.000673286''. We added the descriptions "\% denotes the results are scaled by 1/100.'' in the table captions in our updated version.
> > ***
> >
> > ### **Q9. Distribution differences of observations.**
> >
> > In both of the pre-training and fine-tuning progresses, we use an instance normalization layer to handle the distribution shifts following [6].
> > ***
> >
> > ### **Q10. How to choose the system-specific parameters.**
> > Following [7], we directly adopt the default parameters on systems which are cover by it. For other systems, we resort to their original papers and chose various parameters which are close to the default parameters.
> >
> > ***
> >
> > ### **Q11. The dimension of $\tilde{x}^{(in)}$.**
> >
> > The details about data projection including the dimension of $\tilde{x}^{(in)}$ have been presented in the general comment above and edited in our latest version.
> >
> > ***
> >
> > ### **Q12. The choice of architecture in Eq.2.**
> >
> > Apart from the basic encoder and decoder of a PLM, we adopt a convolutional layer for encoding and a flatten-and-linear layer for decoding following previous works [6,8].
> >
> >
> > ***
> > ### **Q13. The choice of objective function.**
> >
> > We chose $\ell_1$ loss following NDCN[9], one of the most representative neural dynamics modeling method. We also tried the $\ell_2$ MSE loss during empirical studying, and the results show little difference.
> >
> > ***

---

> ### Author Response · Authors · 2024-11-24
> **Response to Reviewer #2 (3iEp) (3/3)**
>
> ### **Q14. Computational complexity and scaling on number of observations.**
> The computational complexity consists of the several following parts: data projection, encoding by convolutional layer, fine-tuning with LM, decoding by linear layer and the integration approximation in Neural ODE. The detailed computational complexity is denoted as $O(M(s(N^2H+NH^2)+2NL_p^2V+3HL_pN-2HL_p+LH+LH^2))$, where $M$ denotes the number of observation sets; $N$ denotes the number of objects in one system; $L_p$ denotes the patch length; $V$ denotes the system-specific dimension; $H$ denotes the hidden dimension of PLM; $L$ denotes the number of layers in PLM; $s$ denotes the solving step in Neural ODE. We can find the complexity is linear with the number of observations sets; and quadratic to the number of objects, which is caused by the GNN layer in Neural ODE. In our practical studies, the runtime of our proposal is faster than baselines, which are introduced above (see Q5 in Response to Reviewer #1 (B9zD) (3/3)).
> ***
>
> [5] Zhou, Tian, et al. One fits all: Power general time series analysis by pretrained lm. Advances in neural information processing systems 2023, 36: 43322-43355.
>
> [6] Tian Zhou, Peisong Niu, Liang Sun, Rong Jin, et al. One fits all: Power general time series analysis by pretrained lm. Advances in neural information processing systems, 36:43322–43355, 2023.
>
> [7] Takamoto, Makoto, et al. Pdebench: An extensive benchmark for scientific machine learning. Advances in Neural Information Processing Systems 35 (2022): 1596-1611.
>
> [8] Ching Chang, Wen-Chih Peng, and Tien-Fu Chen. Llm4ts: Two-stage fine-tuning for time-series forecasting with pre-trained llms. arXiv preprint arXiv:2308.08469, 2023.
>
> [9] Chengxi Zang and Fei Wang. Neural dynamics on complex networks. In Proceedings of the 26th ACM SIGKDD International Conference on Knowledge Discovery \& Data Mining, pp. 892–902,
> 2020.
> ***
> We hope to hear back from you if you have further questions.

---

> > ### Comment · Reviewer_3iEp · 2024-11-29
> > **response to authors**
> >
> > Dear Authors,
> >
> > First, I apologize for my delayed response and thank you for your patience. I appreciate your comments and the additional changes you made to the PDF.
> >
> > 1) In your response, you wrote, ``that can handle all dynamics, or at least many, can be a fundamental research goal for dynamics modeling [1-2]``. This is exactly the issue I raised. the usage of ``all dynamics``, I assume your approach is tailored towards observations with noise that follows normal distribution? What are your assumptions about the statistics and the nature of `dynamics`? The phrase "all dynamics" is too vague. For instance, if the data comes from a Poisson distribution, will the model still work? Or if it includes high-frequency noise (e.g., LFP recordings)? It is important to clearly state the statistics and conditions of the dynamics you are focusing on. The references you cited address fluid and autonomic control dynamics, which have their own properties, but may not represent `all dynamics`.
> >
> > 2) Regarding interpretability, while applying SINDy to the embeddings is an interesting idea, I do not see how the table you provided teaches us about the interpretability or goes beyond error quantification. A key advantage of SINDy is its ability to decompose a system into basic functions or components—not just considering reconstruction accuracy as the primary metric. Could you analyze the SINDy weights and provide some interpretations? I understand that interpreting complex, multi-way data is difficult, but offering insights into the components is critical when applying this to scientific problems.
> >
> > 3) It seems like you also skipped my critical point 3. If your approach is not intended for neural dynamics, that makes sense, but it should be explicitly stated. Otherwise, please explain why neural dynamics or other works from the field are not relevant.
> >
> > 4) If the graphs are required or need to be calculated from labeled data, this should be stated explicitly in the text. Many real-world dynamical datasets lack pre-defined graphs or labels for graph construction, so this need should be acknowledged clearly.
> >
> > 5) On Q8, why not use standard scientific notation (e.g., (e^(-3))  ) then?
> >
> > 9) You also discussed the instance normalization layer. While it addresses certain issues, what happens if the data follows a Poisson distribution? Again, I do not expect a method to handle every type of data, but you should explicitly outline the statistical assumptions about the data. Hence this does not answer the question "What happens if some observations are from a different distribution than the others?".
> >
> > 10) I do not understand the response to "Q7. How many observations for the method to succeed.", or that is not what I asked. Assume you have only one dataset for pre-training, and then you apply it to a different dataset for fine-tuning. I assume the advantage of pre-training on the first dataset depends on how similar the two datasets are. As the number of datasets for pre-training increases, the pre-trained model should become more robust. There must be some assumption about the relationship between the number of datasets, their similarity, and how similar they are to the datasets you aim to apply them to later. What is the scale of the number of datasets for training needed to achieve a robust pre-trained model and how it changes with more or less datasets? even if you do not provide an experiment for that, some explanation of this effect is important I believe.
> >
> > 11)  while including L1 regularization in the cost function makes sense, it is important to explain  (also in the paper) why you chose it, as L1 promotes sparsity (unlike L2), which is probably the reason you observed different results under these penalties.

---

> > > ### Author Response · Authors · 2024-12-04
> > > **Response to Reviewer 3iEp**
> > >
> > > Thank you so much for your patient comment.
> > >
> > > ### **Q1&6. About the fundamental dynamics and dataset statistics assumptions.**
> > >
> > > We are sorry that we didn’t present our motivation clearly in our latest response. We kindly argue that we don’t want to learn a fundamental dynamics model to handle all dynamics, which is a rather tough and challenging problem. Therefore, we didn’t make assumptions on the observation statistics. Rather than explicitly learning the shared governing basic dynamics, we want to learn dynamics in an easier and lighter way, which learn generalizable embeddings to better approximate dynamics in downstream task.
> > > ***
> > >
> > > ### **Q2. About analyzing the SINDy weights.**
> > >
> > > Thanks for your advice. We will add this empirical study in our future version.
> > > ***
> > >
> > > ### **Q3.  About adding discussions.**
> > > We are sorry for missing this question. We will add more discussion about the related work as you suggested. Besides, we kindly argue that any dynamics modeling method could be appended to our proposed PDEDER for fine-tuning to learn certain dynamics, including both neural and shallow, discrete and continuous methods, etc.
> > > ***
> > >
> > > ### **Q4. About graph calculation.**
> > > Thanks for your advice. We will add specific details about how to calculate graph for each system. For all systems we adopted, the graphs are either provided by the original dataset, or calculated by a grid network of each object node.
> > > ***
> > >
> > > ### **Q5. About the formats of small values.**
> > > Thanks for your advice. We will reformat the results in the standard scientific notation in the next version.
> > > ***
> > >
> > > ### **Q7. About the pre-training datasets scale.**
> > > Thank you so much for your detailed comment. We are sorry for misunderstanding this question. In our early attempts, we designed several empirical studies to examine the effect of datasets scale used in pre-training. Due to time limitation, we only examined the effect of leaving one system and one parameter out. And the results show more pre-training datasets perform better and the two LOO versions also show competitive performance. We will try to explore more about this task in our future version.
> > > ***
> > >
> > > ### **Q8. About the chosen of objective function.**
> > > The L2 loss may be more sensitive to some outlier values comparing with L1 loss, and may lead to less penalty on those normal values, especially for the realistic systems, where the ground truth after de-normalization may be rather large.
> > > ***
> > > Thank you so much for you valueable comments which helps us a lot to improve our paper. We will revise our paper carefully as you suggested.

---

### Official Review · Reviewer_B9zD · 2024-11-11

**Soundness:** 3
**Presentation:** 3
**Contribution:** 3
**Rating:** 6
**Confidence:** 4

**Summary:**

This paper proposes a generalized framework to learn system dynamics across different settings, by utilizing a pretrained language model from massive observational data, and jointly fine-tuning the pretrained language model and a Graph ODE-based  neural simulator. The proposed PDEDER is pre-trained on 153 sets of observations from 24 complex systems, using a pre-trained language model updated via tokenization techniques. Experiments evaluate PDEDER on 18 dynamic systems for long/short-term forecasting in both in-domain and cross-domain settings.

**Strengths:**

1. The proposed generalized pre-trained dynamics encoder is well-motivated and technically sound.

2. The proposed approach achieves good in-domain and cross-domain performance, highlighting its generalization ability.

**Weaknesses:**

1. The writing needs further improvement. For example, the citation in the main text sometimes should be \citep (line 36-39 for example, with references within brackets) instead of \cite. Also in the problem setting section, can the authors justify if the graph structure (edges) are fixed or evolve over time?

2. I feel the experiments can be further improved: for the baselines, they are all neuralODE-based approaches. However for dynamical system modeling, there are also many discrete neural simulators [1]. The authors are suggested to justify why these approaches are not compared in this paper. As mentioned in the abstract part, the proposed framework should be easily coupled with any dynamic modeling methods (besides neural ODEs). Also, there are works [2] that learn a generalized neural simulator trained from multiple systems. It is also suggested to include them in the paper for a more comprehensive comparison.



[1] Learning to Simulate Complex Physics with Graph Networks.

[2] Generalizing Graph ODE for Learning Complex System Dynamics across Environments

**Questions:**

1. For the model implementations, I wonder if the model performance will be large affected by the dynamic modeling module during fine-tuning stage, such as changing into discrete GNNs or trained with one-step/multiple-step losses?

2. What would be the runtime of the proposed method compared to others?

---

> ### Author Response · Authors · 2024-11-24
> **Response to Reviewer #1 (B9zD) (1/3)**
>
> Thanks for your valuable comments. Here we respond to your comments and address the issues.
>
> ### **Q1. The usage of citation format.**
> Following your suggestion, we modified the format of citations in our updated paper (see the first paragraph of Introduction).
>
> ***
>
> ### **Q2. If the graph structure (edges) are fixed or evolve over time.**
>
> In this paper, we focus on the dynamics systems with fixed interacting graph as an early attempt in our research lines. While, we could also approximate any dynamics with evolving graph structures by substituting the dynamics learner by any specific methods, including both while-box and black-box learners, continuous and discrete learners, one-step and multi-step learners.
>
> ***
>
>
> ### **Q3. About adding baseline methods.**
>
> Following your suggestion, we added GNS as our baseline method and the detailed results are presented below. It's a pity that the source codes of GG-ODE are unavailable until we response the comments. We will try to reproduce GG-ODE and compare with it in the next version.
>
>
> **Results of GNS:**
>
> | **System**      |        | **MSE**   | **MAE**   | **MRAE**       |
> | --------------- | ------ | --------- | --------- | ------------ |
> | Mutualistic     | 10%    | 0.328     | 0.475     | 2.840        |
> |                 | 20%    | 0.520     | 0.609     | 4.281        |
> |                 | 50%    | 0.855     | 0.770     | 3.221        |
> |                 | 70%    | 0.912     | 0.806     | 2.599        |
> |                 | 80%    | 0.930     | 0.817     | 2.405        |
> |                 | 100%   | 0.956     | 0.833     | 2.134        |
> | Heat            | 10%    | 0.483     | 0.545     | 2.613        |
> |                 | 20%    | 0.498     | 0.558     | 3.192        |
> |                 | 50%    | 0.512     | 0.579     | 9.268        |
> |                 | 70%    | 0.518     | 0.587     | 19.715       |
> |                 | 80%    | 0.519     | 0.589     | 22.421       |
> |                 | 100%   | 0.516     | 0.589     | 26.257       |
> | 2D_CFD          | 10%    | 0.486     | 0.483     | 8.452        |
> |                 | 20%    | 0.494     | 0.480     | 8.384        |
> |                 | 50%    | 0.465     | 0.449     | 11.434       |
> |                 | 70%    | 0.444     | 0.429     | 12.306       |
> |                 | 80%    | 0.434     | 0.419     | 12.745       |
> |                 | 100%   | 0.415     | 0.401     | 14.753       |
> | DarcyFlow       | 10%    | 0.005     | 0.050     | 21.489       |
> |                 | 20%    | 0.005     | 0.049     | 20.680       |
> |                 | 50%    | 0.005     | 0.049     | 21.351       |
> |                 | 70%    | 0.005     | 0.049     | 20.978       |
> |                 | 80%    | 0.005     | 0.049     | 21.000       |
> |                 | 100%   | 0.005     | 0.049     | 20.839       |
> | Gene            | 10%    | 0.596     | 0.633     | 1.854        |
> |                 | 20%    | 0.636     | 0.654     | 1.984        |
> |                 | 50%    | 0.648     | 0.650     | 2.199        |
> |                 | 70%    | 0.643     | 0.639     | 2.136        |
> |                 | 80%    | 0.639     | 0.633     | 2.067        |
> |                 | 100%   | 0.631     | 0.622     | 2.111        |
> | ShallowWater    | 10%    | 0.955     | 0.565     | 1.057        |
> |                 | 20%    | 0.978     | 0.572     | 1.049        |
> |                 | 50%    | 0.991     | 0.577     | 1.023        |
> |                 | 70%    | 0.993     | 0.578     | 1.019        |
> |                 | 80%    | 0.994     | 0.578     | 1.017        |
> |                 | 100%   | 0.995     | 0.579     | 1.015        |
> | 2D_DiffReac     | 10%    | 1.168     | 0.850     | 10.476       |
> |                 | 20%    | 1.146     | 0.841     | 6.732        |
> |                 | 50%    | 1.135     | 0.837     | 5.386        |
> |                 | 70%    | 1.133     | 0.837     | 5.156        |
> |                 | 80%    | 1.132     | 0.836     | 4.989        |
> |                 | 100%   | 1.131     | 0.836     | 4.775        |

---

> > ### Author Response · Authors · 2024-11-24
> > **Response to Reviewer #1 (B9zD) (2/3)**
> >
> > | **System**      |        | **MSE**   | **MAE**   | **ND**       |
> > | --------------- | ------ | --------- | --------- | ------------ |
> > | LA              | 10%    | 0.992     | 0.789     | 2.552        |
> > |                 | 20%    | 0.994     | 0.789     | 2.574        |
> > |                 | 50%    | 0.995     | 0.789     | 2.625        |
> > |                 | 70%    | 0.995     | 0.788     | 2.666        |
> > |                 | 80%    | 0.995     | 0.788     | 2.793        |
> > |                 | 100%   | 0.995     | 0.787     | 2.700        |
> > | SD              | 10%    | 1.027     | 0.741     | 4.052        |
> > |                 | 20%    | 1.027     | 0.743     | 3.489        |
> > |                 | 50%    | 1.027     | 0.746     | 3.391        |
> > |                 | 70%    | 1.027     | 0.747     | 3.241        |
> > |                 | 80%    | 1.027     | 0.748     | 3.225        |
> > |                 | 100%   | 1.026     | 0.747     | 3.282        |
> > | NYCTaxi         | 24     | 0.323     | 0.396     | 51.990       |
> > |                 | 48     | 0.330     | 0.400     | 36.132       |
> > |                 | 96     | 0.336     | 0.403     | 43.938       |
> > |                 | 192    | 0.341     | 0.407     | 53.038       |
> > |                 | 336    | 0.341     | 0.407     | 62.130       |
> > |                 | 720    | 0.341     | 0.407     | 62.789       |
> > | CHIBike         | 24     | 0.719     | 0.259     | 15.736       |
> > |                 | 48     | 0.720     | 0.258     | 49.071       |
> > |                 | 96     | 0.720     | 0.258     | 39.272       |
> > |                 | 192    | 0.721     | 0.259     | 53.351       |
> > |                 | 336    | 0.722     | 0.259     | 60.742       |
> > |                 | 720    | 0.723     | 0.259     | 89.368       |
> > | Tdrive          | 24     | 0.225     | 0.266     | 7574.169     |
> > |                 | 48     | 0.250     | 0.283     | 12647.690    |
> > |                 | 96     | 0.269     | 0.296     | 15117.720    |
> > |                 | 192    | 0.284     | 0.306     | 14503.859    |
> > |                 | 336    | 0.309     | 0.324     | 14032.609    |
> > |                 | 720    | 0.350     | 0.352     | 13592.221    |
> > | PEMS03          | 24     | 0.969     | 0.803     | 8.454        |
> > |                 | 48     | 1.022     | 0.826     | 7.937        |
> > |                 | 96     | 1.088     | 0.856     | 7.184        |
> > |                 | 192    | 1.141     | 0.881     | 7.096        |
> > |                 | 336    | 1.120     | 0.871     | 6.993        |
> > |                 | 720    | 1.138     | 0.879     | 8.358        |
> > | PEMS04          | 24     | 1.032     | 0.689     | 4.049        |
> > |                 | 48     | 1.029     | 0.688     | 3.813        |
> > |                 | 96     | 1.027     | 0.687     | 4.201        |
> > |                 | 192    | 1.026     | 0.687     | 4.092        |
> > |                 | 336    | 1.026     | 0.687     | 4.168        |
> > |                 | 720    | 1.026     | 0.687     | 4.317        |
> > | PEMS07          | 24     | 1.104     | 0.826     | 4.080        |
> > |                 | 48     | 1.098     | 0.824     | 4.099        |
> > |                 | 96     | 1.092     | 0.821     | 4.455        |
> > |                 | 192    | 1.090     | 0.820     | 4.471        |
> > |                 | 336    | 1.089     | 0.819     | 4.663        |
> > |                 | 720    | 1.088     | 0.819     | 4.788        |
> > | PEMS08          | 24     | 0.933     | 0.688     | 2.788        |
> > |                 | 48     | 0.934     | 0.688     | 3.150        |
> > |                 | 96     | 0.935     | 0.688     | 3.638        |
> > |                 | 192    | 0.935     | 0.688     | 3.887        |
> > |                 | 336    | 0.935     | 0.688     | 3.784        |
> > |                 | 720    | 0.935     | 0.687     | 3.977        |
> > | NOAA            | 24     | 0.567     | 0.560     | 17.798       |
> > |                 | 48     | 0.603     | 0.580     | 20.415       |
> > |                 | 96     | 0.700     | 0.626     | 22.065       |
> > |                 | 192    | 0.900     | 0.710     | 21.805       |
> > |                 | 336    | 0.914     | 0.715     | 21.309       |
> > |                 | 720    | 0.907     | 0.711     | 21.129       |
> >
> > ***

---

> > > ### Author Response · Authors · 2024-11-24
> > > **Response to Reviewer #1 (B9zD) (3/3)**
> > >
> > > ### **Q4. About the effect of dynamics modeling module on model performance.**
> > >
> > > As presented above, the dynamics modeling module could be substituted into any strong learner. We substitute the GNN-based dynamics learner into the white-box SINDy and the results are still comparable. This indicates that our pre-training process has learnt effective representations for observations, which can benefit dynamics model learning with specific dynamics learner.
> > > Furthermore, We kindly argue that our main contribution lies in pre-training to learn better representations. Analogous to pre-trained language models, our pre-trained \baby concentrates on how to learn better representations. And the dynamics modeling module can be analogous to the classification head or prediction head when fine-tuning a language model for downstream tasks.
> > >
> > > |              | PDEDER|            |           |            | PDEDER+SINDy|            |           |           |
> > > |--------------|------------|------------|-----------|------------|--------------|------------|-----------|-----------|
> > > |              | short-term |            | long-term |            | short-term   |            | long-term |           |
> > > |              | MSE        | MAE        | MSE       | MAE        | MSE          | MAE        | MSE       | MAE       |
> > > | Mutualistic  | 0.362      | 0.452      | 0.809     | 0.675      | 1.014        | 1.014      | 0.334     | 0.334     |
> > > | Heat         | 0.003      | 0.045      | 0.006     | 0.052      | 0.886        | 0.884      | 1.577     | 1.586     |
> > > | 2D CFD       | 0.223      | 0.303      | 0.152     | 0.236      | 1.001        | 0.984      | 1.139     | 1.164     |
> > > | DarcyFlow    | 0.001      | 0.020      | 0.001     | 0.021      | 0.858        | 0.851      | 1.103     | 1.104     |
> > > | Gene         | 0.035      | 0.136      | 0.076     | 0.172      | 0.613        | 0.537      | 0.783     | 0.783     |
> > > | ShallowWater | 0.674      | 0.358      | 1.145     | 0.527      | 0.538        | 0.463      | 1.040     | 1.047     |
> > > | 2D DiffReac  | 0.960      | 0.723      | 1.057     | 0.794      | 0.126        | 0.126      | 0.807     | 0.808     |
> > >
> > >
> > > ***
> > >
> > >
> > > ### **Q5. Runtime comparison.**
> > >
> > > We present the running time comparisons of 1 epoch for each method on all dynamics systems. We can find that our proposal owns higher running speed overall.
> > >
> > > |                | **PDEDER** | **NDCN** | **STGDOE** | **MTGODE** | **GNS** |
> > > | ---------------| -------- | -------- | ---------- | ---------- | ------- |
> > > | mutualistic    | 39s      | 183s     | 69s        | 57s        | 2995s   |
> > > | heat           | 68s      | 193s     | 126s       | 43s        | 2647s   |
> > > | 2D CFD         | 12s      | 17s      | 15s        | 12s        | 358s    |
> > > | DarcyFlow      | 60s      | 98s      | 60s        | 56s        | 1649s   |
> > > | gene           | 50s      | 43s      | 52s        | 49s        | 777s    |
> > > | ShallowWater   | 31s      | 94s      | 60s        | 56s        | 1400s   |
> > > | 2D DiffReac    | 42s      | 32s      | 74s        | 66s        | 625s    |
> > > | LA             | 1s       | 2s       | 3s         | 2s         | 33s     |
> > > | SD             | 1s       | 2s       | 4s         | 2s         | 31s     |
> > > | TDrive         | 53s      | 14s      | 30s        | 23s        | 256s    |
> > > | CHIBike        | 6s       | 17s      | 19s        | 15s        | 282s    |
> > > | NYCTaxi        | 9s       | 79s      | 17s        | 14s        | 210s    |
> > > | PEMS03         | 26s      | 127s     | 110s       | 108s       | 417s    |
> > > | PEMS04         | 15s      | 78s      | 75s        | 61s        | 370s    |
> > > | PEMS07         | 64s      | 136s     | 191s       | 185s       | 493s    |
> > > | PEMS08         | 11s      | 79s      | 65s        | 45s        | 384s    |
> > > | NOAA           | 7s       | 31s      | 23s        | 18s        | 307s    |
> > >
> > >
> > > We hope to hear back from you if you have further questions.

---

> > > > ### Comment · Reviewer_B9zD · 2024-11-26
> > > >
> > > > Thank for the detailed response. My questions are mostly resolved. I hope the authors to incoporate those new results and discussions in the revised version. For the \citep v.s. \cite part, it is also observed in other parts of the paper and please revise them accordingly. Also, I feel it is necessary to have a detailed discussion towards neural simulators including both discrete and continuous approaches, and in general some generalized neural simulators in the paper. I have raised my score to 6.

---

> > > > > ### Author Response · Authors · 2024-12-04
> > > > > **Response to Reviewer B9zD**
> > > > >
> > > > > Thank you so much for raising the score. We will carefully revise and improve our paper as you suggested!

---

### Author Response · Authors · 2024-11-24
**General Response**

We thank all reviewers for your careful consideration. We greatly appreciate the positive comments and address major concerns below.

### **Q1. About the dimension reduction in Data Projection.**

We are grateful to thank all reviewers for pointing out this problem. We have modified this section to make it clearer. In practice, we adopt a system-specific flatten-linear layer $f(\cdot;\mathbf{W}_{dp}^s)$ to align the feature dimensions. The detailed modifications are presented below and we modified in our latest PDF version.

> **Data Projection**
To handle dimension diversity of states across different systems, we adopt a flatten-linear data projection module to align the observations by mapping into same dimensions. For each patched tokens
$\overline{\mathbf{x}} _ {m,n}^{(in)}\in\mathbb{R}^{P _ m \times L _ p \times V _ s}$, we first flatten it into $\overline{\mathbf{x}} _ {m,n}^{(in)(fl)} \in\mathbb{R}^{P _ m \times (L _ p \cdot V _ s) }$ , and then project it by a linear layer into the dimension of $L _ p$ for all systems $\tilde{\mathbf{x}} _ {m,n}^{(in)} = f(\overline{\mathbf{x}} _ {m,n}^{(in)(fl)};\mathbf{W} _ {dp}^{s})$, where $\mathbf{W} _ {dp}^{s}\in\mathbb{R}^{(L _ p \cdot V _ s) \times L _ {p}}$ denotes the system-specific trainable parameters.


***

### **Q2. About improving the experiments.**

Thanks for all reviewers for helping us improve the empirical studies. Following your valuable comments, we improved our experiments from the following aspects:
+ Adding a baseline method [1];
+ Adding an evaluation metric MRAE;
+ Adding the forecasting visualizations;
+ Adding sensitivity studies on hyper-parameters patch length and stride;
+ Adding a white-box dynamics learner SINDy [2] on downstream fine-tuning to express the interpretability of embeddings generated by our proposed pre-trained embedder PDE\textsc{der}\xspace.
+ Adding the runtime comparisons over baselines.
+ Renaming the "Ablative Study'' into "Impact Evaluation of Pre-training on downstream Dynamics Modeling'' and edited its settings.

The details of improved experiments are presented below, and also modified in the updated PDF of our paper.

[1] Sanchez-Gonzalez, Alvaro, et al. Learning to simulate complex physics with graph networks. In International conference on machine learning. 2020: 8459-8468.

[2] Brunton S L, Proctor J L, Kutz J N. Discovering governing equations from data by sparse identification of nonlinear dynamical systems. Proceedings of the national academy of sciences, 2016, 113(15): 3932-3937.

---

### Meta-Review · Area_Chair_QQ9M · 2024-12-23

**Metareview:**

The paper introduces a method for learning unified latent representations to model the dynamics of multiple physical phenomena. This unified encoding is intended to be utilized within an encode-process-decode framework for modeling temporal or spatio-temporal dynamics. Experiments are conducted on a series of dynamical systems.

In response to the reviewers' comments, the authors enhanced the initial version of the paper and provided additional experimental validation. However, concerns persist regarding the organization and clarity of the technical descriptions, as well as technical contributions, such as the connections between the pre-training objective and the modeling of dynamics. Overall, this is an interesting contribution, but it remains preliminary and requires further refinement before publication.

**Additional Comments On Reviewer Discussion:**

The main concerns pertain to organization, clarity, and technical contributions. Although the authors provided new experimental results, the majority of reviewers found these insufficient.

---

### Decision · Program_Chairs · 2025-01-22

Reject